JCB Journal of Cell Biology

# HSV-1 exploits host heterochromatin for nuclear egress

Hannah C. Lewis[1,2]*, Laurel E. Kelnhofer-Millevolte[1,2,3]*, Mia R. Brinkley[1], Hannah E. Arbach[1], Edward A. Arnold[1,4], Saskia Sanders[5,6,7,8], Jens B. Bosse[5,6,7,8], Srinivas Ramachandran[9,10], and Daphne C. Avgousti[1]

**Herpes simplex virus (HSV-1) progeny form in the nucleus and exit to successfully infect other cells. Newly formed capsids navigate complex chromatin architecture to reach the inner nuclear membrane (INM) and egress. Here, we demonstrate by transmission electron microscopy (TEM) that HSV-1 capsids traverse heterochromatin associated with trimethylation on histone H3 lysine 27 (H3K27me3) and the histone variant macroH2A1. Through chromatin profiling during infection, we revealed global redistribution of these marks whereby massive host genomic regions bound by macroH2A1 and H3K27me3 correlate with decreased host transcription in active compartments. We found that the loss of these markers resulted in significantly lower viral titers but did not impact viral genome or protein accumulation. Strikingly, we discovered that loss of macroH2A1 or H3K27me3 resulted in nuclear trapping of capsids. Finally, by live-capsid tracking, we quantified this decreased capsid movement. Thus, our work demonstrates that HSV-1 takes advantage of the dynamic nature of host heterochromatin formation during infection for efficient nuclear egress.**

## Introduction

Nuclear-replicating viruses must contend with host chromatin to establish a successful infection. Like most DNA viruses, herpes simplex virus (HSV-1) takes advantage of host chromatin factors both by incorporating histones onto its genome to promote gene expression (Knipe and Cliffe, 2008) and by reorganizing host chromatin during infection (Monier et al., 2000; Kulej et al., 2017; Aho et al., 2019). In response to infection, changes to histone modifications on interferon response genes can also hinder HSV-1 gene expression (Johnson et al., 2014). HSV-1 progeny capsids egress from the nucleus by a unique mechanism of budding into the inner nuclear membrane and then fusing with the outer nuclear membrane for further maturation in the cytosol (Roller and Johnson, 2021; Arii, 2021). Thus, the host chromatin that accumulates in the nuclear periphery during HSV-1 infection creates a potential barrier for capsids to egress from the nucleus. The redistribution of host chromatin during infection allows for capsids to traverse the nucleus such that the transport of HSV-1 capsids through chromatin is the rate-limiting step of nuclear egress (Aho et al., 2021). The progress of HSV-1 infection is also associated with areas of less dense chromatin in the nuclear periphery, also

termed channels (Myllys et al., 2016). However, it is not known if these channels are necessary for viral egress, and the mechanisms by which they form are unclear.

Heterochromatin density and subnuclear localization are affected by the presence and ratio of specific histone modifications. In uninfected cells, histone modifications such as trimethylation of histone H3 lysine 27 (H3K27me3) and the histone variant macroH2A1, among others, delineate heterochromatin regions that are largely localized to the nuclear periphery. H3K27me3 is deposited by the EZH2 enzyme (Chi et al., 2010), a member of the polycomb repressive complex (PRC2; Margueron and Reinberg, 2011). This modification is bound by PRC2, leading to the modification of adjacent nucleosomes, which results in the formation of heterochromatin domains that repress transcription. MacroH2A, the largest of the histone variants, consists of a canonical histone fold domain, a small linker region, and a C-terminal 25 kD macrodomain that is thought to protrude from the nucleosome (Gamble and Kraus, 2010). There are three isoforms referred to collectively as macroH2A: macroH2A1.1, macroH2A1.2, and macroH2A2. MacroH2A1.1 and macroH2A1.2 are splice variants of the same gene

[1]Human Biology Division, Fred Hutchinson Cancer Research Center, Seattle, WA, USA; [2]Molecular and Cellular Biology, Graduate Program, University of Washington and Fred Hutchinson Cancer Research Center, Seattle, WA, USA; [3]UW Medical Scientist Training Program, Seattle, WA, USA; [4]Microbiology Graduate Program, University of Washington, Seattle, WA, USA; [5]Institute of Virology, Hannover Medical School, Hannover, Germany; [6]Leibniz Institute of Virology (LIV), Hamburg, Germany; [7]Centre for Structural Systems Biology, Hamburg, Germany; [8]Cluster of Excellence RESIST (EXC 2155), Hannover Medical School, Hannover, Germany; [9]RNA Bioscience Initiative, University of Colorado School of Medicine, Aurora, CO, USA; [10]Department of Biochemistry and Molecular Genetics, University of Colorado School of Medicine, Aurora, CO, USA.

*H.C. Lewis and L.E. Kelnhofer-Millevolte contributed equally to this paper. Correspondence to Daphne C. Avgousti: avgousti@fredhutch.org; Srinivas Ramachandran: srinivas.ramachandran@cuanschutz.edu.



that differ by one exon resulting in a 28-amino acid difference in the macrodomain. MacroH2A1 was found to be downregulated in melanoma (Kapoor et al., 2010), suggesting a key role in the maintenance of genome integrity. Importantly, the loss of macroH2A1 and macroH2A2 results in a significant decrease in heterochromatin in the nuclear periphery observed by electron microscopy (Douet et al., 2017). Furthermore, macroH2A1 also demarcates regions of host chromatin that associate with the nuclear lamina (Fu et al., 2015), termed lamina-associated domains (LADs), highlighting its importance in linking chromatin with the nuclear envelope to support nuclear integrity.

In this study, we visualized HSV-1 nuclear egress by transmission electron microscopy (TEM) and found that capsids reach the inner nuclear membrane in regions of less densely stained chromatin. Therefore, we hypothesized that HSV-1 exploits host heterochromatin dynamics to successfully egress from the nuclear compartment. We examined chromatin structure by TEM in the absence of heterochromatin markers macroH2A1 and H3K27me3 and discovered that peripheral heterochromatin is largely dependent on these marks. We used chromatin profiling of macroH2A1 and H3K27me3 during HSV-1 infection to define the specific host genomic regions bound by these markers and found that they demarcate broad regions of heterochromatin that form in transcriptionally active compartments. Importantly, we found that the loss of macroH2A1 results in significantly lower viral titers but does not impair viral transcription, protein production, or replication in both lab-adapted and clinical isolates of HSV-1. Furthermore, by inhibiting EZH2 deposition of H3K27me3, we found that reduction of H3K27me3 also leads to a significant decrease in viral titers but did not affect viral protein or genome accumulation. Finally, we determined by TEM that loss of macroH2A1 or H3K27me3 results in significantly more viral capsids trapped in the nuclear compartment, pinpointing the importance of heterochromatin dynamics in viral nuclear egress. Our study is the first to demonstrate that HSV-1 infection takes advantage of heterochromatin changes to successfully egress from the nuclear compartment.

## Results

### HSV-1 capsids associate with regions of less dense chromatin
To investigate the journey of HSV-1 capsids to the inner nuclear membrane, we used TEM to image nuclei and examined heterochromatin formation in primary and diploid human foreskin fibroblast (HFF) cells. In uninfected cells, we observed dark staining, characteristic of dense heterochromatin, in the nuclear periphery (Fig. 1 a, arrowhead). Upon infection with HSV-1, we observed capsids interacting with the inner nuclear membrane primarily in regions of low-density chromatin indicated by lighter staining (Fig. 1 b, arrows). These results are consistent with a previous report in African green monkey kidney cells (Vero) showing that viral capsids can reach the inner nuclear membrane via channels in the marginalized chromatin (Aho et al., 2017; Myllys et al., 2016). Because heterochromatin in the nuclear periphery was dependent on macroH2A presence in hepatoma cells (Douet et al., 2017) and macroH2A1 commonly overlaps with H3K27me3 (Ghiraldini et al., 2021), we chose to

examine heterochromatin formation in the absence of these markers in HFFs.

### Heterochromatin markers macroH2A1 and H3K27me3 support densely stained regions in the nuclear periphery
We used CRISPR-Cas9 to knock-out (KO) expression of the *macroH2A1* gene in hTERT-immortalized HFF cells (HFF-T) cells, which resulted in the loss of total macroH2A1 (Fig. 1 c), termed macroH2A1 KO cells. To reduce H3K27me3 levels, we targeted the EZH2 enzyme that deposits this mark (Rickels et al., 2016) using a well-characterized inhibitor called tazemetostat (EPZ-6438; Lue and Amengual, 2018). We treated cells with 10 μM of tazemetostat or DMSO control for 3 d to allow for steady-state reduction of H3K27me3 (Fig. 1 c). We found by TEM that macroH2A1 KO cells and H3K27me3 depleted cells had strikingly less heterochromatin in the nuclear periphery than WT cells (Fig. 1, d and e). We quantified the width of heterochromatin at the nuclear periphery in each cell type through binary thresholding of the intensity of electron-dense regions and found that loss of macroH2A1 results in a significant reduction in peripheral heterochromatin (Fig. 1, f and g). It is important to note that in cells with observed decreased heterochromatin, there is no decrease in total chromatin. These cells are simply not able to condense their chromatin to a degree that would be visible by TEM. Importantly, the genomic regions bound by macroH2A1 and H3K27me3 are sufficiently massive that changes are visible by TEM. Therefore, we next examined the genomic distribution of these markers by chromatin profiling.

### MacroH2A1 and H3K27me3 bind broad regions of the host genome that are redistributed during infection
We hypothesized that macroH2A1 and H3K27me3 are deposited at specific genomic loci on the host genome during infection to promote the formation of heterochromatin. To test this hypothesis, we used CUT&Tag (Kaya-Okur et al., 2019) to profile the genomic localization of macroH2A1 at 4, 8, and 12 h post-infection (hpi) in wild type (WT) and macroH2A1 KO HFF-T cells. MacroH2A1 KO cells showed no expression of total macroH2A1, macroH2A1.1, or macroH2A1.2; however, macroH2A2 levels were unchanged (Fig. S1, b and c). We also examined the chromatin profile of H3K27me3 under these conditions. On the human genome, we observed clear enrichment of macroH2A1 and H3K27me3 compared with IgG in WT cells (Fig. S1, d–f). The enrichment of macroH2A1 and H3K27me3 was observed as large domains, many of which were gained upon viral infection (Fig. 2 a), suggesting that the host landscape is altered upon infection. These gains were reflected in an increase in total protein levels measured by mass spectrometry (Fig. S1 a). Since large regions are not amenable to traditional peak-based analysis, we instead used domain-based analysis. With the minimum domain size of 1 kb, we observed ~50,000 H3K27me3 domains and ~70,000 macroH2A1 domains genome-wide across the conditions we profiled (Fig. S1 f). We observed <10 macroH2A1 domains in the datasets generated from macroH2A1 KO cells, indicating that our algorithm was identifying robust domains (Fig. S1 f). Furthermore, there is a high correlation in macroH2A1 domain enrichments between macroH2A1 CUT&Tag in WT cells, between

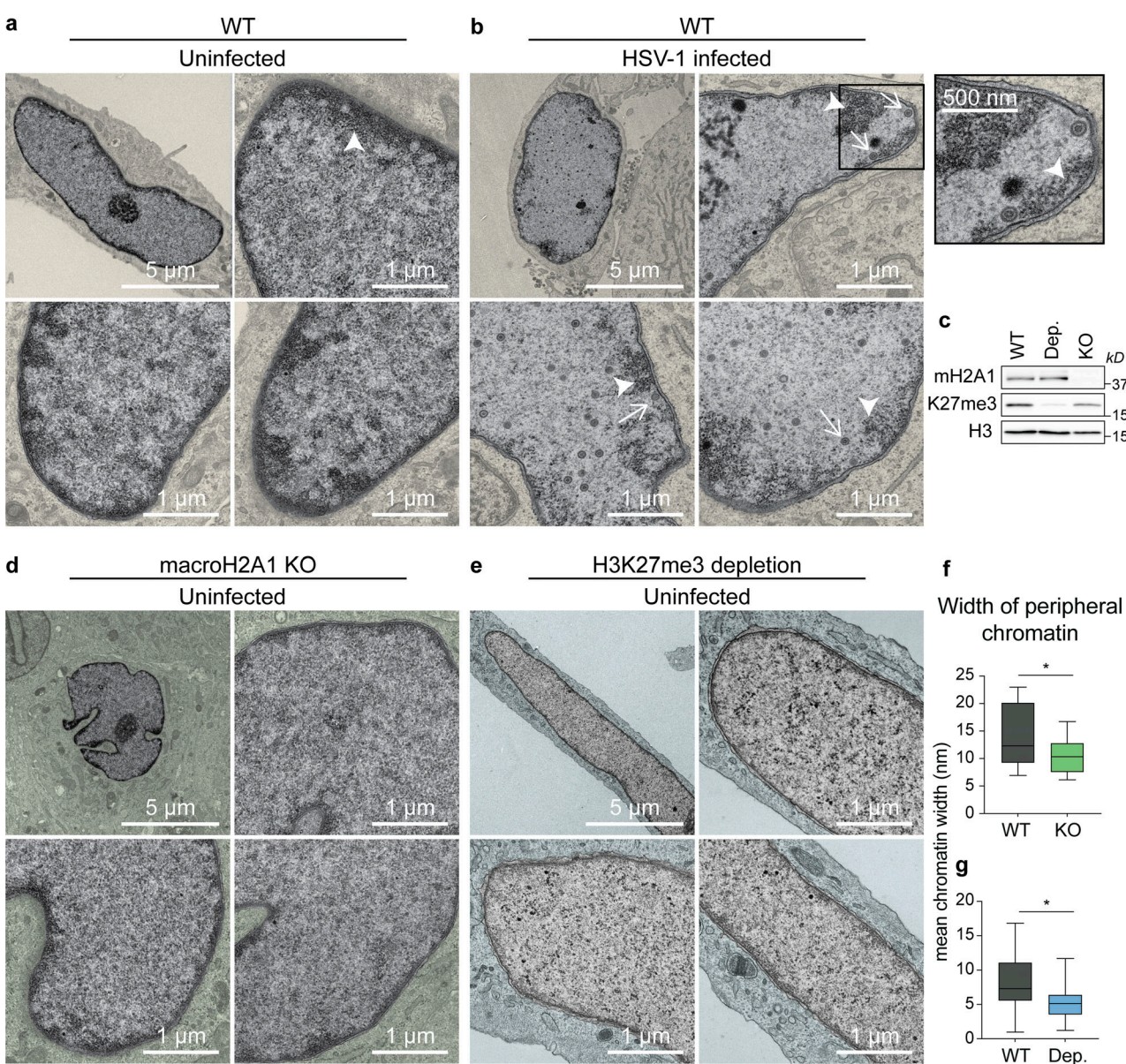

Figure 1. **HSV-1 capsids navigate through regions of less dense chromatin to reach the inner nuclear membrane in HFF cells. (a)** TEM images of representative uninfected nuclei in WT HFF-Ts. Regions outside of the nucleus are colorized yellow. Dark regions represent high-density chromatin (arrowhead). Scale bars as indicated. **(b)** TEM images of representative WT nuclei at 18 hpi with HSV-1. Inset shows an enlarged view of the respective boxed area. The arrowhead indicates high-density chromatin, arrows indicate HSV-1 capsids. **(c)** Representative Western blots with proteins as indicated showing macroH2A1 KO and H3K27me3 depletion (Dep.). **(d)** TEM images of representative uninfected nuclei in macroH2A1 knockout HFF-T cells. Regions outside of the nucleus are colorized green. **(e)** TEM images of representative uninfected nuclei in H3K27me3 depleted conditions. Regions outside of the nucleus are colorized blue. **(f)** Quantification of peripheral heterochromatin width in nuclei from a and d. Width was measured in nm from the nuclear periphery via binary thresholding from intensity profiles sampled every 10 pixels. Mean width was plotted for each nucleus. P = 0.0275 (n = 16 WT, n = 22 macroH2A1 KO) by unpaired t test. For the box plot, the box marks upper and lower quartiles, center line marks median, and error bars denote minimum and maximum values for the population. **(g)** Quantification as in f in nuclei from a and e. P = 0.0144 (n = 22 WT, n = 20 H3K27me3 depleted) by unpaired t test. For the box plot, the box marks upper and lower quartiles, center line marks median, and error bars denote minimum and maximum values for the population. Source data are available for this figure: SourceData F1.

H3K27me3 datasets themselves in both WT and macroH2A1 KO, and between WT macroH2A1 and H3K27me3 in both WT and macroH2A KO cells. MacroH2A1 CUT&Tag from macroH2A1 KO cells has the lowest correlation coefficients in comparison across the board (Fig. S1 g). This implies a clear loss of signal in macroH2A1 CUT&Tag from macroH2A1 KO cells, but concordant signals from macroH2A1 CUT&Tag in WT, and H3K27me3 from

WT and macroH2A1 KO cells at domains were defined just using WT macroH2A1 time series. To identify regions in the genome where macroH2A1 was changing with infection, we first defined non-overlapping sections of the genome where macroH2A1 domains were observed in at least one of the datasets. We then calculated the change in the enrichment of macroH2A1 in these sections across the infection time course. We used k-means

clustering during the time course of infection compared with mock to identify patterns of macroH2A1 gain or loss during infection (Fig. 2 b). With k = 6, we observed two clusters to substantially gain macroH2A1 over the course of infection (clusters 5 and 6, Fig. 2 b left and Fig. 2 c). Clusters 1–3 had significant decreases in macroH2A1, whereas cluster 4 had a minor increase (Fig. 2 b left and Fig. 2 c). We then asked how these macroH2A1-defined clusters behaved with respect to H3K27me3 and found that the overall trends across clusters were preserved in H3K27me3 (Fig. 2 b right and Fig. 2 d), suggesting that H3K27me3 is largely enriched in the same broad regions. Thus, our clustering analysis demonstrates significant redistribution of H3K27me3 and macroH2A1 on the host genome during HSV-1 infection.

To determine whether H3K27me3 deposition was dependent on macroH2A1, we also examined H3K27me3 enrichment in domains in the absence of macroH2A1. We plotted the H3K27me3 changes at the WT clusters for macroH2A1 KO cells and found that the trends of H3K27me3 were similar between WT and macroH2A1 KO cells (Fig. 2, e and f). This result suggests that H3K27me3 deposition is independent of macroH2A1. In summary, HSV-1 infection results in the formation of new heterochromatin domains that span ~10–100 s of kilobases. Importantly, our results demonstrate that heterochromatin accumulating during HSV-1 infection represents new regions of heterochromatin that are macroH2A1- and H3K27me3-dependent.

In contrast, the macroH2A1 CUT&Tag on the viral genome showed similar enrichments in both WT and macroH2A1 KO cells. Further, the H3K27me3 signal on the viral genomes also mirrored the IgG control. These results indicated that there was a significant background signal from the viral genome that could not be accounted for (Fig. S2, a–d). Furthermore, there is a high correlation between all datasets regardless of the antibody or genetic background (Fig. S2 e), implying that there is the same signal from the viral genome regardless of experimental condition, and it most probably represents non-specific tagmentation. We performed immunofluorescence on macroH2A1 KO, H3K27me3-depleted, and respective control cells (Fig. S2, f and g). We observed no crossreactivity of antibodies in macroH2A1 KO and H3K27me3-depleted cells and minimal colocalization of these heterochromatin marks with ICP8 (HSV-1 DNA binding protein [Weller and Coen, 2012]). Therefore, we disregarded viral genome reads in our dataset and focused instead on the host genome. Taken together, these results indicate that multiple forms of heterochromatin were gained at specific host genomic loci during HSV-1 infection.

## MacroH2A1 and H3K27me3 deposition correlates with decreased transcription in active compartments

To investigate the transcriptional output of newly formed macroH2A1 and H3K27me3-bound regions, we performed RNA-seq on WT and macroH2A1 KO cells over the course of HSV-1 infection and calculated fold changes in RNA levels at each time point over the mock control. We identified genes that are contained within domains in each cluster defined in Fig. 2 and plotted the distribution of RNA-fold changes of the genes

grouped by the clusters to which they belonged (Fig. S3, a and c). We found that total RNA levels anti-correlate with macroH2A1 presence: clusters 1–3 had an increase in RNA levels, whereas clusters 4–6 had a decrease in RNA levels over the course of infection (Fig. S3, a and c). Strikingly, the gene expression changes in macroH2A1 KO cells mirror that of WT cells, leading us to conclude that macroH2A1 deposition is not driving changes in total RNA (Fig. S3, b and c). Because macroH2A1 and H3K27me3-bound regions span a large portion of the host genome, we asked whether any specific gene ontology (GO) categories were over-represented for the genes in each macroH2A1-defined cluster (Fig. S3 d). Genes in clusters 1 and 2 (with decreased macroH2A1 and increased expression) were associated with response to dsRNA and inflammatory responses, as expected during viral infection. Surprisingly, clusters 4–6, with increasing macroH2A1, consisted mostly of housekeeping genes. Taken together, these results indicate that macroH2A1 deposition is downstream of transcription changes. Thus, the deposition of macroH2A1 is not directly affecting the expression of specific genes that would be pro- or anti-viral, but rather it is likely that these changes in transcription are a result of the stress response to infection.

To determine whether these changes in RNA reflected changes in transcription, we analyzed published 4sU-RNA-labeling data during HSV-1 infection (Hennig et al., 2018). The time course comparison revealed that in clusters 5 and 6, where we found an increase in macroH2A1 and H3K27me3 presence during infection, the 4sU-labeled RNA decreased at 8 hpi compared with mock, indicating that the gain in heterochromatin correlates with a reduction in active transcription (Fig. S4 a). Anticorrelation between transcription and heterochromatin gain is also seen in clusters 1–3, which feature a loss of macroH2A1 and H3K27me3 and shows a significant increase in active transcription. Interestingly, cluster 4 diverged between total RNA and 4sU-RNA data: 4sU-RNA increased whereas total RNA decreased. This indication of active transcription explains why cluster 4 features a weak macroH2A1 gain. Taken together, these results indicate that the presence of macroH2A1 and H3K27me3 correlates with a decrease in transcription in active regions. Interestingly, this 4sU-RNA labeling dataset also included cells treated with heat shock and salt stress. We analyzed these conditions and found that under these treatments, once again clusters 5–6 had reduced transcript levels (Fig. S4, b and c). These results strongly indicate that the global changes in transcription, which we defined by redistribution of macroH2A1, are a universal stress response.

Finally, we asked if the gain or loss of macroH2A1 happened across previously defined genome compartments (Lieberman-Aiden et al., 2009). We determined the distribution of the eigenvector corresponding to A/B compartments for IMR90 Hi-C data from the 4DN project (Rao et al., 2014). Here, positive values correspond to the A compartment, which features higher gene density and accessible or active chromatin, whereas negative values correspond to the B compartment, which features inactive chromatin. All clusters except cluster 2 are significantly biased toward one of the compartments: clusters 1 and 3 are significantly biased towards compartment B, whereas clusters

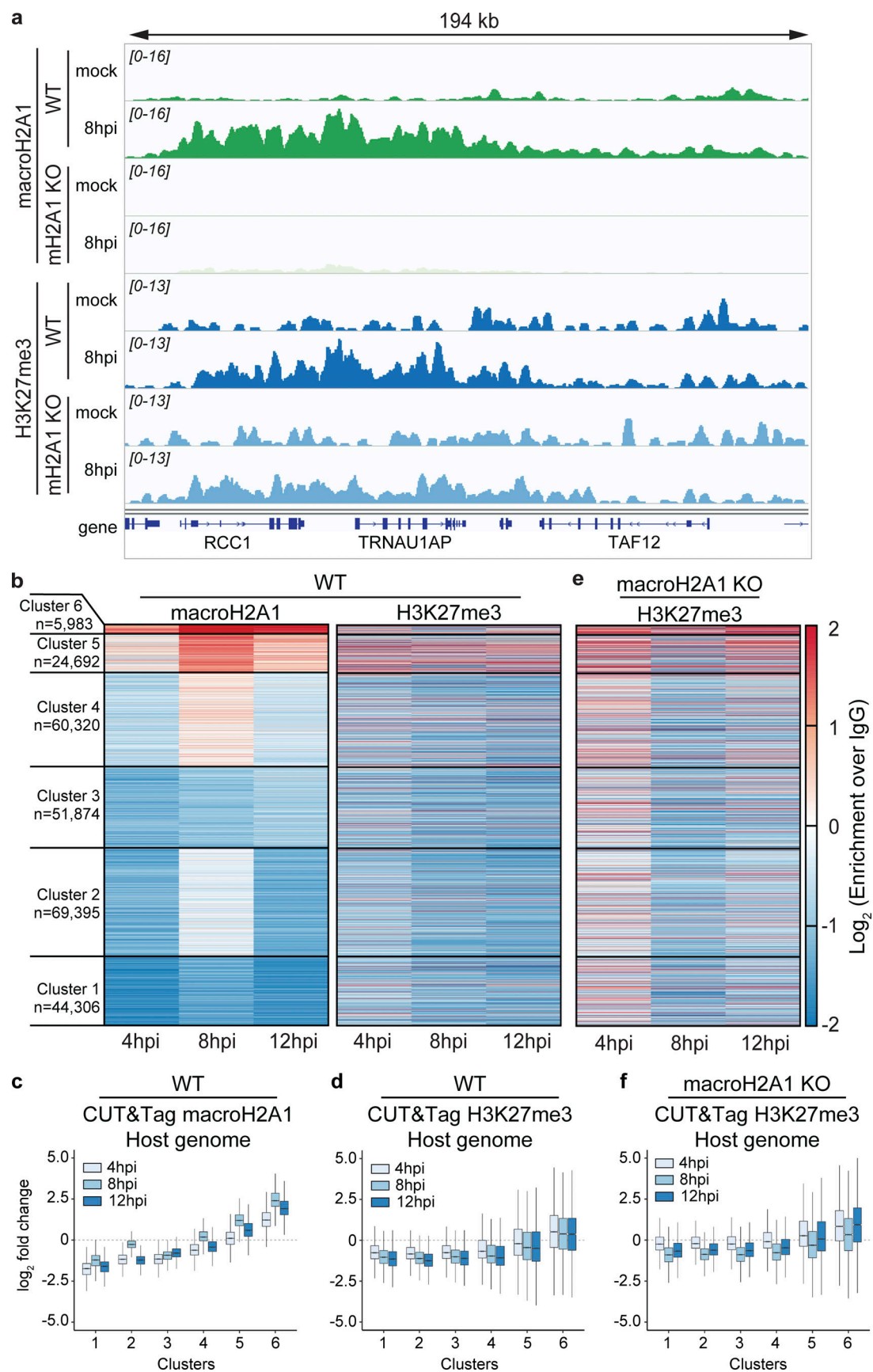

Figure 2. **MacroH2A1 and H3K27me3 bind broad chromatin regions on the host genome that are redistributed over the course of HSV-1 infection. (a)** Representative genome browser snapshots of spike-in normalized CUT&Tag enrichment of macroH2A1 and H3K27me3 showing increases at 8 hpi of HSV-1

infection as measured by CUT&Tag in WT or macroH2A1 KO HFF-T cells as indicated. The region shown is found on chromosome 1. **(b)** Changes in log$_2$ enrichment of spike-in normalized CUT&Tag of macroH2A1 and H3K27me3 over IgG compared with mock treatment are shown as a heatmap. Each line in the heatmap represents a domain of macroH2A1. **(c)** Quantification of heat maps from b showing macroH2A1 enrichment in WT HFF-T cells across each cluster. **(d)** Quantification of heat maps from b showing H3K27me3 enrichment in clusters defined by macroH2A1 in HFF-T cells. **(e)** Changes in log$_2$ enrichment as in b of H3K27me3 over IgG in macroH2A1 KO HFF-T cells. **(f)** Quantification as in c in macroH2A1 KO HFF-T cells.

4–6 are significantly biased toward compartment A (Fig. S4 d). Strikingly, the median of the clusters increases moving from cluster 1 to cluster 6, correlating with macroH2A1 gain. Thus, macroH2A1 gains and losses happen in distinct genomic compartments upon HSV-1 infection. In summary, macroH2A1 and H3K27me3 gain correlates with decreased transcription over the course of HSV-1 infection or stress response in transcriptionally active compartments.

### Loss of macroH2A1 or H3K27me3 results in reduced viral progeny but does not affect viral genome or protein accumulation

To determine the impact of macroH2A1 in HSV-1 infection, we infected WT and macroH2A1 KO HFF-T cells with HSV-1. First, we used RNA-seq to compare viral transcripts in WT and macroH2A1 KO cells. We found no significant change in viral transcripts at 4, 8, or 12 hpi between the two cell types (Fig. 3 a). Next, we examined viral protein accumulation and observed similar levels of ICP0 (immediate early [Cai and Schaffer, 1992]), VP16 (early [Naldinho-Souto et al., 2006]), and glycoprotein H (gH, late [Lorentzen et al., 2001]) in both WT and macroH2A1 KO cells (Fig. 3 b). We also examined the impact of H3K27me3 on HSV-1 infection by treatment with tazemetostat to inhibit EZH2 and deplete H3K27me3 levels (as in Fig. 1). Tazemetostat treatment at 10 µM for multiple days did not impact cell counts or cell viability (Fig. S4, e and f). To examine any synergistic effects of both heterochromatin markers, we treated both WT and macroH2A1 KO cells with tazemetostat. We found that the accumulation of viral proteins ICP0, VP16, and gH were not affected by the reduction of H3K27me3, with or without macroH2A1 (Fig. 3, b–d; and Fig. S4, g and h). Consistent with our findings, a previous report also showed no change in viral transcripts when THP-1 cells were treated with 10 µM tazemetostat prior to infection (Gao et al., 2020). These data indicate that despite changes to heterochromatin dynamics, viral RNA and protein production are not affected by the loss of macroH2A1. Additionally, the depletion of H3K27me3 does not impact viral protein production.

We next examined HSV-1 replication under the same conditions and measured viral genome accumulation by droplet digital PCR (ddPCR). We observed no significant difference in viral genome accumulation between WT and macroH2A1 KO cells (Fig. 3 e, black and green bars), indicating that neither HSV-1 replication nor protein production is affected by macroH2A1 loss. We found that the reduction of H3K27me3 did not significantly affect viral genome accumulation compared with control (Fig. 3 e, black and blue bars). We observed no additional decrease in viral genomes in macroH2A1 KO cells treated with the inhibitor, suggesting that regardless of H3K27me3 levels, loss of macroH2A1 does not impact viral replication as we observed above (Fig. 3 e, striped bars).

To determine the impact of macroH2A1 or H3K27me3 loss on viral progeny production, we measured viral genome accumulation in the supernatant. We discovered a ninefold decrease in the levels of HSV-1 genomes produced in the supernatant of macroH2A1 KO cells compared with those produced from WT cells at 12 hpi (Fig. 3 f, green and black bars). We found a significant fourfold decrease at 12 hpi in tazemetostat-treated cells compared with DMSO-treated cells (Fig. 3 f, blue bars). Because viral genomes measured in the supernatant could also capture defective particles, we also measured infectious progeny by plaque assay. We further discovered that infectious progeny produced from macroH2A1 KO cells was dramatically decreased compared with those produced from WT cells at 8 and 12 hpi (Fig. 3 g, green and black bars). Infectious progeny production was also significantly decreased at 12 hpi in cells with depleted H3K27me3 (Fig. 3 g, black and blue bars). Interestingly, H3K27me3 reduction in macroH2A1 KO cells did not result in any additional defects, suggesting the two markers function in the same genetic pathway (Fig. 3, e–g, striped bars). Furthermore, we did not observe any significant differences in the ratio of genomes to plaque-forming units (pfu) upon loss of macroH2A1, H3K27me3, or both compared with control conditions (Fig. 3 h), indicating that loss of these markers does not result in significantly more defective particles. This effect was consistent across cell types with similar results observed in macroH2A1 KO RPE cells (Fig. S5, a–g). These data indicate that loss of macroH2A1 leads to a significant defect in infectious viral progeny but not viral protein, RNA, or genome accumulation. Similarly, depletion of H3K27me3 results in a significant reduction in infectious progeny but no significant changes in viral protein or genome accumulation. Therefore, we conclude that macroH2A1 and H3K27me3 are important either for proper capsid assembly or efficient viral egress.

### Clinical HSV-1 isolates also require macroH2A1 for progeny production, but not replication or protein production

To determine how robust the requirement for macroH2A1-dependent chromatin is for HSV-1 egress, we used clinical HSV-1 viruses isolated from "low-shedding" or "high-shedding" patients. We measured protein accumulation at 4, 8, and 12 hpi, and viral genome accumulation in cells and supernatant by ddPCR. We found that there was no significant change in viral protein or viral genome accumulation for either clinical isolate in macroH2A1 KO cells (Fig. 4, a–d). We measured viral genomes in the supernatant and found that there was a modest reduction in HSV-1 genomes in the supernatant of macroH2A1 KO cells compared with control cells (Fig. 4 e, green bars). For both clinical isolates, we found that macroH2A1 KO cells produced significantly fewer progeny than WT cells at 12 hpi (Fig. 4 f). Taken together, these results

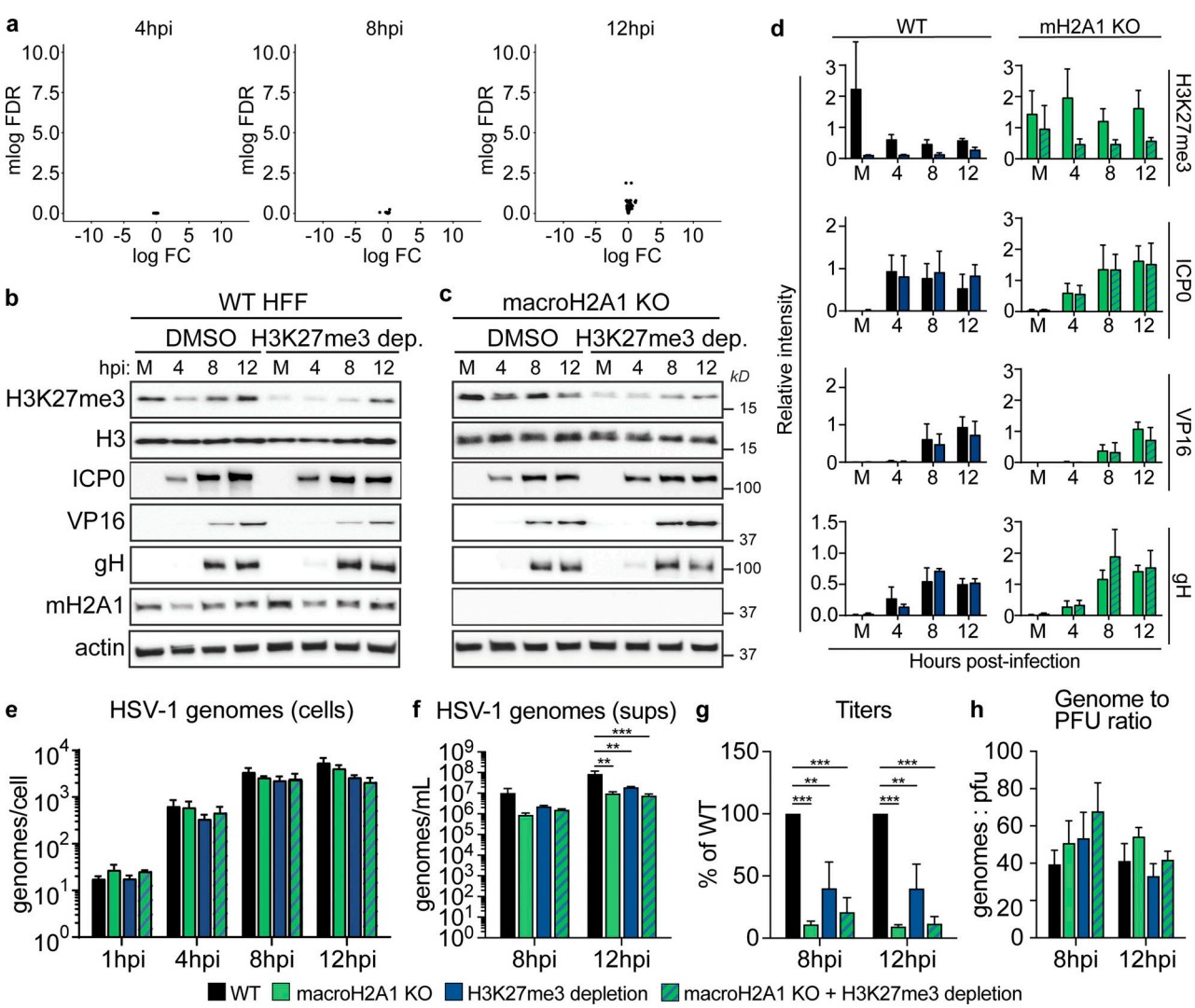

Figure 3. **HSV-1 requires heterochromatin marks macroH2A1 and H3K27me3 for progeny production but not replication or protein production.** **(a)** Volcano plots comparing differential viral RNA levels between WT and macroH2A1 KO HFF-T cells from RNA-seq at 4, 8, and 12 hpi. N = 3 biological replicates. Genes are plotted with log(Fold Change) on x-axis and −1 × log(False Discovery Rate) plotted on the y-axis. Points without significant change in expression are plotted in black, significant reduction in expression are plotted in blue, and significant increase in expression are plotted in red. There are no significantly changing genes. **(b)** Representative Western blots of proteins were shown as indicated upon H3K27me3 depletion or DMSO control–treated HFF-T cells during HSV-1 infection at mock-infected (M) or 4, 8, and 12 hpi. Actin and H3 are shown as loading controls. **(c)** Representative Western blots as in b for macroH2A1 KO HFF-T cells. **(d)** Mean relative intensity of H3K27me3, ICP0, VP16, and gH normalized to H3 quantified from Western blots as in b and c as indicated from WT or macroH2A1 KO. Error bars represent ± SD of three biological replicates. **(e)** Droplet digital (ddPCR) quantification of HSV-1 genomes extracted from infected cells as indicated for each time point. Error bars represent the SEM of three biological replicates. No significance by Dunnett's multiple comparisons test. **(f)** ddPCR quantification of HSV-1 genomes released from cells treated as indicated and isolated from supernatants (sups). Error bars represent the SEM of three biological replicates, **P < 0.01, ***P < 0.001 by Dunnett's multiple comparisons test. **(g)** Infectious progeny produced from HSV-1 infected cells treated as indicated and quantified by plaque assay. Viral yield is indicated as the percent yield compared to WT at each indicated time point. Error bars represent the SEM of three biological replicates, **P < 0.01, ***P < 0.001 by Dunnett's multiple comparisons test. **(h)** Genome to PFU ratio 8 and 12 hpi in indicated conditions as measured by paired plaque assays and ddPCR of genomes isolated from supernatants as in f. Error bars represent the SEM of three biological replicates, no significance by Dunnett's multiple comparison test. Source data are available for this figure: SourceData F3.

indicate that the egress of clinically isolated HSV-1 is also dependent on macroH2A1.

It is important to note that while we measured significantly lower infectious progeny produced from the infected macroH2A1 KO cells, we did not detect a corresponding increase in intracellular genomes. This is likely because we reached the maximum detection limit of our system at 8 hpi such that there is no significant difference in the genomes detected between 8

and 12 hpi (Fig. 3 e and Fig. 4 d), consistent with the notion that chromatin is the bottleneck for HSV-1 egress.

**MacroH2A1 or H3K27me3-dependent heterochromatin is critical for efficient HSV-1 nuclear egress**
Our results indicate that HSV-1 capsids in the nucleus access the nuclear membrane through channels bracketed by highly dense chromatin at the nuclear periphery (Fig. 1), consistent with

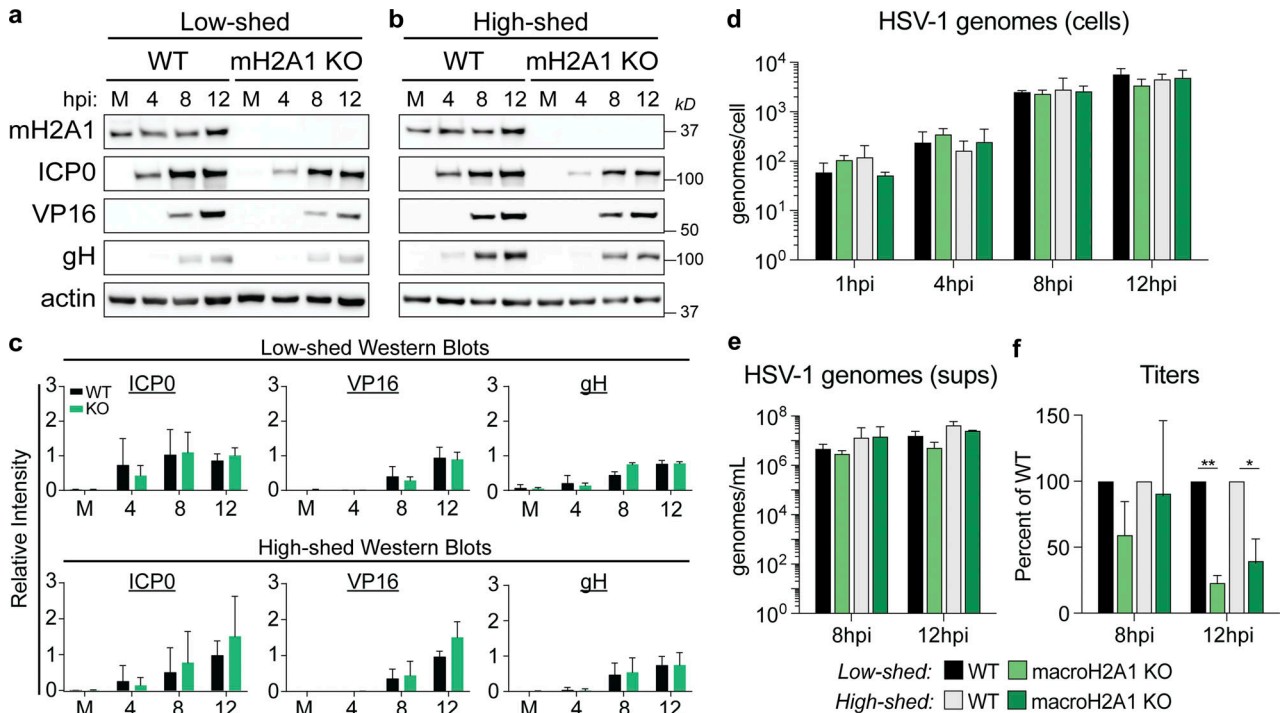

Figure 4. **Clinical HSV-1 isolates also require heterochromatin mark macroH2A1 for progeny production but not replication or protein production.** **(a)** Western blots of proteins as indicated for mock (M) 4, 8, and 12 hpi in WT and macroH2A1 KO cells infected with a low-shedding HSV-1 clinical isolate. **(b)** Western blot as in a for a high-shedding HSV-1 clinical isolate. **(c)** Mean relative intensity of ICP0, VP16, and gH from low-shed (top) or high-shed (bottom) clinical isolate Western blots. Error bars represent ± SD of three biological replicates, no significance by unpaired t test. **(d)** ddPCR quantification of HSV-1 genomes extracted from infected cells as indicated for each time point in WT and macroH2A1 KO cells infected with HSV-1 low- and high-shedding clinical isolates. Error bars represent the SEM of three biological replicates. No significance by Tukey's multiple comparisons test. **(e)** ddPCR quantification of HSV-1 genomes released from cells infected with clinical isolates of HSV-1 as indicated and isolated from supernatants (sups). Error bars represent the SEM of three biological replicates. No significance by Tukey's multiple comparisons test. **(f)** Infectious progeny of cells as indicated infected with clinical HSV-1 isolates quantified by plaque assay. Viral yield is indicated as the percent yield compared to WT, error bars represent the SEM of three biological replicates, *P < 0.05, **P < 0.01 by Tukey's multiple comparisons test. Source data are available for this figure: SourceData F4.

previous reports in other cell types (Aho et al., 2017; Myllys et al., 2016). We found that loss of macroH2A1 or depletion of H3K27me3 resulted in decreased heterochromatin in the nuclear periphery (Fig. 1) and caused a decrease in infectious virus progeny released from infected cells (Figs. 3 and 4). Therefore, we hypothesized that efficient HSV-1 egress out of the nuclear compartment requires macroH2A1- or H3K27me3-dependent heterochromatin in the nuclear periphery. To test this, we infected macroH2A1 KO cells, visualized nuclei by TEM, and quantified capsids in the nucleus (Fig. 5 a). We found that infected macroH2A1 KO nuclei have significantly more HSV-1 capsids than those of WT cells (Fig. 5 b), indicating that loss of macroH2A1-dependent heterochromatin is detrimental to efficient nuclear egress. Similarly, we used tazemetostat treatment to deplete H3K27me3 levels to a steady state, infected cells with HSV-1, and visualized capsids in the nuclei of infected cells by TEM. Strikingly, we found that significantly more capsids accumulated at the nuclear membrane compared with control cells (Fig. 5 c). We quantified this difference by counting the number of capsids that accumulated at the inner nuclear membrane (INM) at any one location and found that significantly more capsids lined up at the INM upon H3K27me3 depletion (Fig. 5 d).

To determine whether capsid assembly was impacted by heterochromatin disruption, we quantified capsid type as a proportion of the total capsids from our TEM. Herpesvirus infection produces three subtypes of capsids: A capsids that are empty, B capsids that contain scaffolding proteins, and C capsids that contain viral DNA and are considered the precursor to infectious virions (Gibson and Roizman, 1972; Yu et al., 2003; Fig. 5 e). We quantified the proportion of each capsid type in WT and macroH2A1 KO cells and found no difference in the proportion of each capsid type (Fig. 5 f). We carried out the same analysis upon infection in H3K27me3-depleted cells and similarly found no difference in the proportion of capsid type (Fig. 5 g). Furthermore, we examined the levels of viral nuclear egress complex (NEC) component UL34 to investigate whether the loss of this important factor may explain the decreased titers. We did not observe any change in UL34 levels by Western blot upon loss of macroH2A1 nor depletion of H3K27me3 (Fig. S4, g and h), indicating that this is unlikely to account for the phenotype. Together, these results indicate that neither capsid formation nor NEC component levels are impacted by the disruption of host heterochromatin. Rather, we conclude that it is the ability of the capsids to egress from the nuclear compartment that is dependent on heterochromatin. Taken

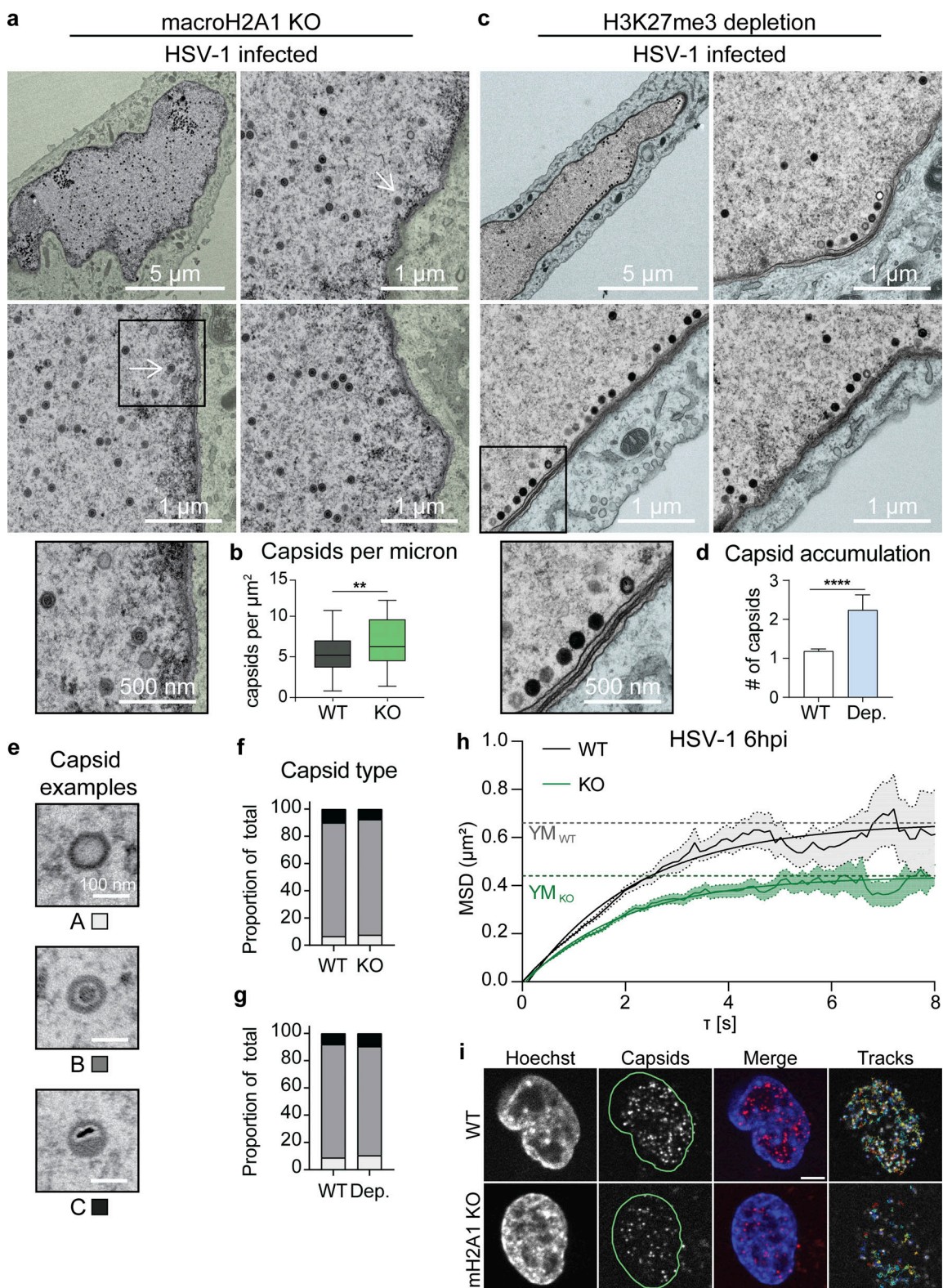

Figure 5. **HSV-1 requires macroH2A1- and H3K27me3-dependent heterochromatin for movement through host chromatin to access the inner nuclear membrane (INM). (a)** TEM images of representative macroH2A1 KO HFF-T nuclei infected with HSV-1 at 18 hpi. Insets show enlarged views of the respective box. Arrows indicate HSV-1 capsids. Scale bars as indicated. **(b)** Quantification of capsids within nuclei compared to Fig. 1 b. Number of capsids was normalized according to nucleus area in µm², P = 0.0036 (n = 40 WT, n = 55 mH2A1 KO) by unpaired t test. For the box plot, the box marks upper and lower quartiles, center line marks median, and error bars denote minimum and maximum values for the population. **(c)** TEM images of representative H3K27me3 depleted nuclei infected with HSV-1 presented as in a. **(d)** Quantification of capsids accumulating at the INM. The number of capsids within 200 nm of the membrane scored per chain as capsids within 300 nm of another capsid, P = 0.0008 by Mann-Whitney test (n = 61 for DMSO, n = 51 for H3K27me3 depleted). Error bars

represent ± SEM of the population. **(e)** TEM images of representative A (empty), B (intermediate; scaffolding proteins, but no genome), and C (full) HSV-1 capsids. Scale bars as noted. **(f)** Quantification of capsid type within nuclei from a in WT and macroH2A1 KO cells. Values for each capsid type are shown as a proportion of total capsids. No significance by Chi-square test, $n$ = 120 capsids per condition. **(g)** Quantification of capsid type as in f in nuclei from c in WT and H3K27me3 depleted cells. No significance by Chi-square test, $n$ = 120 capsids per condition. **(h)** Average mean squared displacement (MSD) plots ± SEM of nuclear capsid tracks in HSV-1 mCherry-VP26 infected WT or macroH2A1 KO HFF-T cells at 6 hpi. Non-linear fits of MSD plots and exponential plateau as dotted lines (YM) are indicated. MSD plots represent 1,148 tracks in six nuclei for WT and 993 tracks in six nuclei for macroH2A1. Plateaus are at $YM_{WT}$ = 0.66 $\mu m^2$ and $YM_{KO}$ = 0.44 $\mu m^2$. **(i)** Representative live-cell images of HSV-1 mCherry-VP26 infected WT or macroH2A1 KO HFF-T nuclei at 6 hpi and corresponding tracks from single-particle tracking used for MSD analysis in h. Scale bar is 5 $\mu m$.

together, these data demonstrate that macroH2A1- and H3K27me3-dependent heterochromatin is critical for efficient nuclear egress.

We and others (Bosse et al., 2015; Aho et al., 2021) have previously shown that infection-induced chromatin modifications promote capsid translocation to the INM. Specifically: (1) HSV-1 infection alters host heterochromatin such that open space is induced at heterochromatin boundaries, termed "corrals," in which viral capsids diffuse, and (2) the movement of viral capsids through the host heterochromatin is the rate-limiting step in HSV-1 nuclear egress. We, therefore, hypothesized that macroH2A1 was critical for the infection-induced formation of open space at heterochromatin boundaries. To test whether the egress defect was a result of restricted capsid movement, we employed live-cell imaging and single-particle tracking to directly measure the effect of macroH2A1-dependent heterochromatin changes on chromatin corral size and viral capsid motility. Using an HSV-1 mutant expressing mCherry-VP26, we found that the mean squared displacement (MSD) of diffusing capsids in macroH2A1 KO cells plateaued at lower levels than in WT cells, indicating a smaller corral size (Fig. 5, h and i). To determine the approximate reduction in corral diameter, we fitted the MSD curves using a non-linear fit with an exponential plateau in the statistical software GraphPad Prism to determine the exact values (see Materials and methods). The plateau height YM was 0.44 $\mu m^2$ for macroH2A1 KO cells and 0.66 $\mu m^2$ for WT cells. We calculated the corral diameter by the following equation (modified from Bosse et al. [2015]):

$$d_{corral} = d_{particle} + 2 \times \sqrt{YM},$$

where $d_{corral}$ is the corral diameter, $d_{particle}$ is the estimated HSV-1 capsid size of 125 nm (Baker et al., 1990), and YM is the plateau of the respective MSD curve. This resulted in 1.75 $\mu m$ mean corral diameter for WT and 1.45 $\mu m$ for macroH2A1 KO cells, indicating that corrals in knockout cells were about 300 nm smaller than in WT cells, restricting capsid diffusion in cellular chromatin. We observed similar limitations in capsid movement in macroH2A1 KO RPE cells with a decrease in corral size of about 165 nm compared with WT cells (Fig. S5, h and i). These results are consistent with the finding that macroH2A1 limits chromatin plasticity both in vitro (Muthurajan et al., 2011) and in cells (Kozlowski et al., 2018) and support our hypothesis that macroH2A1-dependent heterochromatin is critical for the translocation of HSV-1 capsids through the host chromatin to reach the INM. Furthermore, these data support a model in which macroH2A1 supports chromatin rearrangement induced during infection (Fig. 6).

## Discussion

Host chromatin is emerging as a newly appreciated critical barrier to viral success. Elegant studies showed that HSV-1 infection induces host chromatin redistribution to the nuclear periphery (Monier et al., 2000; Myllys et al., 2016). Here, we showed that viral capsids subsequently egress through low-density regions of chromatin indicated through less staining (Fig. 1), consistent with reports in different cell types (Aho et al., 2017). We investigated the impact of heterochromatin markers histone variant macroH2A1 and H3K27me3 on chromatin architecture and HSV-1 infection. We found that loss of macroH2A1 resulted in significantly less heterochromatin visible in the nuclear periphery by TEM in HFF cells (Fig. 1), similar to what was observed in hepatoma cells (Douet et al., 2017). To our knowledge, this is the first instance of TEM imaging of heterochromatin upon depletion of H3K27me3, which produced a similar phenotype to macroH2A1 KO with significantly less heterochromatin in the nuclear periphery (Fig. 1). Visualization of chromatin by TEM is inherently challenging, which is why we carefully compared our chromatin-disrupted conditions only to their respective controls that were processed on the same day and the same time. Our findings are consistent with those reported by other groups, including the importance of macroH2A for nuclear integrity (Fu et al., 2015; Douet et al., 2017), therefore, we are confident in our conclusion that heterochromatin is disrupted in the macroH2A1 KO and H3K27me3-depleted conditions.

We then demonstrated that macroH2A1 and H3K27me3 bind broad regions of the host genome that are redistributed during HSV-1 infection (Fig. 2). Our chromatin profiling analysis required a custom algorithm to accommodate the massive domains created by macroH2A1 and H3K27me3, consistent with regions large enough to be visible by TEM. We demonstrated that the regions that gain macroH2A1 and H3K27me3 are primarily found in transcriptionally active compartments defined by Hi-C in uninfected cells, supporting the idea that newly formed heterochromatin is globally redistributed during HSV-1 infection. Unfortunately, as our controls strongly indicated a lack of specific signal from our viral genome reads, at this time we are unable to make conclusions about the presence or absence of these marks from the viral genome (Fig. S2). Future technical advances will allow for chromatin profiling of viral genomes from single cells to more accurately interpret chromatin marks during infection on the lytic viral genome.

Arbuckle et al. (2017) showed that high dose inhibition of EZH2, the enzyme that deposits heterochromatin mark H3K27me3, caused upregulation of interferon-stimulated genes (ISGs), creating an antiviral state in the cell in the absence of

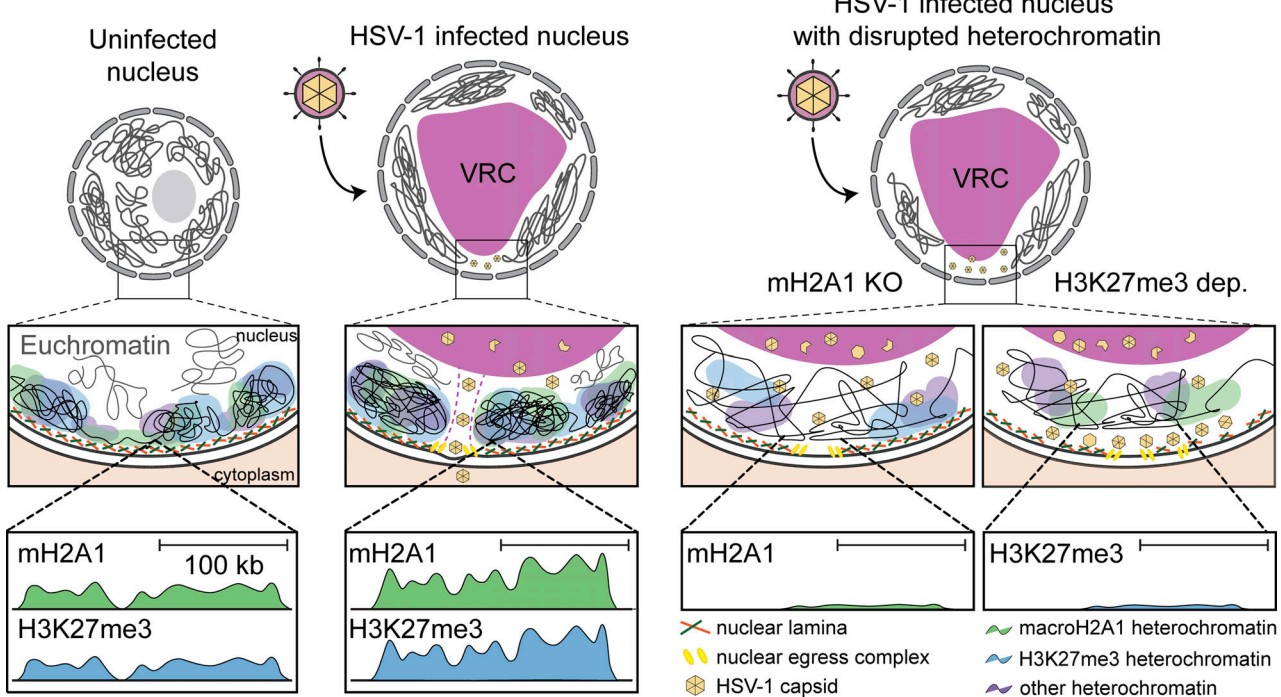

Figure 6. **Model for heterochromatin support of HSV-1 nuclear egress.** Left: Model of previously described chromatin movement during HSV-1 infection. Right: Proposed model of effects of heterochromatin disruption in HSV-1 infection.

infection (Arbuckle et al., 2017). This antiviral state was then shown to prevent infection of several viruses, including HSV-1, adenovirus, HCMV, and ZIKA (Arbuckle et al., 2017). This broad response suggests that it was not specific to HSV-1 but rather a blunt disruption of host heterochromatin to which the cell reacted by initiating defense mechanisms. In contrast, H3K27me3 levels can be decreased to stable levels or a steady-state using a low dose of inhibition over several days. Low-dose inhibition of EZH2 is being developed to target cancer types with EZH2 mutations as these cancers are sensitive to inhibition while WT cells are unaffected (Brach et al., 2017; Knutson et al., 2014). This is important because we can differentiate between blunt force inhibition of EZH2 (i.e., high dose) causing an immune response and may have off-target effects on other methyltransferases, and low-dose inhibition (i.e., steady-state) that decreases total heterochromatin but has little effect on other targets or global cellular processes. Further, low-dose inhibition, as we used, was reported to have no impact on the accumulation of HSV-1 transcripts (Gao et al., 2020). Finally, our analysis of 4sU-RNA generated from heat shock and salt stress demonstrates that there is a decrease in transcription in genes found in macroH2A1-defined clusters upon either stressor (Fig. S4). Thus, our finding that depletion of H3K27me3 does not impact viral protein or genome accumulation is a distinct scenario from what has been previously reported. In addition, the striking phenotype of capsids lined up at the INM suggests a different mechanism of involvement of H3K27me3 compared with macroH2A1 in which capsids are trapped throughout the nucleus. It will be interesting to determine in future work how these two marks work together or in concert to promote nuclear

egress. Together, these results support a model in which host heterochromatin formation creates a structural highway for HSV-1 capsids to successfully access the INM and egress (Fig. 6).

Mining of proteomics on histone marks indicates that several heterochromatin marks including H3K9me3 are also changing during HSV-1 infection (Kulej et al., 2017). This suggests that there may be other mechanisms of heterochromatin dynamics at play. It will be fascinating to consider the diverse implications of heterochromatin domains not only as a means of transcriptional regulation but also as a structural component that impacts many facets of viral infection. Our finding that macroH2A1-defined clusters exhibit decreases in transcription upon heat shock and salt stress suggests that global changes to heterochromatin are likely a consequence of the cell's stress response to infection that HSV-1 exploits to access the INM. Our results strongly indicate that it is the structural aspect of heterochromatin that is important for nuclear HSV-1 egress rather than the transcription of any single gene or a group of genes. Therefore, we propose that the deposition of macroH2A1 and H3K27me3 in specific genomic regions during HSV-1 infection supports the formation of heterochromatin through which HSV-1 capsids travel to reach the inner nuclear membrane. Our study also provides a functional mechanism of how viral capsids can access the INM by diffusion in the absence of an active transport mechanism (Aho et al., 2021; Bosse et al., 2014, 2015). It is important to note that capsids may move more freely in the replication compartments than at the periphery, and our live-tracing measurements were taken in the whole nucleus, including replication compartments and periphery. Because we assessed the average MSD for all particles detected within the confocal volume, we can conclude

## JCB

Table 1. **Cell line sources**

| Cell line | Source |
|---|---|
| Human: RPE-1 (hTERT-immortalized) | Gift of E. Hatch Lab (Hatch et al., 2010) (Invitrogen), Fred Hutch Cancer Center, Seattle, WA, USA |
| Human: Primary HFF | Gift of D. Galloway lab, Fred Hutch Cancer Center, Seattle, WA, USA |
| Human: HFF (hTERT-immortalized) | Gift of J. Kamil Lab (Li et al., 2015), Louisiana State University, Shreveport, LA, USA |
| African green monkey: Vero | Gift of A. Geballe Lab (Child et al., 2021), Fred Hutch Cancer Center, Seattle, WA, USA |

that macroH2A1 knockout has a significant impact on the MSD, which is likely due to a portion of capsids located in the nuclear periphery. When averaged together, the movement of capsids is significantly slower in the absence of macroH2A1-dependent heterochromatin, consistent with our model. At the inner nuclear membrane, capsids are aided by the nuclear egress complex, comprised of viral proteins UL34 and UL31, to bud into the inner nuclear membrane and fuse with the outer nuclear membrane for further maturation in the cytoplasm (Arii, 2021). Budding into the nuclear membrane also requires nuclear lamina disruption. Multiple reports describe that not only does lamina disruption occur independently of capsid formation but also that chromatin is likely the limiting factor in viral egress (Bjerke and Roller, 2006; Simpson-Holley et al., 2004, 2005; Leach and Roller, 2010; Bahnamiri and Roller, 2021; Bigalke et al., 2014; Draganova et al., 2021). Together with our results, this suggests that there are multiple steps in the capsid's journey to the inner nuclear membrane preceding nuclear budding. In summary, our study demonstrates that the contribution of heterochromatin to the successful egress of HSV-1 capsids is a critical aspect of viral infection.

## Materials and methods
### Cells and viruses
Primary HFFs, hTERT-immortalized HFFs, and hTERT-immortalized macroH2A1 knockout HFFs (Table 1) were cultured using standard methods with 10% FBS and 1% penicillin-streptomycin as previously described (Lynch et al., 2021). Cells were grown at 37°C with 5% $CO_2$ and tested for mycoplasma contamination approximately once a month.

The lab-adapted strain of HSV-1, *syn 17+* (Brown et al., 1973), was used for all experiments unless otherwise noted. Monolayers of cells were infected for 1 h at 37°C as previously described (Lilley et al., 2005). An MOI of 3 was used for all experiments, except for EM samples which used an MOI of 10. Cells were collected at mock, 4, 8, and 12 hpi for Western blot. The supernatant was collected at 8- and 12 hpi for plaque assays. HSV-1 infected samples were harvested for TEM at 18 hpi. Virus stock was grown by infecting Vero cells at an MOI of 0.0001. The virus was harvested ~60–80 hpi and titered on U2OS cells to determine stock plaque-forming units per ml (PFU/ml). Experimental plaque assays were set up in Vero cells. Plaque assays were set up on serial 10-fold dilutions in serum-free DMEM. The

virus was left on the cells for 1 h and then aspirated. Cells were washed with 1× PBS (pH 7.46) and 2% methylcellulose overlay in DMEM with 2% FBS, and 0.5% penicillin-streptomycin was added to wells. Plaques were fixed with 0.2% crystal violet between 96 and 100 hpi and plaques were counted by hand. All plaque assays were set up with two technical replicates.

Clinical isolates were acquired in 1994 and 1995 from oral swab collections. Samples were deidentified on collection. Patients provided daily home collections of oral swabs, and qPCR was used to detect how many days HSV was detected in these swabs. Patients that had detectable HSV in only a few swabs were classified as "low-shedders" and patients that had a high percentage of days with positive swabs were classified as "high-shedders." Stocks were a gift from the lab of Keith Jerome.

### Knockout of macroH2A1
MacroH2A1 guide RNA (gRNA, see Table 3) was cloned into TLCv2 (plasmid: 87360; Addgene), a plasmid encoding doxycycline-inducible Cas9-2A-GFP and gRNA expression. To generate lentiviral particles, $1.0 \times 10^7$ HEK293T cells were transfected with 12 µg TLCV2_sg_mH2A1, 8 µg pMDL (plasmid: 12251; Addgene), 4 µg VSVg, and 2.5 µg pREV (plasmid: 115989; Addgene) using Attractene transfection reagent (301005; Qiagen). Lentivirus was harvested at 24-, 48-, and 72-h posttransfection, filtered through 0.2-µm filters, aliquoted, flash-frozen with liquid nitrogen, and stored at –80°C. 2 ml of thawed filtered lentivirus was used to transduce ~$2.25 \times 10^6$ HFF-T or $3 \times 10^6$ RPE cells in 10-cm plates with 8 µg/ml polybrene (TR-1003-G; Millipore-Sigma). Cells were allowed to reach confluence before selection with 1 µg/ml puromycin for 3 d. Cells were then sorted by GFP positivity or counted by hemocytometer, plated at one cell per well into 96-well plates, and selected through serial expansion of colonies. Selected cells were screened for macroH2A1 knockout by Western blot.

### Infections with tazemetostat pretreatment
HFFs and macroH2A1 KO HFFs were treated with DMSO or 10 µM of tazemetostat (HY-13803; MedChem) in DMSO for 3 d prior to infection. Cells were then infected at an MOI of 3, and after 1 h of incubation with the virus, fresh media with 10 µM Taz was added to previously treated cells. Control samples were treated with equivalent volumes of DMSO. Samples were harvested as above.

### Western blotting
Western blotting was performed as previously described (Lynch et al., 2021). Briefly, cells were counted, pelleted, resuspended in 1× NuPAGE lithium dodecyl sulfate (LDS) sample buffer (NP007; Thermo Fisher Scientific) + 5% 2-betamercaptoethanol at 300,000 cells per 200 µl, and boiled for 15 min. Protein lysates were separated by 13.5% SDS-PAGE gels using 1× NuPAGE MOPS buffer (NP0001; Thermo Fisher Scientific) at 75 V for 30 min, then 110 V for 100 min, and then wet transferred to a nitrocellulose membrane (Bio-Rad) at 100 V for 70 min using Transfer Buffer (25 mM Tris Base, 100 mM glycine, 20% methanol). Membranes were ponceau stained and imaged. Membranes were blocked in 5% milk in Tris-buffered saline with Tween (TBST) for 1 h and then probed with primary antibody overnight (see Table 2). Membranes were washed with TBST for 30 min,

incubated with secondary antibodies conjugated to horseradish peroxidase (α-mouse or α-rabbit; 1:5,000) at room temperature for 1 h, washed with TBST for 30 min, and detected using Clarity Western ECL Substrate (1705061; Bio-Rad) and Chemidoc MP Imaging System (Bio-Rad). Images were formatted using Adobe Photoshop and Illustrator. Densitometry analysis was quantified using ImageJ.

Table 2.  **Sources and identification of antibodies used**

### Quantification of HSV-1 genomes by ddPCR

Cells were harvested at the indicated times after infection by trypsinization, washed with 1× PBS, and centrifuged at 5,000 × $g$ for 2 min. Pellets were flash-frozen in liquid nitrogen and stored at –80°C until processed. HSV-1 DNA within cells was isolated from frozen pellets using QIAamp DNAMini Kit (51304; Qiagen).

Supernatants were harvested at the indicated times after infection, centrifuged at >3,500 × $g$, and filtered through 40-μm sterile syringe filters. DNA on the exterior of filtered capsids was digested for 1 h at 25°C with 20.3 units DNase (79254; Qiagen) supplemented with 10 mM $MgCl_2$. DNase was inactivated at 75°C for 10 min followed by vortexing. Capsids were digested with 3 mg/ml proteinase K (BP1700; Thermo Fisher Scientific) in 100 mM KCl, 25 mM EDTA, 10 mM Tris-HCl pH 7.4, and 1% Igepal for 1 h at 50°C. HSV-1 genomes from digested capsids were isolated using QIAamp DNAMini Kit.

A duplexed droplet digital PCR was performed to measure the levels of cellular or supernatant HSV-1 genomes on the QX100 droplet digital PCR system (Bio-Rad Laboratories) using a primer/probe set specific to HSV-1 gB. Cell numbers were determined using a primer/probe set specific to human Beta-globin, a reference gene that exists at two copies per cell. See Table 3 for oligonucleotide sequences, as have been previously established for HSV-1 (Aubert et al., 2014). The ddPCR reaction mixture consisted of 12.5 μl of a 2× ddPCR Supermix for Probes no dUTP (1863024; Bio-Rad), 1.25 μl of each 20× primer-probe mix, and 10 μl of template DNA. 20 μl of each reaction mixture was loaded onto a disposable plastic cartridge (1864008; Bio-Rad) with 70 μl of droplet generation oil (1863005; Bio-Rad) and placed in the droplet generator (Bio-Rad). Droplets generated were transferred to a 96-well PCR plate (12001925; Bio-Rad), and PCR amplification was performed on a Bio-Rad C1000 Touch Thermal Cycler with the following conditions: 95°C for 10 min, 40 cycles of 94°C for 30 s, and 60°C for 1 min, followed by 98°C for 10 min, and ending at 4°C. After amplification, the plate was loaded onto the droplet reader (QX200; Bio-Rad) and the droplets from each well of the plate were automatically read with droplet reader oil (186–3004; Bio-Rad) at a rate of 32 wells per hour. Data were analyzed with QuantaSoft analysis software and the quantitation of target molecules presented as copies per microliter of the PCR reaction. HSV-1 genome values were standardized to cellular β-globin levels. Experiments were completed in biological triplicate and statistical analysis was performed as indicated in figure legends using Prism v10 (GraphPad Software).

### Cleavage under targets and tagmentation (CUT&Tag)

Two biological replicates per time point were obtained from independent infections. The protocol was adapted from the established CUT&Tag methods (Kaya-Okur et al., 2019). Cells were

| Antibody | Source | Identifier | Use (concentration) |
|---|---|---|---|
| Mouse anti-actin | Abcam | Cat: 5441 | WB (1:10,000) |
| Mouse anti-gH | Abcam | Cat: 110227 | WB (1:1,000) |
| Rabbit anti-H3 | Abcam | Cat: 1791 | WB (1:20,000) |
| Rabbit anti-H3K27me3 | Cell Signal Technologies | Cat: 9733T | WB/CUT&Tag (1:100) IF (1:500) |
| Rabbit anti-H3K9me3 | Abcam | Cat: 8898 | WB (1:1,000) |
| Rabbit anti-H4 | Abcam | Cat: 10158 | WB (1:1,000) |
| Mouse anti-ICP0 | Santa Cruz Biotechnology | Cat: 53070 | WB (1:1,000) |
| Mouse anti-ICP8 | Abcam | Cat: 20194 | IF (1:500) |
| Chicken anti-UL34 | Kind gift of Richard Roller, University of Iowa, Iowa City, IA, USA | N/A | WB (1:1,000) IF (1:500) |
| Rabbit anti-macroH2A1 | Abcam | Cat: 37264 | WB (1:1,000) IF (1:500) CUT&Tag (1:100) |
| Rabbit anti-macroH2A1.1 | Cell Signaling Technology | Cat: 12455S | WB (1:1,000) |
| Mouse anti-macroH2A1.2 | Cell Signaling Technology | Cat: 4827S | WB (1:1,000) |
| Rabbit anti-macroH2A2 | Active Motif | Cat: 39874 | WB (1:1,000) |
| HRP-conjugated anti-VP16 | Santa Cruz Biotechnology | Cat: 7546 | WB (1:1,000) |
| Peroxidase-AffiniPure goat anti-mouse | Jackson Immunoresearch Laboratories | Cat: 115-035-003 | WB (1:5,000) |
| Peroxidase-AffiniPure goat anti-rabbit | Jackson Immunoresearch Laboratories | Cat: 111-035-045 | WB (1:5,000) |
| Rabbit anti-mouse IgG | Abcam | Cat: 46540 | CUT&Tag (1:100) |
| Guinea pig anti-rabbit IgG | Antibodies online | Cat: AB1N101961 | CUT&Tag (1:100) |
| Goat anti-rabbit IgG (H+L) AlexaFluor 568 | Thermo Fisher Scientific | Cat: A-11011 | IF (1:300) |
| Goat anti-mouse IgG (H+L) AlexaFluor 488 | Thermo Fisher Scientific | Cat: A-11001 | IF (1:300) |

harvested using trypsin, washed three times with ice-cold phosphate-buffered saline (PBS) via centrifugation at 600 × $g$ for 3 min, and counted using a hemocytometer. Nuclei from 600,000 cells were isolated by hypotonic lysis in 1 ml buffer NE1 (20 mM HEPES-KOH pH 7.9; 10 mM KCl; 0.5 mM spermidine; 0.1% Triton X-100; 20% Glycerol; Roche EDTA-free protease inhibitor) for 10 min on ice followed by centrifugation at 1,300 × $g$ for 4 min. Nuclei were resuspended in Wash buffer (20 mM

**Table 3. Primers used for knockout and ddPCR**

| CRISPR MacroH2A1 gRNA | 5′-CCTCAATAGCAAGCCATCCTGT-3′ |
|---|---|
| HSV1_gB_qPCR_F1 | 5′-CCGTCAGCACCTTCATCGA-3′ |
| HSV1_gB_qPCR_R1 | 5′-CGCTGGACCTCCGTGTAGTC-3′ |
| HSV1_gB_probe1 | 6FAM-5′-CCACGAGATCAAGGACAGCGGCC-3′-BHQ1 |
| beta-globin_qPCR_F1 | 5′-TGAAGGCTCATGGCAAGAAA-3′ |
| beta-globin_qPCR_R1 | 5′-GCTCACTCAGTGTGGCAAAGG-3′ |
| beta-globin_probe2_hex | 5HEX/5′-TCCAGGTGAGCCAGGCCATCACTA-3′/3BHQ_1 |

HEPES-NaOH pH 7.5; 150 mM NaCl; 0.5 mM spermidine; Roche EDTA-free protease inhibitor) and counted using a hemocytometer. BioMag Plus Concanavalin A-coated beads (86057-3; Polysciences) were equilibrated with Binding buffer (20 mM HEPES-KOH pH 7.9; 10 mM KCL; 1 mM CaCl$_2$; 1 mM MnCl$_2$). Beads (5 μl) were mixed with aliquots of 75,000 nuclei and incubated at 25°C for 10 min followed by magnetic separation of beads. Beads were resuspended in 50 μl primary antibody (anti-mH2A1 ab37624; Abcam), anti-H3K27me3 (9733; Cell Signaling), or anti-mouse IgG (ab46540; Abcam) in Wash buffer supplemented with 2 mM EDTA and 0.1% bovine serum albumin (BSA) and incubated on a nutator at 4°C overnight. The beads were decanted on a magnet stand and then resuspended in 50 μl secondary antibody (Guinea pig anti-rabbit IgG [Antibodies-Online: ABIN101961] 1:100) in Wash buffer supplemented with 2 mM EDTA and 0.1% BSA and incubated on a nutator at room temperature for 1 h. The beads were decanted on a magnet stand and washed with 200 μl Wash buffer, resuspended in 50 μl pA-Tn5 (1:200) in 300-Wash buffer (Wash buffer containing 300 mM NaCl), and incubated on a nutator at room temperature for 1 h. The beads were washed twice with 200 μl 300-Wash buffer, resuspended in 50 μl Tagmentation buffer (300-Wash buffer supplemented with 10 mM MgCl$_2$), and incubated at 37°C for 1 h. Beads were then washed with 50 μl TAPS wash buffer (10 mM TAPS pH 8.5, 0.2 mM EDTA), resuspended in 5 μl TAPS wash buffer supplemented with 0.1% SDS, and incubated at 58°C for 1 h. SDS was neutralized on ice with 15 μl 0.67% Triton-X100. 2 μl of 10 mM indexed P5 and P7 primer solutions and 25 μl NEBnext High-Fidelity 2× Master Mix (ME541L; New England BioLabs) were added. Gap-filling and 15 cycles of PCR were performed using an MJ PTC-200 Thermocycler. Library clean-up was performed by incubating beads with 65 μl SPRI bead slurry for 5–10 min, then magnetization and two washes with 200 μl 80% ethanol. Libraries were eluted with 22 μl Tris-HCl pH 8.0 and 2 μl was used for Agilent 4200 Tapestation analysis. The barcoded libraries were mixed to achieve equimolar representation as desired aiming for a final concentration as recommended by the manufacturer for sequencing on an Illumina HiSeq 2500 2-lane Turbo flow cell.

## CUT&Tag data processing
CUT&Tag raw sequencing data were aligned to a custom genome made by concatenating human (hg38), HSV (JN555585.1), and

*Escherichia coli* (U00096.3 *E. coli* str. K-12 substr. MG1655, complete genome). We performed alignments using bowtie2.

## Domain calling
Coverage at 100-bp windows of the human hg38 reference genome was calculated as the number of reads of a given length (120–1,000 bp for the analyses presented here) that mapped at that window normalized by the factor $N$:

$$N = 10,000/(\text{Total number of reads that mapped to } E.\ coli \text{ genome}).$$

Here, 10,000 is an arbitrarily chosen number. We used *E. coli* DNA as a spike-in to normalize all datasets. We partitioned all chromosomes into domains of macroH2A1: domains had an enrichment that was two times the genome-wide median and at least fourfold higher than the IgG control. The normalized coverage at each base-pair from each replicate was averaged when combining multiple replicates. The normalized read density in 100-bp bins was then smoothed with a running average over the bins spanning ±1,000 bp around each bin. We then calculated the genome-wide distribution of normalized read density and medians that were plotted in Fig. S1. We averaged medians across WT macroH2A1 datasets (mock, 4, 8, and 12 hpi) and multiplied the average by 2 to set the cutoff for domain definition for macroH2A1 datasets. We defined a similar cutoff for H3K27me3 datasets using H3K27me3 WT datasets.

## Identifying domain level dynamics of macroH2A1 over time course of infection
To measure changes in macroH2A1 across time points where the domain boundaries are not the same, we first concatenated domain definitions from all macroH2A1 datasets and then defined a set of non-overlapping intervals using the "disjoin" method of GenomicRanges R package (Lawrence et al., 2013). We then calculated the log$_2$ ratio of macroH2A1 enrichment over IgG for the non-overlapping regions for the mock, 4, 8, and 12 hpi. The 4, 8, and 12 hpi enrichments were then divided by the enrichment of the mock dataset to obtain a change in macroH2A1 over time course at the non-overlapping regions. k-means clustering (k = 6) using R Core Team (2019) was performed on the matrix where the rows are the non-overlapping regions and the columns are changes in macroH2A1 over mock at 4, 8, and 12 hpi. We extended the non-overlapping regions by 5 bp on each end and then merged regions within each cluster using the "reduce" method of GenomicRanges to obtain domains in each cluster. We recalculated change in macroH2A1 at 4, 8, and 12 hpi over mock, which was used to plot heatmaps and boxplots shown in Fig. 1 and Fig. S1. H3K27me3 enrichment for WT and macroH2A1 KO cells was calculated at clustered macroH2A1 domains. The unique code for this work can be found here: https://doi.org/10.5281/zenodo.6783949.

## RNA sequencing
Three biological replicates per time point were obtained from independent infections. Cells were harvested at the indicated times after infection by trypsinization, washed with PBS, and centrifuged at 5,000 × *g* for 2 min. Pellets were lysed with TRIzol

(15-596-026; Thermo Fisher Scientific) and total RNA was harvested according to the manufacturer's instructions. RNAs were then treated with DNase (79254; Qiagen) on RNeasy columns (74104; Qiagen) per the manufacturer's instructions. RNA was precipitated with three volumes of ice-cold 96% ethanol, one volume of 3 M sodium acetate pH 5.5, and 1 μl glycogen (R055; Thermo Fisher Scientific) overnight at –80°C. Precipitated RNAs were pelleted at 15,000 × *g* and 4°C for 30 min, washed with ice-cold 75% ethanol, and spun as above for 10 min. RNA was resuspended in nuclease-free water.

RNA was quantified by Nanodrop and integrity was analyzed with the 4200 Tapestation Bioanalyzer system (Agilent). 500 ng of total RNA with an RNA Integrity Number (RIN) >9.5 were used to prepare sequencing libraries with the TruSeq Stranded mRNA Library Prep Kit (20020594; Illumina). Library concentrations were measured with Qubit dsDNA HS Assay Kit (Q32854; Thermo Fisher Scientific) and then analyzed with Agilent High Sensitivity D5000 ScreenTape System and pooled. Libraries were sequenced with 100-bp paired-end reads on an Illumina NovaSeq 6000 SP sequencer at the Fred Hutch Genomics Core Facility.

### RNA/4sU-RNA/Hi-C data processing
Fastq files were filtered to exclude reads that didn't pass Illumina's base call quality threshold. STAR v2.7.1 (Dobin et al., 2013) with two-pass mapping was used to align paired-end reads to a combined reference of human genome build hg38 and HSV1 genome JN555585.1 (https://www.ncbi.nlm.nih.gov/nuccore/JN555585.1/). FastQC 0.11.9 (https://www.bioinformatics.babraham.ac.uk/projects/fastqc/) and RseQC 4.0.0 (Wang et al., 2012) were used for QC including insert fragment size, read quality, read duplication rates, gene body coverage, and read distribution in different genomic regions. FeatureCounts (Liao et al., 2014) in Subread 1.6.5 was used to quantify gene-level expression by strand-specific paired-end read counting.

Gene annotations were based on GENCODE V31 (https://www.gencodegenes.org/human/) and GCA_000859985.2_ViralProj15217 (https://www.ncbi.nlm.nih.gov/data-hub/genome/GCF_000859985.2/). For HSV1, annotated genes were collapsed into non-overlapping transcribed regions, e.g., X indicating a transcribed region unique to gene X, X:Y indicating an overlapping transcribed region for genes X and Y, and so on.

Bioconductor package edgeR 3.26.8 (Robinson et al., 2010) was used to detect differential gene expression between sample groups. Genes with low expression were excluded by requiring at least one count per million in at least *N* samples (*N* is equal to the number of samples in the smaller group). The filtered expression matrix was normalized by the TMM method (https://genomebiology.biomedcentral.com/articles/10.1186/gb-2010-11-3-r25) and subject to significance testing using quasi-likelihood pipeline implemented in edgeR. Genes were deemed differentially expressed if fold changes were >2 in either direction, and Benjamini-Hochberg adjusted P values were <0.01.

The intervals representing the start and end of each gene were intersected with clustered macroH2A1 domains to obtain cluster assignments for genes. The intersection was performed using "intersect" function of bedtools (Quinlan and Hall, 2010).

Genes that did not uniquely intersect with domains in a single cluster were discarded. The cluster assignments for genes were used for plotting total RNA fold changes in Fig. S3. Gene Ontology analysis was performed using WebGestalt (Liao et al., 2019). Reads per kilobase of transcript per million reads mapped (RPKM) values for 4sU-RNA were obtained from GEO (GSE59717) and converted to transcripts per million (TPM). The TPM values across two 4sU-RNA replicates for each condition were averaged, and box plots were generated similarly to total RNA.

For calculating eigenvector distributions, the compartment wig file was downloaded from the following link: https://4dn-open-data-public.s3.amazonaws.com/fourfront-webprod/wfoutput/b543cbf4-ce54-4d2d-8960-281528ff18a6/4DNFI342UZP1.bw.

Regions from compartment file intersecting with each domain cluster were extracted, and the values were used for generating box plots shown in Fig. 2 D.

### Immunofluorescence
Cells were plated on poly-L coated glass coverslips the day prior to infection. Cells were then infected with HSV-1 at an MOI of 3 and collected at mock, 4, 8, and 12 hpi. For harvest, cells were fixed with cold 4% PFA in 1× PBS for 15 min. Cells were permeabilized with 0.5% Triton-X in 1× PBS for 10 min, then blocked in 10% goat serum in 1× PBS for 1 h, incubated with primary antibody (diluted as noted) in 3% BSA in 1× PBS for 1 h. Slides were incubated with secondary antibodies at a dilution of 1:300 in 3% BSA in 1× PBS for 1 h. Coverslips were fixed to microscope slides with Invitrogen ProLong Gold Antifade Mountant. Images were taken on Leica Stellaris Confocal with 63× oil objective at room temperature.

### Electron microscopy
Cells were fixed in 2% paraformaldehyde and 2.5% glutaraldehyde in 0.1 M sodium cacodylate buffer (pH 7.3) at 4°C. Fixed cells were rinsed briefly in 1% sucrose in 50 mM cacodylate (pH 7.2), then postfixed on ice for 30 min in a solution of 1% osmium (RT19152; EM Sciences) and 0.8% potassium ferricyanide in 50 mM cacodylate (pH 7.2). Cell pellets were washed twice briefly at 25°C in 1% sucrose in 50 mM cacodylate (pH 7.2) and then washed in three changes of 50 mM cacodylate (pH 7.2) for 5 min each. Cell pellets were treated with 0.2% tannic acid (1401-55-4; Sigma-Aldrich) in 50 mM cacodylate (pH 7.2) for 15 min at 25°C and then rinsed several times in water. Cells were dehydrated through a graded ethanol series and embedded in Epon 12 resin (18010; Ted Pella). 70-nm thin sections were cut using an Ultracut UC7 ultramicrotome (Leica Mikrosysteme) and collected on 200 mesh formvar/carbon copper grids (01800; Ted Pella). Sections were stained with 2% aqueous uranyl acetate and Reynolds lead citrate. Cell pellet sections were imaged using a Talos L120C microscope operated at 120 kV with a Ceta-16 M (4,096 × 4,096) camera (Thermo Fisher Scientific).

All data were collected at spot size 5 with a 100-μm C2 aperture and 70-μm objective aperture. For quantification, nuclei were targeted at 1,250× and manually circled. Autofocus was set to –2.0 μm. Nuclei were then imaged at 11,000× as a montage and stitched together automatically using SerialEM (Nexperion

Inc). Stitched maps were exported as uncompressed 16-bit tif files for further analysis. For qualitative analysis, images were manually focused to −2.0 μm defocus and then 20 1-s frames were collected and drift-corrected using Velox software (Thermo Fisher Scientific). Final summed images were exported as 16-bit tif files and cropped using ImageJ.

Image analysis pipelines for counting capsids and for measuring chromatin density at the nuclear envelope were deployed in MATLAB R2020b. Scripts are available on the Fred Hutch GitHub repository. Capsid pipeline follows three steps: (1) nuclear boundaries identification, (2) capsid detection, (3) capsid classification. Nuclear boundaries were identified with user input by outlining a freehand contour of the nucleus of interest from which a binary mask is extracted, and its area and perimeter were calculated. Capsid detection was performed on the median-filtered complement of the original image using a Circular Hough Transform-based algorithm, with phase coding for radii estimation (Atherton and Kerbyson, 1999), and search radius ranging from 10.6 to 31.7 nm. Capsids residing within the nuclear mask were then counted and classified. Detected capsids were classified into three categories (empty, intermediate, and full) depending on the distribution of the pixel grayscale intensities within the capsids relative to a normal distribution. Moreover, the distance of the capsids from the nearest nuclear membrane pixel was measured using a distance transform, and capsids within a 200 nm distance from the nuclear membrane were counted.

The width of heterochromatin abutting the nuclear envelope was quantified by measuring the length of the binarized chromatin from 1D intensity profiles along the normal of the nuclear perimeter, sampled at every 10-perimeter pixels. Dense heterochromatin was binarized using global Otsu's thresholding method applied to the background-corrected complement of the contrast-adjusted original image. Noise from the binarized image was further reduced by applying a 2D order statistic filter using the minimum value of a varying domain interactively defined by the user, with a default value of 8-by-8 pixels. The resulting heterochromatin density distribution was normalized to the total length of the nucleus' perimeter.

### Live-cell imaging, tracking, and calculation of the mean squared displacement (MSD)

Cells were infected with HSV-1 VP26mCherry (a kind gift from Beate Sodeik, Medizinische Hochschule Hannover, Hannover, Germany; Sandbaumhüter et al., 2013) and an MOI of 3. Nuclei were stained prior to imaging with Hoechst 33258 (Sigma-Aldrich) diluted 1:1,000 in cell culture media. Cells were imaged at indicated time points on a Nikon TI2 (Nikon) spinning-disk fluorescence microscope equipped with a Yokogawa W2 spinning-disk unit (Yokogawa), an Andor DU-888 X-11633 electron-multiplying charge-coupled device (EMCCD) camera (Andor Technology), and a 100×1.49 numerical aperture (NA) Apo-TIRF objective (Nikon) resulting in 130 nm pixel size. The setup also included 405, 488, 561, and 640 nm laser lines and respective filter sets. Physiological growth conditions were kept constant at 37°C, 5% $CO_2$ using an environmental control system consisting of an objective heater and a humidified incubation chamber.

Cells were continuously imaged for 40 s with a 561-nm wavelength laser in a single plane and a frame rate of 9.5 frame/sec. Single particle tracking was performed in ImageJ Fiji (Schindelin et al., 2012) with the plugin TrackMate v.7.6.1 (Ershov et al., 2022). Tracking settings were as follows: LoG detector, estimated diameter 0.8 μm, quality threshold 50, preprocessing with median filter, sub-pixel localization, spot filter "Signal/Noise ratio" >0.47, simple LAP tracker with the settings: linking max distance 0.5 μm, gap-closing max distance 0.5 μm, gap-closing max frame gap 1, and track filter "Number of spots in the track" >20 spots. MSD analysis was performed using MotilityLab (http://2ptrack.net, accessed 17.02.2023) based on the R-package CelltrackR (Wortel et al., 2021). To calculate the maximum nuclear corral size, a non-linear fit with exponential plateau was fitted to the meanMSD curves using the model $Y=YM-(YM-Y0)\times exp(-k\times x)$ with GraphPad Prism v.9.5.0. Y0 represents the starting population (in $[\mu m^2]$), K determines the rate constant (inverse of X [sec]), and YM represents the plateau (same units as Y). We then calculated the mean corral diameter using the following equation: (modified from Bosse et al. [2015]), where $d_{corral}$ is the corral diameter, $d_{particle}$ is the estimated HSV-1 capsid size of 125 nm (Baker et al.,1990), and YM is the plateau of the respective MSD curve.

### Online supplemental material

Fig. S1 shows the quantification of macroH2A1 and H3K27me3 enrichment on host genomes during HSV-1 infection. Fig. S2 shows quantification of macroH2A1 and H3K27me3 enrichment on viral genomes during HSV-1 infection. Fig. S3 shows that macroH2A1 and H3K27me3 presence on host genomes correlates with decreased transcription. Fig. S4 shows that clusters with more macroH2A1 and decreasing transcription during HSV-1 infection also show less transcription after salt stress or heat shock and correlate with active compartments. Fig. S5 shows that macroH2A1 is also required for efficient viral egress, but not protein production, in RPE cells.

## Data availibility

The data generated are available in the published article and its online supplemental material. The CUT&Tag and RNA-seq data can be found here: Series GSE209820.

## Acknowledgments

We thank members of the Avgousti lab, M. Emerman, A. Geballe, M. Lagunoff, K. Lynch, S. Parkhurst, and M. Weitzman for their insightful comments. We thank D. Janssens and the Henikoff lab for the reagents and assistance with CUT&Tag. We thank the Salama and Overbaugh labs for assistance with ddPCR. We thank the Jerome lab for the clinical HSV isolates and the Cellular Imaging, Electron Microscopy (EMSR), F. Wu and Genomics and Bioinformatics Shared Resource Facilities at the Fred Hutchinson Cancer Research Center for help with sequencing and data analysis. We thank J. Dubrulle and the Cellular Imaging Shared Resource (CISR) for their help with image analysis.

The EMSR, CISR, and Genomics core are supported in part by the Fred Hutch/University of Washington Cancer Consortium (P30 CA015704). J.B. Bosse is funded by the Deutsche Forschungsgemeinschaft (DFG) under Germany's Excellence Strategy EXC 2155—project no. 390874280, DFG Research Unit FOR 5200 DEEP-DV (443644894, project BO 4158/5-1), and by the Wellcome Trust through a Collaborative Award (209250/Z/17/Z). This study was also supported by start-up funds from the RNA Bioscience Initiative at the University of Colorado School of Medicine (S. Ramachandran), the Fred Hutch (D.C. Avgousti), and National Institutes of Health funding to H.C. Lewis (AI083203), E.A. Arnold (AI083203), S. Ramachandran (GM133434), and D.C. Avgousti (GM133441).

Author contributions: Conceptualization: H.C. Lewis, L.E. Kelnhofer-Millevolte, and D.C. Avgousti; methodology: H.C. Lewis, L.E. Kelnhofer-Millevolte, S. Sanders, J.B. Bosse, and D.C. Avgousti; formal analysis: H.C. Lewis, L.E. Kelnhofer-Millevolte, S. Ramachandran, and D.C. Avgousti; investigation: H.C. Lewis, L.E. Kelnhofer-Millevolte, M.R. Brinkley, H.E. Arbach, E.E. Arnold, S. Sanders, J.B. Bosse, and D.C. Avgousti; resources: H.C. Lewis, L.E. Kelnhofer-Millevolte, M.R. Brinkley, H.E. Arbach, E.E. Arnold, S. Sanders, J.B. Bosse, S. Ramachandran, and D.C. Avgousti; data curation: H.C. Lewis, L.E. Kelnhofer-Millevolte, M.R. Brinkley, S. Sanders, J.B. Bosse, S. Ramachandran, and D.C. Avgousti; writing—original draft: H.C. Lewis, L.E. Kelnhofer-Millevolte, M.R. Brinkley, S. Ramachandran, and D.C. Avgousti; writing—review and editing: H.C. Lewis, L.E. Kelnhofer-Millevolte, M.R. Brinkley, H.E. Arbach, E.E. Arnold, S. Sanders, J.B. Bosse, S. Ramachandran, and D.C. Avgousti; visualization: H.C. Lewis, L.E. Kelnhofer-Millevolte, S. Ramachandran, and D.C. Avgousti; supervision: H.C. Lewis, L.E. Kelnhofer-Millevolte, J.B. Bosse, and D.C. Avgousti; funding acquisition: H.C. Lewis, E.A. Arnold, J.B. Bosse, S. Ramachandran, and D.C. Avgousti.

Disclosures: The authors declare no competing interests exist.

Submitted: 28 April 2023

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

# Supplemental material

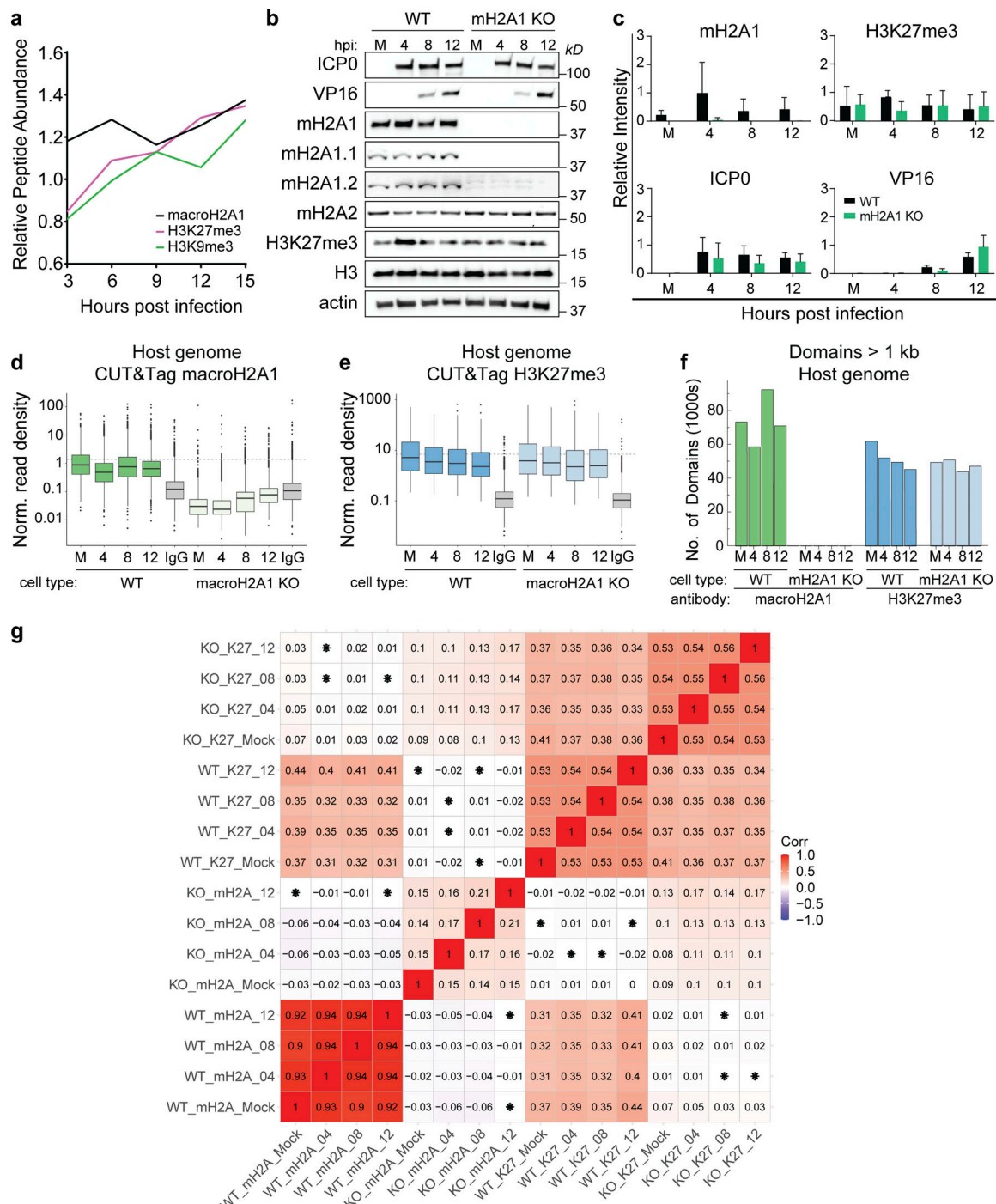

Figure S1. **Quantification of macroH2A1 and H3K27me3 enrichment on host genomes during HSV-1 infection. (a)** Relative peptide abundance of macroH2A1, H3K27me3, and H3K9me3 over the course of HSV-1 infection in primary HFFs from three biological replicates normalized to mock peptide counts (original data from Kulej et al. [2017]). **(b)** Representative Western blots of total viral and host proteins as indicated at mock (M) 4, 8, and 12 hpi with HSV-1 in WT and macroH2A1 KO HFF-T cells. H3 and actin are shown as loading controls. **(c)** Mean relative intensity of total macroH2A1, H3K27me3, ICP0, and VP16 as indicated. Error bars show ± SD of three replicates of Western blots as in b. **(d)** Spike-in normalized CUT&Tag read density of macroH2A1 on WT and macroH2A1 KO HFF-T host genomes during HSV-1 infection. M indicates mock-infected cells; 4, 8, and 12 indicate 4, 8, and 12 hpi. For box plots, the lower and upper hinges correspond to the first and third quartiles (the 25th and 75th percentiles). The upper whisker extends from the hinge to the largest value no further than 1.5 * IQR from the hinge (where IQR is the interquartile range or distance between the first and third quartiles). The lower whisker extends from the hinge to the smallest value at most 1.5 * IQR of the hinge. Data beyond the end of the whiskers are called "outlying" points and are plotted individually. **(e)** Spike-in normalized CUT&Tag read density of H3K27me3 on WT and macroH2A1 KO host genomes during HSV-1 infection presented as in d. **(f)** Total number of macroH2A1 and H3K27me3 domains over 1 kb as measured by CUT&Tag of WT and macroH2A1 KO host genomes during HSV-1 infection. **(g)** Correlation matrix of host genome enrichment from all datasets at domains as defined in Fig. 2. Pearson correlation coefficient is plotted both as a heatmap and the values are printed inside each cell. If the P value of a correlation coefficient is not significant, there is an asterisk over the number in that cell. Source data are available for this figure: SourceData FS1.

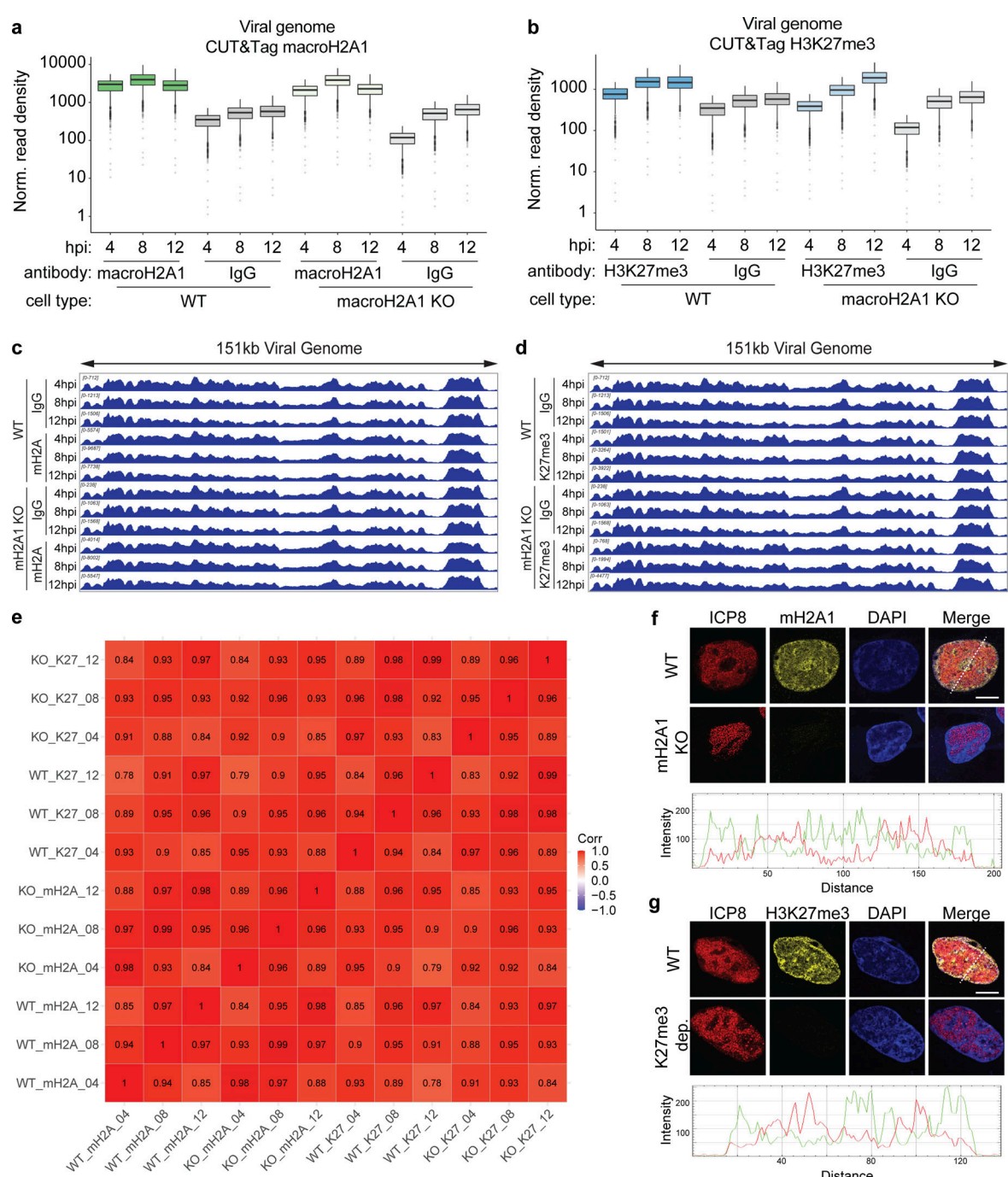

Figure S2. **Quantification of macroH2A1 and H3K27me3 enrichment on viral genomes during HSV-1 infection. (a)** Spike-in normalized CUT&Tag read density of macroH2A1 on HSV-1 genomes during infection of WT and macroH2A1 KO HFF-T cells. The lower and upper hinges correspond to the first and third quartiles (the 25th and 75th percentiles). The upper whisker extends from the hinge to the largest value no further than 1.5 * IQR from the hinge (where IQR is the interquartile range or distance between the first and third quartiles). The lower whisker extends from the hinge to the smallest value at most 1.5 * IQR of the hinge. Data beyond the end of the whiskers are called "outlying" points and are plotted individually. **(b)** Spike-in normalized CUT&Tag read density of H3K27me3 on HSV-1 genomes during infection of WT and macroH2A1 KO HFF-T cells. Data graphed as in a. **(c)** Genome browser snapshots of spike-in normalized CUT&Tag enrichment of macroH2A1 on viral genomes in WT or macroH2A1 KO HFF-T cells as indicated. The full viral genome is shown. **(d)** Genome browser snapshots of spike-in normalized CUT&Tag enrichment of H3K27me3 on viral genomes in WT or macroH2A1 KO HFF-T cells as indicated. The full viral genome is shown. The same IgG control was used to generate IgG WT and KO control graphs for S2c-d. **(e)** Correlation matrix of CUT&Tag signals aligned to the viral genome from all datasets. Pearson correlation coefficient is plotted both as a heatmap and printed inside each cell. If the P value of a correlation coefficient is not significant, there is an asterisk over the number in that cell. **(f)** Top: Representative immunofluorescence of total macroH2A1 (yellow), viral DNA binding protein ICP8 (red), and DAPI (blue) at 10 hpi in WT and macroH2A1 KO HFF-T cells infected with HSV-1. Scale bar is 10 μm. Bottom: Histogram of macroH2A1 (green) and ICP8 (red) intensity at dotted transect line. **(g)** Top: Representative Immunofluorescence of H3K27me3 (yellow), ICP8 (red), and DAPI (blue) at 10 hpi in HSV-1 infected WT and H3K27me3 depleted HFF-T cells. Bottom: Histogram of H3K27me3 (green) and ICP8 (red) intensity at dotted transect line.

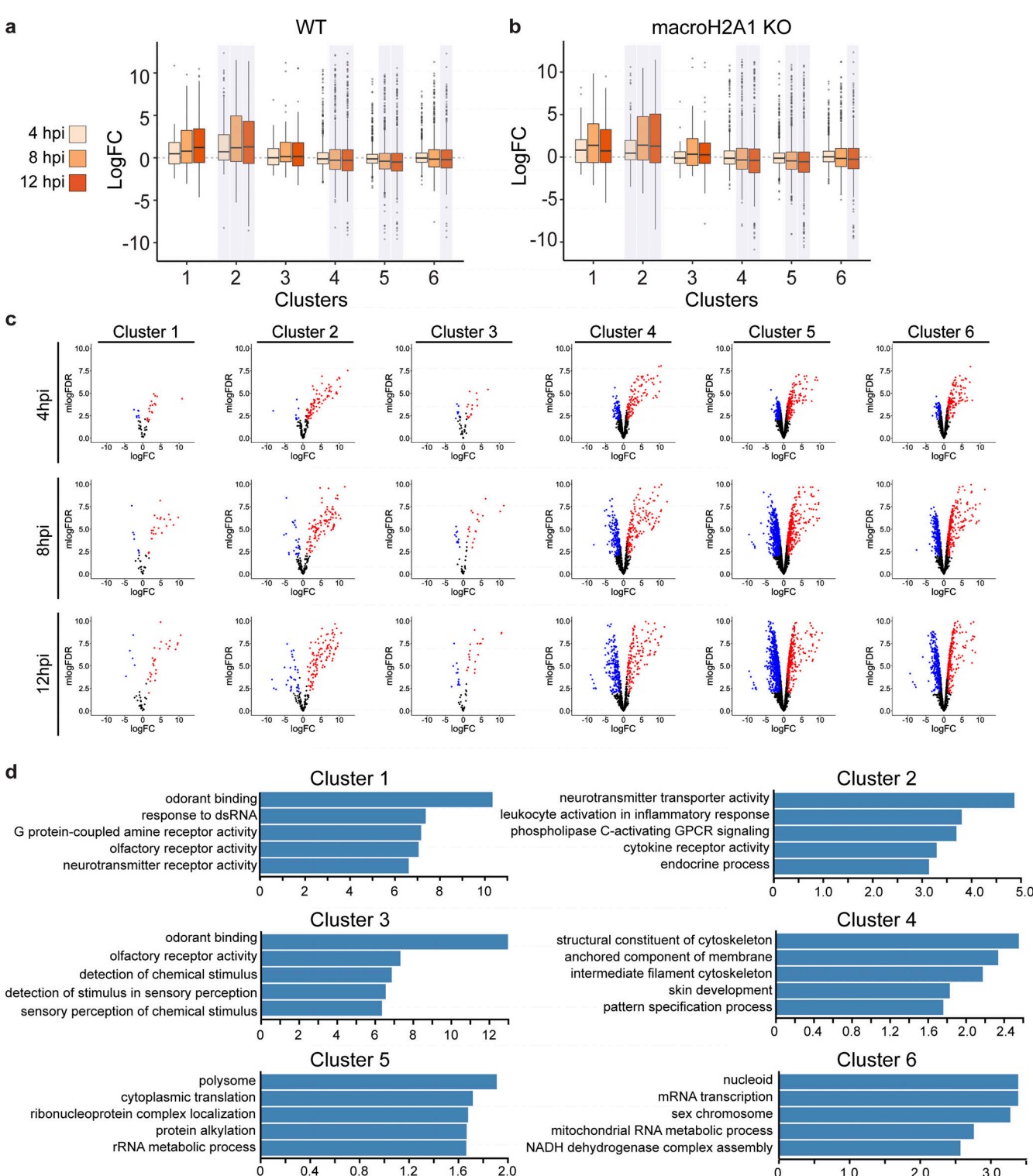

Figure S3. **MacroH2A1 and H3K27me3 presence on host genomes correlates with decreased transcription. (a)** Box plots of total host RNA from RNA-seq in HFF-T WT for genes overlapping clusters from Fig. 2. Kolmogrov–Smirnov test (in R) was performed comparing logCPM values of genes at each time point with logCPM values of genes from the mock dataset for each cluster. The P values were corrected for multiple testing, and the time points and clusters with corrected P value <0.05 are shaded. **(b)** Same as a for macroH2A1 KO HFF-T cells. **(c)** Host differential RNA levels between WT and macroH2A1 KO HFF-T cells from RNA-seq at 4, 8, and 12 hpi by cluster as indicated. N = 3 biological replicates. Volcano plot for host genes in each cluster is plotted as in Fig. 3 a with log(Fold Change) on x-axis and −1 × log(False Discovery Rate) plotted on the y-axis. Points without significant change in expression are plotted in black, significant reduction in expression are plotted in blue, and significant increase in expression are plotted in red. **(d)** Enrichment gene sets from gene ontology (GO) analysis of genes belonging to each cluster as indicated. P value <0.001 and FDR was <0.05 for all shown GO clusters.

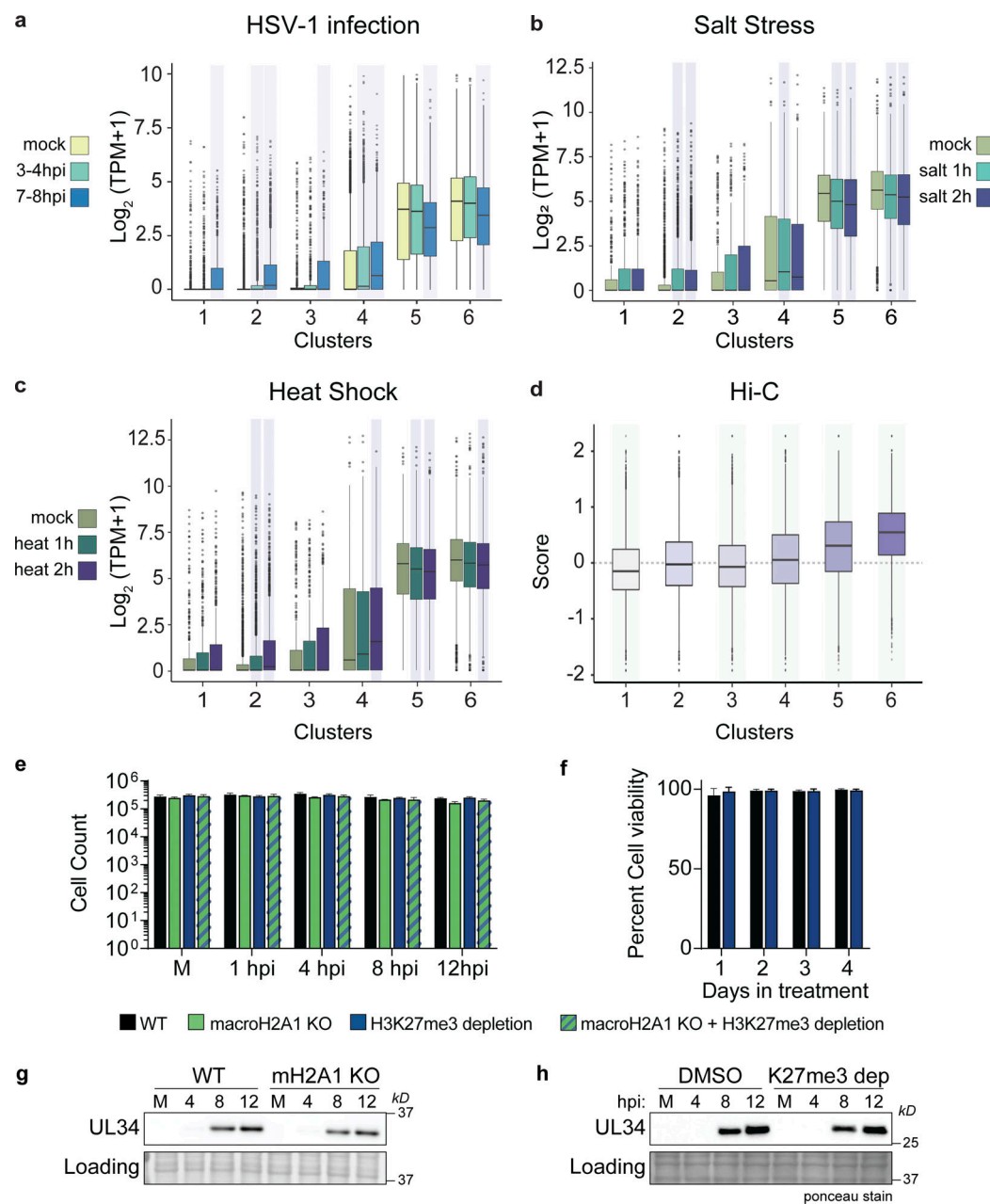

Figure S4. **Clusters with more macroH2A1 and decreasing transcription during HSV-1 infection also show less transcription after salt stress or heat shock and correlate with active compartments. (a)** 4sU-RNA counts for genes from Hennig et al. (2018) overlapping with clusters from Fig. 2 shown as box plots. Kolmogrov–Smirnov test (in R) was performed comparing logCPM values of genes at each time point with logCPM values of genes from the mock dataset for each cluster. The P values were corrected for multiple testing, and the time points and clusters with corrected P value <0.05 are shaded. For all box plots in this figure, lower and upper hinges correspond to the first and third quartiles (the 25th and 75th percentiles). The upper whisker extends from the hinge to the largest value no further than 1.5 * IQR from the hinge (where IQR is the interquartile range, or distance between the first and third quartiles). The lower whisker extends from the hinge to the smallest value at most 1.5 * IQR of the hinge. Data beyond the end of the whiskers are called "outlying" points and are plotted individually. **(b)** Replicate RPKM values from published 4sU-RNA-seq data (Hennig et al., 2018; GSE100469) from HFF treated with 80 mM KCl (increase in final concentration) for 1 and 2 h were converted to TPM, averaged, and intersected with gene lists for each of the six clusters. The distribution of the log₂(TPM+1) for genes in each cluster is shown as boxplots. **(c)** Same as b for heat stress (44°C) also from GSE100469. **(d)** Box plots of Hi-C eigenvector scores of regions overlapping with clusters from Fig. 2. The Hi-C compartment eigenvector scores were obtained from 4DN project website. Wilcoxon signed rank test with alternate hypothesis that the true location is not equal to 0 was performed on the distribution of eigenvector scores for each cluster. The P values were corrected for multiple testing, and the clusters with corrected P value <0.05 are shaded. Bonferroni correction was used for P value adjustments. **(e)** Cell counts for experiments as described in Fig. 3, b–h. N = 3 biological replicates. No significance by paired t-test. Error bars represent the SEM of three biological replicates. **(f)** Cell viability of HFF-T in DMSO or 10 µM Tazemetostat for 4 d posttreatment. N = 3 biological replicates. No significance by paired t test. Error bars represent the SEM of three biological replicates. **(g)** Representative Western blots of HSV-1 protein UL34 in WT and macroH2A1 KO HFF-T cells during HSV-1 infection at mock-infected (M) or 4, 8, and 12 hpi. Actin is shown as the loading control. **(h)** Representative Western blots of HSV-1 protein UL34 in WT and H3K27me3 depleted cells during HSV-1 infection at mock-infected (M) or 4, 8, and 12 hpi. Ponceau stain is shown as the loading control. Source data are available for this figure: SourceData FS4.

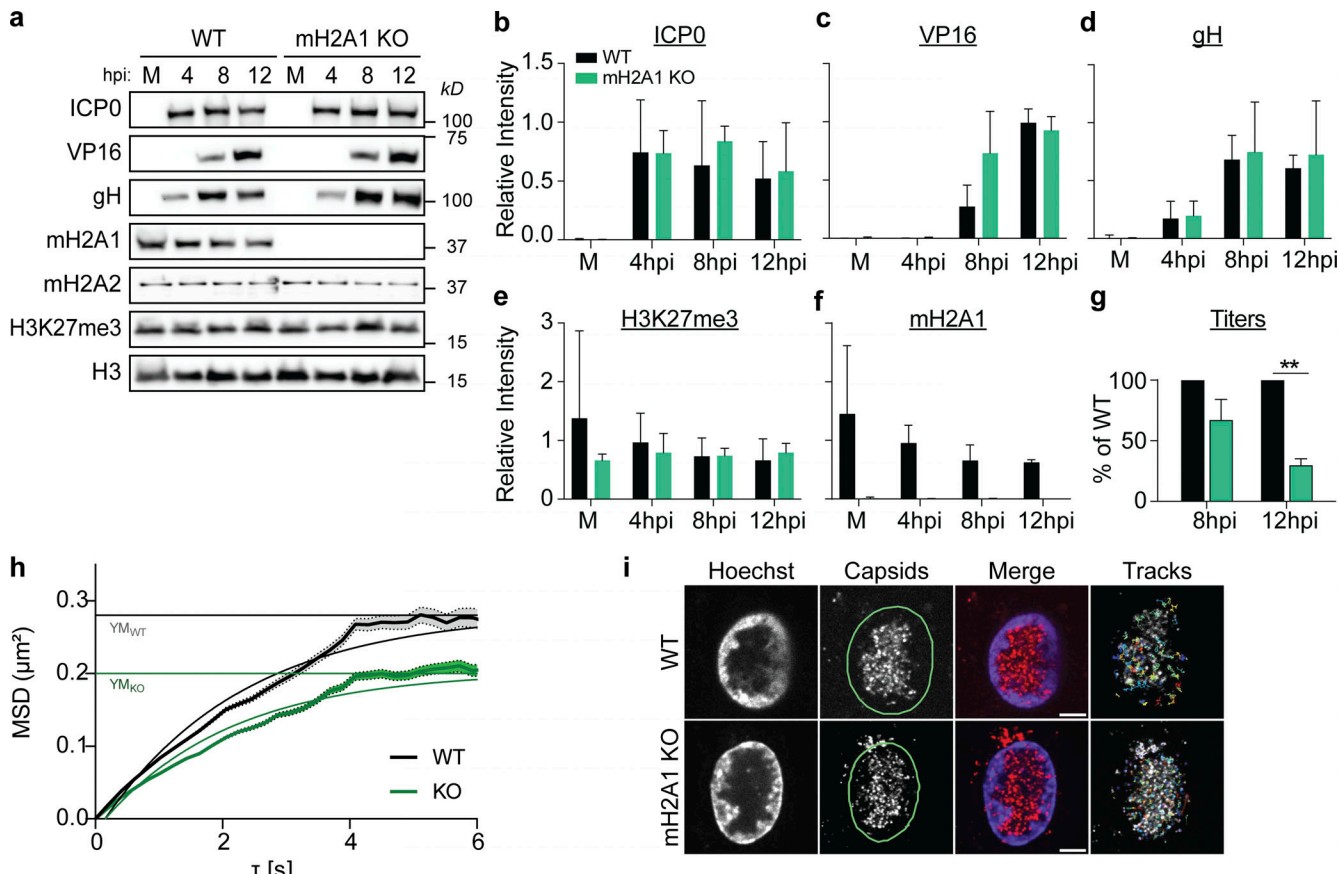

Figure S5. **MacroH2A1 is also required for efficient viral egress, but not protein production, in RPE cells. (a)** Western blot of proteins as indicated in mock(M) or 4, 8, and 12 hpi in WT and macroH2A1 KO RPE cells. H3 is shown as a loading control. **(b)** Mean relative intensity of ICP0 over H3 at indicated time points during HSV-1 infection in WT or macroH2A1 KO RPEs as in a. Error bars represent ± SD from three biological replicates. **(c)** Same as b for VP16. **(d)** Same as b for gH. **(e)** Same as b for H3K27me3. **(f)** Same as b for macroH2A1. **(g)** Infectious progeny produced by WT or macroH2A1 knockout RPE cells quantified by plaque assay on cell-free supernatant. Viral yield calculated as in Fig. 3 g for conditions as indicated, **P < 0.01 by unpaired *t* test. Error bars represent the SEM of three biological replicates. **(h)** Average MSD plots ± SEM of nuclear capsid tracks in HSV-1 mCherry-VP26 infected WT or macroH2A1 KO RPE cells at 8 hpi. Non-linear fits of mean MSD plots and exponential plateau as dotted lines (YM) are indicated. **(i)** Representative live-cell images of HSV-1 mCherry-VP26 infected WT or macroH2A1 KO RPE nuclei at 6 hpi and corresponding tracks from single-particle tracking used for MSD analysis in h. Scale bar is 5 μm. Source data are available for this figure: SourceData FS5.

