## [Peer Review File · The Journal of Cell Biology]

HSV-1 exploits host heterochromatin for nuclear egress

Hannah Lewis, Laurel Kelnhofer-Millevolte, Mia Brinkley, Hannah Arbach, Edward Arnold, Saskia Sanders, Jens Bosse, Srinivas Ramachandran, and Daphne Avgousti

Corresponding Author(s): Daphne Avgousti, Fred Hutchinson Cancer Center and Srinivas Ramachandran, University of Colorado Anschutz Medical Campus

Review Timeline:

Submission Date:	2023-04-28
Editorial Decision:	2023-06-07
Revision Received:	2023-06-13

Monitoring Editor: Andres Leschziner

Scientific Editor: Tim Fessenden

Transaction Report:

DOI: <https://doi.org/10.1083/jcb.202304106>

Revision 0

Review #1

1. Evidence, reproducibility and clarity:

Evidence, reproducibility and clarity (Required)

Summary

This study by Lewis et al. examines the role of heterochromatin in the nuclear egress of herpesvirus capsids. They show that heterochromatin markers macroH2A1 and H3K27me3 are enriched at specific genome regions during the infection. They also show that when macroH2A1 is removed or H3K27me3 is depleted (both of which reduce the amount of heterochromatin at the nuclear periphery), the capsids are not able to egress as effectively. This is interesting since it could be argued that heterochromatin acts as a hindrance to the transport of viral capsids to the nuclear envelope and that the loss of it would allow capsids to reach the nuclear envelope more easily. However, this paper seems to show that heterochromatin formation, on the contrary, is necessary for efficient egress.

Overall, the study seems comprehensive. The methodology is solid, and the experiments are very well controlled. However, some issues need to be addressed before publication.

Major comments

1. In line 49, the authors state, "Like most DNA viruses, herpes simplex virus (HSV-1) takes advantage of host chromatin factors both by incorporating histones onto its genome to promote gene expression and by reorganizing host chromatin during infection". In addition, HSV1 expression can be hindered by the host's interferon response via histone modifications.

Ref. Johnson KE, Bottero V, Flaherty S, Dutta S, Singh VV, Chandran B. IFI16 restricts HSV-1 replication by accumulating on the HSV-1 genome, repressing HSV-1 gene expression, and directly or indirectly modulating histone modifications. PLoS Pathog. 2014 Nov 6;10(11):e1004503. doi: 10.1371/journal.ppat.1004503. Erratum in: PLoS Pathog. 2018 Jun 6;14(6):e1007113. PMID: 25375629; PMCID: PMC4223080.

2. Reference 5 is misquoted in the sentence, "This redistribution of host chromatin results in a global increase in heterochromatin⁵". In that reference, the amount of heterochromatin is not analyzed in any way. However, that particular paper shows that the transport of capsid through chromatin is the rate-limiting step in nuclear egress, which is important considering this study. Further, the article by Aho et al. shows that when the infection proceeds capsids can more easily traverse from the replication compartment into the chromatin, which means that infection can modify chromatin for easier capsid transport. For that reason, the article is an important reference, but it needs to be cited correctly.

3. The term heterochromatin channel at lines 54, 102, and 303 is misleading since the channels seen in the original referred paper are less dense chromatin areas. Also, this term is not used in the original paper where the phenomenon was first described. These less dense interchromatin

channels were found by soft-X-ray tomography imaging and analyses, not by staining.

4. It is difficult to visualize chromatin using TEM microscopy. The values of peripheral chromatin thickness given in Figure 1e (5-15 nm) do not seem realistic given that the thickness of just one strand of histone-wrapped DNA is 11 nm. Why are the two values for WT (in the top and bottom parts) different? If you can get so different values for WT, it is a bit worrisome (switching the WT results between the top and bottom parts of Fig. 1e would for example result in very different conclusions on the effect of macroH2A1 KO for the thickness of the chromatin layer).

5. In lines 134-137 it says that "The enrichment of macroH2A1 and H3K27me3 was observed as large domains that were gained upon viral infection (Fig 2a), suggesting that the host landscape is altered upon infection. These gains were reflected in an increase in total protein levels measured by western blot (Fig 2b)." However, the protein levels of H3K27me3 do not seem to increase during infection. In other presented data as well (Figs. 2a, 2b, 2c, S2a) it is difficult to justify the statement that H3K27me3 is enriched in infection. When this is the case, the conclusion that the amount of heterochromatin increases in the infection (the quotation above and the one in line 315) is not supported. The statement in line 315 is also not specific since it is unclear what "newly formed heterochromatin increases" means.

6. Quantitation of viral capsid location in H3K27me3-depleted cells seems somewhat arbitrary. It would have been more robust to calculate the number of capsids per unit length of the nuclear envelope with and without depletion.

7. In lines 300-302 it says "Elegant electron microscopy work showed that HSV-1 infection induces host chromatin redistribution to the nuclear periphery^{2,8}." However, the redistribution data in reference 8 is based on soft x-ray tomography and not on electron microscopy."

8. The authors bundle together the effects of macroH2A1 removal and H3K27me3 depletion by saying that they both decrease the amount of heterochromatin at the nuclear periphery and therefore hinder capsid egress. This seems overly simplistic and macroH2A1 and H3K27me3 seem to act very differently, which is manifested in the drastic difference in nuclear capsid localization between the two cases. This difference needs to be discussed more.

****Major comments****

Line 45: Nuclear replicating viruses -> Nuclear-replicating viruses

Line 56: is -> are

Line 64: 25kDa -> 25 kDa

Line 159: macroH2A1 cells -> macroH2A1 KO cells

Line 289: The term gDNA is rarely used for viral DNA. Replace gDNA with viral DNA.

Line 405: 8hpi -> 8 hpi

Line 449: mm² -> μm²

"Scale bar as indicated" words can be removed in the figure legends or at least should not be repeated many times within one figure legend.

2. Significance:

Significance (Required)

These findings would appeal to a broad audience in the field of virology. Specifically, the researcher in the fields of virus-cell and virus-nucleus interactions.

This manuscript analyses herpesvirus-induced structural changes in the chromatin structure and organization in the nucleus that are also likely to affect the intranuclear transport of viral capsids.

3. How much time do you estimate the authors will need to complete the suggested revisions:

Estimated time to Complete Revisions (Required)

(Decision Recommendation)

Less than 1 month

No

Review #2

1. Evidence, reproducibility and clarity:

Evidence, reproducibility and clarity (Required)

The manuscript "HSV-1 exploits heterochromatin for egress" describes the effects of heterochromatin at the nuclear periphery, macroH2A1 or H3K27me3 on HSV-1 replication and egress. Knocking out macroH2A1 or depleting H3K27me3 with high concentrations of tazemetostat depleted heterochromatin at the nuclear periphery, may not have affected HSV-1

protein expression and modestly inhibited the production of cell-free infectivity and HSV-1 genomes. macroH2A1 deposition was affected by infection, creating new heterochromatin domains which did not correlate directly with the levels of expression of the genes in them. The authors conclude that heterochromatin at the nuclear periphery dependent on macroH2A1 and H3K27me3 are critical for nuclear egress of HSV-1 capsids.

The experiments leading to the conclusion that HSV-1 capsids egress the nucleus through channels in the peripheral chromatin confirm previously published results (<https://doi.org/10.1038/srep28844>). The previously published EM micrographs show a much larger number of nuclear capsids, more consistent with the images in the classical literature, even in conditions when nuclear egress was not inhibited. Figures 1 and 4 show scarce nuclear capsids, even under the conditions when nuclear egress should be inhibited according to the model and analyses. The large enrichment in nuclear capsids in KO cells predicted by the model is not reflected in figure 4a, which shows only a modest increase in nuclear capsid density (the total number of nuclear capsids would be more informative). The number or density of nuclear capsids is not shown in H3K27 "depleted" cells. The robustness of the analyses of the number of capsids at the membrane in H3K27 "depleted" cells is unclear. For example, the analyses could be repeated with different cut offs, such as 2 or 4. If they are robust, then the conclusions will not change when the cutoff value is changed.

The quantitation of the western blots present no evidence of reproducibility and/or variability. The number of biologically independent experiments analyzed must be stated in each figure and the standard deviation must be presented. As presented, the results do not support the conclusions reached. The quality of western blots should also be improved. It is unclear why figure 2b shows viral gene expression in wild-type cells only, and not in KO or H3K27me3 depleted cells, which are only shown in the supplementary information. These blots presented in Figure S5a and S5b are difficult to evaluate as the signal is rather weak and the controls appear to indicate different loading levels. These blots do not appear to be consistent with the conclusions reached. Some blots (VP16, ICP0 in HFF) appear to indicate a delay in protein expression whereas others (VP16, ICP0 in RPE) appear to indicate earlier expression of higher levels. The claimed "depletion of H3K27me3 is not clear in figure S5d, in which the levels appear to be highly variable in all cases, without a consistent pattern, with no evidence of reproducibility and/or variability, and using a mostly cytoplasmic protein as loading control. All western blots should be repeated to a publication level quality, the number of independent experiments must be clearly stated in each figure, and the reproducibility and/or variability must be indicated by the standard deviation. An enhanced analysis of the RNA-seq data, analyzing all individual genes rather than pooling them together, would provide better support to these conclusions. Then, the western blots are useful to show that the changes in mRNA result in changes in the levels of selected proteins.

Figure S1 raises some questions about the specificity of the macroH2A1 antibody used for CUT&Tag. As expected CUT&Tagging the cellular genome in the KO cells with the specific antibody results in lower signal than with the IgG control antibody. In contrast, viral DNA is CUT&Tagged as efficiently in the KO as in the WT cells, and in both cases significantly above the IgG controls. The simplest interpretation of these results is that the antibody cross-reacts with a protein that binds to HSV-1 genomes. The manuscript must experimentally address this

possibility.

Also, Figure S1 shows that the viral genome is CUT&Tag'ed with H3K27me3 antibody as efficiently in macro H2A1 WT and KO cells, and in both cases above the background signal from IgG control antibody. The authors conclude that the signal with the specific antibody "mirrors" that of the control antibody, but "mirroring" is not defined and the actual data show that there is a large increase in signal with the specific antibody. Not surprisingly, the background signal also increases, as the number of genomes increase while infection progresses. The authors conclude that "these results indicated that there was a significant background signal from the viral genome that could not be accounted for", but no evidence supporting this conclusion is presented. The data show clear signal above the background from the viral genome and that this signal is not affected by the presence or absence of macroH2A1. This section of the manuscript has to be thoroughly re-analyzed as there is clear H3K27 signal.

The concentration of tazemetostat used is high. Normally, concentrations of around 1 μ M are used in cells, and 10 μ M is often cytotoxic (for example <https://doi.org/10.1038/s41419-020-03266-3>; <https://doi.org/10.1158/1535-7163.MCT-16-0840>). The effects on H3K27me3 presented in figure S1b appear to be normalized to mock infected treated cells. If so, they do not allow to evaluate the effectivity of the treatment. Cell viability after the four days treatment must be evaluated, the claimed "depletion" of H3K27me3 must be clearly demonstrated (the blots in figure S5 are not sufficient as presented), and levels of different histone methylations must be tested to support the claimed specificity of tazemetostat for H3K27me3 at the high concentrations used.

****Minor comments.****

Reference No.27 is misquoted in lines 250-251, which state that it shows that "HSV-1 titers, but not viral replication, were reduced upon EZH2 inhibition." The reference actually shows inhibition of HSV-1 infectivity, DNA levels and mRNA for ICP4, ICP22 and ICP27. This reference uses much shorter treatments (12 h and only after infection). It also shows that inhibition of EZH2/1 up regulates expression of antiviral genes.

HFF are primary human cells but they are fibroblasts whereas the primary target of HSV-1 replication is epithelial cells. The wording used "they represent a common site of infection in humans" must be edited

Disruption of macroH2A (1 and 2) results in general defects in nuclear architecture, not just peripheral chromatin (<https://doi.org/10.1242/jcs.199216>; see also figure 1c and 5a, presenting invaginated and lobulated nuclei). The manuscript would benefit from including a broader discussion of the effects of macroH2A defects on the general nuclear architecture.

The title should be edited, as "egress" in virology is commonly used to refer to the egress of virions from the cell, not to the nuclear egress of capsids. Adding the words nuclear and capsid should be sufficient to address this issue.

It is unclear why preferential changes in expression of housekeeping genes would indicate "stress responses to infection". The rationale for this conclusion must be fully articulated and supported.

Statistical methods must be fully described in materials and methods and the number of biologically independent experiments must be stated in each figure.

2. Significance:

Significance (Required)

The major strengths of the manuscript lie on the comprehensive analyses of the effects of knocking histone macroH2A in the nuclear architecture and chromatin organization. These analyses indicate that peripheral heterochromatin is defective in the KO. Another strength lies on the analyses of the new heterochromatin domains in HSV-1 infected cells. The relationship between the lack of correlation between the changes in gene expression and global heterochromatin domains defined by macroH2A1 with the main conclusion is less clear.

The major weakness is that the data presented do not strongly support the conclusions. Additional experiments are required to support the main conclusion that the effects in peripheral heterochromatin result in a biologically significant effect on capsid egress. The authors should also consider that the additional experimentation may not support the conclusion that macroH2A or H3K27me3 play critical roles in the nuclear egress of capsids. Another major weakness is that the results of CUT&Tag of the viral genome are dismissed without proper justification. The authors conclude that the results invalidate the assays, but the results are consistent with cross-reactivity of the macroH2A1 antibody with another protein that interacts with the viral genomes and with H3K27me3 being associated with the viral genomes irrespectively of macroH2A1. If the authors had additional data supporting the claim that these results do not reflect cross-reactivity or association with the viral genomes, these data must be presented. Without that additional data, the conclusions are not supported and these discussions must be removed from the manuscript. The authors may still opt to not analyze any association with the viral genomes, but they should not dismiss them as artifactual without actual evidence to support this claim. Previously published literature is also misquoted.

This study makes an incremental contribution to the previously published evidence showing that HSV-1 capsids egress the nucleus through channels in between the peripheral chromatin. It shows that disruption of the heterochromatin at the nuclear periphery, and the nuclear architecture in general, may have a modest effect on capsid egress. This information may be of interest mostly to a specialized audience focused on the egress of nuclear capsids.

3. How much time do you estimate the authors will need to complete the suggested revisions:

Estimated time to Complete Revisions (Required)

(Decision Recommendation)

Between 3 and 6 months

Yes

Review #3

1. Evidence, reproducibility and clarity:

Evidence, reproducibility and clarity (Required)

Lewis et al. reveal an unexpected role for heterochromatin formation in remodeling the nucleus to facilitate egress of the nuclear-replicating virus HSV1. By performing TEM in HSV1-infected primary human fibroblasts, the authors show that capsids accumulate at the inner nuclear membrane in regions of less densely stained heterochromatin, in agreement with studies in established cell lines. The authors go on to reveal that heterochromatin in the nuclear periphery of HSV1-infected primary fibroblasts was dependent on the histone variant macroH2A1 and is enriched with H3K27me3. CUT & Tag was used to profile macroH2A1 over time during lytic HSV1 infection and showed that both macroH2A1 and H3K27me3 were enriched over newly formed heterochromatic regions 10s-100s of Kb in length in active compartments. Remarkably, loss of macroH2A1 or H3K27me3 reduced released, cell free infection virus progeny and increased intranuclear capsid accumulation without detectably impacting the proportion of mature genome containing capsids, virus genome or protein accumulation. Their finding that newly remodeled heterochromatin forms in HSV infected cells and is a critical determinant for the association of capsids with the inner nuclear membrane is consistent with a critical role in egress.

I have only relatively minor editorial suggestions listed below to improve the manuscript:

Line 92: This subtitle should be revised to more precisely state the findings shown in the Fig 1

data. While the first part of the statement "HSV1 capsids associate with regions of less dense chromatin" is consistent with what is shown, the final phrase "...to escape the nucleus" is an interpretation of the data inferred from the static image.

Line 96: I am not sure the statement that fibroblasts represent a "common" site of infection is supported by ref 15. Fibroblasts do, as indicated in ref 15, express the appropriate receptor(s) for virus entry and in culture support robust virus productive growth. However, in human tissue, infection of dermal fibroblasts appears rare, suggesting it may not be a "common" site of infection (PMCID: PMC8865408). Maybe simply revise wording to indicate fibroblasts represent "a site of infection or can be infected in tissue?".

Line 126-127: As written it states that "...regions of the host genome that increase during infection", implying these genome regions are amplified (increase). I think the authors mean that infection increases binding of mH2A1 and H3K27me3 to broad regions of the host genome. Please clarify.

FigS1, a,b,c,d: please indicate that 4,8,12 indicate hpi, correct? And indicate that in the legend M indicates Mock.

Line 197: "active compartments". Do the authors mean transcriptionally active compartments? Please clarify

Line 232: please replace "productive" with "infectious"

Line 233 - The authors conclude mH2A1 is important for egress, ruling out assembly before even bringing it up. As I read on, it is clear the authors addressed this important issue later on in the manuscript. That said, it was a bit jarring to conclude egress is important without addressing the assembly possibility at this juncture in the manuscript. One way to remedy this would be to move the Fig S6 assembly/capsid type data (lines 286-297, Fig S6) and surrounding text earlier to support the conclusion that mH2A1 did not detectably influence assembly, but is important for egress.

Line 244: "progeny production" - it would be helpful to specify "cell free or released infectious virus progeny"

Line 248: change "produced" to released"

Line 273 replace "productive" with "infectious virus progeny released from infected cells"

Fig S5c: Was the plaque assay performed on cell free supernatants? This should be indicated.

2. Significance:

Significance (Required)

The experiments are well executed, the data are solid with appropriate statistical analysis and their analysis sufficiently rigorous, and the manuscript is clearly written. Moreover, the finding that HSV manipulates host heterochromatin marks to facilitate nuclear egress is significant and exciting. The work reveals an unexpected role for newly assembled heterochromatin in egress of nuclear replicating viruses like HSV1.

3. How much time do you estimate the authors will need to complete the suggested revisions:

Estimated time to Complete Revisions (Required)

(Decision Recommendation)

Less than 1 month

No

Manuscript number: RC-2022-01723

Corresponding author(s): Daphne Avgousti, Srinivas Ramachandran

Reviewer #1 (Evidence, reproducibility and clarity (Required)):

Summary

This study by Lewis et al. examines the role of heterochromatin in the nuclear egress of herpesvirus capsids. They show that heterochromatin markers macroH2A1 and H3K27me3 are enriched at specific genome regions during the infection. They also show that when macroH2A1 is removed or H3K27me3 is depleted (both of which reduce the amount of heterochromatin at the nuclear periphery), the capsids are not able to egress as effectively. This is interesting since it could be argued that heterochromatin acts as a hindrance to the transport of viral capsids to the nuclear envelope and that the loss of it would allow capsids to reach the nuclear envelope more easily. However, this paper seems to show that heterochromatin formation, on the contrary, is necessary for efficient egress.

Overall, the study seems comprehensive. The methodology is solid, and the experiments are very well controlled. However, some issues need to be addressed before publication.

Major comments

1) In line 49, the authors state, "Like most DNA viruses, herpes simplex virus (HSV-1) takes advantage of host chromatin factors both by incorporating histones onto its genome to promote gene expression and by reorganizing host chromatin during infection". In addition, HSV1 expression can be hindered by the host's interferon response via histone modifications. Ref. Johnson KE, Bottero V, Flaherty S, Dutta S, Singh VV, Chandran B. IFI16 restricts HSV-1 replication by accumulating on the HSV-1 genome, repressing HSV-1 gene expression, and directly or indirectly modulating histone modifications. PLoS Pathog. 2014 Nov 6;10(11):e1004503. doi: 10.1371/journal.ppat.1004503. Erratum in: PLoS Pathog. 2018 Jun 6;14(6):e1007113. PMID: 25375629; PMCID: PMC4223080.

We agree with the reviewer and have amended our text and added the reference. See line 57.

2) Reference 5 is misquoted in the sentence, "This redistribution of host chromatin results in a global increase in heterochromatin". In that reference, the amount of heterochromatin is not analyzed in any way. However, that particular paper shows that the transport of capsid through chromatin is the rate-limiting step in nuclear egress, which is important considering this study. Further, the article by Aho et al. shows that when the infection proceeds capsids can more easily traverse from the replication compartment into the chromatin, which means that infection can modify chromatin for easier capsid transport. For that reason, the article is an important reference, but it needs to be cited correctly.

We agree with the reviewer that this citation was misquoted and have corrected the citation. See lines 55 and 62-64.

3) The term heterochromatin channel at lines 54, 102, and 303 is misleading since the channels seen in the original referred paper are less dense chromatin areas. Also, this term is not used in the original paper where the phenomenon was first described. These less dense interchromatin channels were found by soft-X-ray tomography imaging and analyses, not by staining.

We thank the reviewer for pointing out this discrepancy and have amended the text to accurately describe the methods used in the appropriate citations. See lines 65, 115, and 383.

4) It is difficult to visualize chromatin using TEM microscopy. The values of peripheral chromatin thickness given in Figure 1e (5-15 nm) do not seem realistic given that the thickness of just one strand of histone-wrapped DNA is 11 nm. Why are the two values for WT different? If you can get so different values for WT, it is a bit worrisome (switching the WT results between the top and bottom parts of Fig. 1e would for example result in very different conclusions on the effect of macroH2A1 KO for the thickness of the chromatin layer).

We agree with the reviewer that it is difficult to visualize chromatin by TEM. It is also important to note that comparisons can only be made between samples treated on the same day in the same way. Taking this into account, we chose to compare macroH2A1 KO cell stains to controls done at the same time, and the same for H3K27me3 depleted conditions compared to DMSO treated and prepare for EM at the same time. Visually, it is apparent that the staining in the macroH2A1 KO control cells is somewhat different than those of the H3K27me3 depleted control cells, which represents the inherent variability of this method. It is also true that one nucleosome is around 11nm, however, since the cells contain highly compacted chromatin with many other proteins present, this measurement is not appropriate to apply. Adding up the millions of nucleosomes that make up the chromosomes at 11nm each would result in a space much larger than the nucleus, therefore we focus on comparing between control and experimental conditions restricted to this assay as a relative qualitative comparison. Nevertheless, we agree with the reviewer that the notion of changing chromatin is difficult to quantify by EM and so we have taken an additional approach to test our hypothesis and confirm EM interpretations (discussed lines 391-393). We have utilized live capsid trafficking to visualize capsid movement in nuclei in the presence or absence of macroH2A1. The results from these new experiments are presented in new Figure 5 and EV5 and support our model.

5) In lines 134-137 it says that "The enrichment of macroH2A1 and H3K27me3 was observed as large domains that were gained upon viral infection (Fig 2a), suggesting that the host landscape is altered upon infection. These gains were reflected in an increase in total protein levels measured by western blot (Fig 2b)." However, the protein levels of H3K27me3 do not seem to increase during infection. In other presented data as well (Figs. 2a, 2b, 2c, S2a) it is difficult to justify the statement that H3K27me3 is enriched in infection. When this is the case, the conclusion that the amount of heterochromatin increases in the infection (the quotation above and the one in line 315) is not supported. The statement in line 315 is also not specific since it is unclear what "newly formed heterochromatin increases" means.

We agree with the reviewer that our original description was misleading. We now have edited the text to clarify that there is redistribution of macroH2A1 and H3K27me3. In the revised manuscript, we have also included mass spectrometry data mined from Kulej et al. that show peptide counts that reflect increases in the heterochromatin markers described (see new Figure EV1a). Despite this quantitative measure, upon rigorous replicates of our western blots as requested by Reviewer 2, we concluded that the increases originally described are somewhat inconsistent by western blot. This discrepancy between mass spectrometry data and western blot is likely due to the non-linear nature of antibody binding and developing of western blots by the ECL enzymatic reaction. Therefore, our revised manuscript focuses on this redistribution as a reaction to infection and stress responses instead of a global increase as the original manuscript stated. See lines 174, 182, 196, 397 and Fig EV4d in main text and discussion sections.

6) Quantitation of viral capsid location in H3K27me3-depleted cells seems somewhat arbitrary. It would have been more robust to calculate the number of capsids per unit length of the nuclear envelope with and without depletion.

We agree with the reviewer that the quantification of capsids in the H3K27me3-depleted conditions was arbitrary. In our revised manuscript, we have now repeated this quantification to accurately measure the phenotype observed, that is the chains of capsids lined up at the inner nuclear membrane. To do this, we used two measures: 1) the distance from the INM as less than 200nm and 2) the distance from other capsids as less than 300nm. Taking into account these two measures, we quantified the frequency with which multiple capsids lined up at the INM in WT and H3K27me3-depleted conditions. This is represented in the new Figure 5d. In the WT setting, we observe most often 1 single capsid at the INM, with a small fraction of 2 capsids. However, in the H3K27me3-depleted condition, we observe much greater numbers of capsids at the INM more frequently, as many as 16 at a time, leading to an average of 2-3 capsids at any single location. The source data for this figure are also provided. See lines 589 and Fig5d.

7) In lines 300-302 it says "Elegant electron microscopy work showed that HSV-1 infection induces host chromatin redistribution to the nuclear periphery^{2,8}." However, the redistribution data in reference 8 is based on soft x-ray tomography and not on electron microscopy."

We have amended the text to accurately describe the methods used in the citations. See line 384.

8) The authors bundle together the effects of macroH2A1 removal and H3K27me3 depletion by saying that they both decrease the amount of heterochromatin at the nuclear periphery and therefore hinder capsid egress. This seems overly simplistic and macroH2A1 and H3K27me3 seem to act very differently, which is manifested in the drastic difference in nuclear capsid localization between the two cases. This difference needs to be discussed more.

We agree with the reviewer that there is a nuanced difference in the effect on nuclear egress in the absence of the two heterochromatin marks. Specifically, that macroH2A1 loss results in greater numbers of capsids dispersed throughout the nucleus, whereas depletion of H3K27me3 results in capsids reaching the INM and not escaping. To examine these differences further, we have carried out live imaging of capsid trafficking in macroH2A1 KO cells compared to control and found that capsids move much more slowly, consistent with our model, see new Figure 5h-l and EV5h-i. Conversely, H3K27me3 depletion does not prevent the capsids from reaching the INM, raising the question of whether they are successfully able to dock at the nuclear egress complex (NEC). To investigate this further, we obtained an antibody against the NEC component UL34 and probed during infection in our heterochromatin disrupted conditions. We found that UL34 levels are unchanged upon loss of macroH2A1 or depletion of H3K27me3, suggesting the levels of UL34 do not account for the decrease in titers. These data are now presented in new Figure EV3g-h. Furthermore, we have amended our model to include the two different scenarios upon loss of different types of heterochromatin (see new Figure 6) and discussion of these differences. See line 428.

Minor comments

Line 45: Nuclear replicating viruses -> Nuclear-replicating viruses

Line 56: is -> are

Line 64: 25kDa -> 25 kDa

Line 159: macroH2A1 cells -> macroH2A1 KO cells

Line 289: The term gDNA is rarely used for viral DNA. Replace gDNA with viral DNA.

Line 405: 8hpi -> 8 hpi

Line 449: mm² -> μm²

"Scale bar as indicated" words can be removed in the figure legends or at least should not be repeated many times within one figure legend.

We have amended the text to address these comments. See lines 52, 68, 76, 179, 334, 513, and 585.

Reviewer #1 (Significance (Required)):

These findings would appeal to a broad audience in the field of virology. Specifically, the researcher in the fields of virus-cell and virus-nucleus interactions.

This manuscript analyses herpesvirus-induced structural changes in the chromatin structure and organization in the nucleus that are also likely to affect the intranuclear transport of viral capsids.

Reviewer #2 (Evidence, reproducibility and clarity (Required)):

The manuscript "HSV-1 exploits heterochromatin for egress" describes the effects of heterochromatin at the nuclear periphery, macroH2A1 or H3K27me3 on HSV-1 replication and egress. Knocking out macroH2A1 or depleting H3K27me3 with high concentrations of

tazemetostat depleted heterochromatin at the nuclear periphery, may not have affected HSV-1 protein expression and modestly inhibited the production of cell-free infectivity and HSV-1 genomes. macroH2A1 deposition was affected by infection, creating new heterochromatin domains which did not correlate directly with the levels of expression of the genes in them. The authors conclude that heterochromatin at the nuclear periphery dependent on macroH2A1 and H3K27me3 are critical for nuclear egress of HSV-1 capsids.

The experiments leading to the conclusion that HSV-1 capsids egress the nucleus through channels in the peripheral chromatin confirm previously published results (<https://doi.org/10.1038/srep28844>). The previously published EM micrographs show a much larger number of nuclear capsids, more consistent with the images in the classical literature, even in conditions when nuclear egress was not inhibited. Figures 1 and 4 show scarce nuclear capsids, even under the conditions when nuclear egress should be inhibited according to the model and analyses. The large enrichment in nuclear capsids in KO cells predicted by the model is not reflected in figure 4a, which shows only a modest increase in nuclear capsid density (the total number of nuclear capsids would be more informative). The number or density of nuclear capsids is not shown in H3K27 "depleted" cells. The robustness of the analyses of the number of capsids at the membrane in H3K27 "depleted" cells is unclear. For example, the analyses could be repeated with different cut offs, such as 2 or 4. If they are robust, then the conclusions will not change when the cutoff value is changed.

We appreciate the reviewer's observation that the number of capsids we show differs from those published in the publication by Myllys et al. (Sci Rep 2016 PMID 27349677). It is important to note there are several differences between our study and that of Myllys et al. that explain the difference. First, as reviewer 1 pointed out, the Myllys et al. study used three-dimensional soft X-ray tomography combined with cryogenic fluorescence and electron microscopy to observe capsids in 3D rendered nuclei. Since our method uses only single ultrathin 50nm slices of cells, we cannot visualize the total number of capsids per nucleus, rather only per slice, which is why we have averaged slices of many nuclei to generate a statistical comparison between macroH2A1 KO or H3K27me3-depleted and control cells treated at the same time (see response to reviewer 1). Furthermore, these other methods are specialized techniques for 3D imaging that are beyond the scope of our study. Second, the Myllys et al. paper used B cells which are much smaller than HFFs, lending themselves to better tomography studies but not commonly used to study HSV-1 biology. Third, the Myllys et al. paper also used a different MOI and time point than we have. Taken together, these differences account for the disparity in visualizing capsids which is why we quantified capsid number across many images. We agree with the reviewer that our quantification in the H3K27me3-depleted cells compared to control was somewhat arbitrary. As stated in the response to Reviewer 1 above, in our revised manuscript we have now repeated this quantification to accurately reflect the phenotype observed, that is the chains of capsids lined up at the inner nuclear membrane. To do this, we used two measures: 1) the distance from the INM as less than 200nm and 2) the distance from other capsids as less than 300nm. Taking into account these two measures, we quantified the frequency with which multiple capsids lined up at the INM in WT and H3K27me3-depleted conditions. This is represented in the new Figure 5d. In the WT setting, we observe most often 1

single capsid at the INM, with a small fraction of 2 capsids. However, in the H3K27me3-depleted condition, we observe much greater numbers of capsids at the INM more frequently, as many as 16 at a time, leading to an average of 2-3 capsids at any single location. The source data for this figure are also provided. See lines 589 and Fig 5d.

Furthermore, we have now also carried out live-imaging analysis of single capsids during infection which show the appropriate number of capsids expected when the full nucleus is visible. These results are presented in the new Figure 5 and EV5.

The quantitation of the western blots present no evidence of reproducibility and/or variability. The number of biologically independent experiments analyzed must be stated in each figure and the standard deviation must be presented. As presented, the results do not support the conclusions reached. The quality of western blots should also be improved. It is unclear why figure 2b shows viral gene expression in wild-type cells only, and not in KO or H3K27me3 depleted cells, which are only shown in the supplementary information. These blots presented in Figure S5a and S5b are difficult to evaluate as the signal is rather weak and the controls appear to indicate different loading levels. These blots do not appear to be consistent with the conclusions reached. Some blots (VP16, ICP0 in HFF) appear to indicate a delay in protein expression whereas others (VP16, ICP0 in RPE) appear to indicate earlier expression of higher levels. The claimed "depletion of H3K27me3 is not clear in figure S5d, in which the levels appear to be highly variable in all cases, without a consistent pattern, with no evidence of reproducibility and/or variability, and using a mostly cytoplasmic protein as loading control. All western blots should be repeated to a publication level quality, the number of independent experiments must be clearly stated in each figure, and the reproducibility and/or variability must be indicated by the standard deviation.

As reviewer 1 also pointed out, we appreciate that there is some variability with respect to the stated 'increase' in these heterochromatin marks during infection. As stated in response to reviewer 1, in our revised manuscript we have included a deeper analysis of these marks from global mass spectrometry that indicates an increase in total levels. Please see response to reviewer 1.

In the revised manuscript, we have now included mass spectrometry data mined from Kulej et al. that show peptide counts that reflect increases in the heterochromatin markers described (see new Figure EV1a). Despite this quantitative measure, upon rigorous replicates of our western blots as requested by Reviewer 2, we concluded that the increases originally described are somewhat inconsistent by western blot. This discrepancy between mass spectrometry data and western blot is likely due to the non-linear nature of antibody binding and developing of western blots by the ECL enzymatic reaction. Nevertheless, our genome-wide chromatin profiling showed consistent, reproducible, and statistically significant redistribution of macroH2A1 and H3K27me3 upon HSV-1 infection. Therefore, our revised manuscript now focuses on this redistribution as a reaction to infection and stress responses instead of a global increase as the original manuscript stated. See lines 174, 182, 196, 397 and Fig EV4b-c.

With respect to viral protein levels, although there is slight variation in the levels of VP16 or ICP0 in RPEs compared to HFFs, we do not feel that this difference is biologically significant as several other measures of viral infection progression are unchanged (viral RNA, viral genome accumulation within infected cells). Furthermore, the significant difference in titers we observe is not explained by slight differences in ICP0 or VP16. Nevertheless, to document this variability in western blot and assuage any concern of impact infection progression, we have repeated each western blot presented in the paper three separate times and used these blots to quantify each relevant protein. Graphs of western blot quantitation can be found in each figure accompanying a western blot as follows:

Western blots:

Figures 3b-c, 4ab, EV1b, EV5a

Quantitation of western blots:

Figures 3d, 4c, EV1c, EV5b-f

An enhanced analyses of the RNA-seq data, analyzing all individual genes rather than pooling them together, would provide better support to these conclusions. Then, the western blots are useful to show that the changes in mRNA result in changes in the levels of selected proteins.

We appreciate the reviewer's interest in the RNA-seq data, however, we feel that reviewer has not understood the analysis we presented in the initial submission. To clarify, we calculated fold changes for individual genes and did not pool RNA-seq data anywhere in the manuscript. We show boxplots of log₂ fold changes of individual genes. Boxplots enable summarization of the salient features of a distribution while still representing individual gene analysis. Here, the distribution being plotted is the log₂ fold change of individual genes that intersect with macroH2A1 domains that change due to infection. As such, clusters 1-3 of macroH2A1 domains feature a loss in macroH2A1 due to infection and the boxplots show that the majority of genes are upregulated. To highlight this point further, in our revised manuscript we have included volcano plots of genes intersecting with each cluster also showing the split between the number of genes significantly upregulated and downregulated in each cluster at each time point (see new Figure EV3c). As expected from the boxplots, clusters 1-3 feature a much higher fraction of genes are significantly upregulated, whereas cluster 5 features a higher fraction of genes downregulated with concomitant increase in macroH2A1 due to infection. Taken together with the gene ontology analysis (new Figure Sd), these results support our model in which macroH2A1 is deposited in active regions to block transcription and promote heterochromatin formation. To further support these conclusions, we have also carried out analysis of 4sU-RNA data generated upon salt stress or heat shock and found that the regions defined by gain of macroH2A1 (i.e. clusters 5 and 6) also exhibit significant decreases in new transcription at just 1-2 hours after treatment. These data, which are presented in new Figure EV3b-c, strongly support our model in which macroH2A1 is deposited in genes downregulated upon stress response to generate new heterochromatin.

Figure S1 raises some questions about the specificity of the macroH2A1 antibody used for

CUT&Tag. As expected CUT&Tagging the cellular genome in the KO cells with the specific antibody results in lower signal than with the IgG control antibody. In contrast, viral DNA is CUT&Tagged as efficiently in the KO as in the WT cells, and in both cases significantly above the IgG controls. The simplest interpretation of these results is that the antibody cross-reacts with a protein that binds to HSV-1 genomes. The manuscript must experimentally address this possibility.

We agree with the reviewer that there is a possibility that antibodies cross react. However, we are confident that this is not the case in this scenario for the following reasons:

1 – We have carried out immunofluorescence analysis of macroH2A1 or H3K27me3 during HSV-1 infection and observe no overlap with ICP8 staining. We have included these images together with a histogram documenting the lack of overlap in the new Figure EV2f-g.

2 – CUT&Tag relies on the Tn5 transposase to insert barcodes into accessible regions of the genome. An inherent limitation of this method during viral infection is that the replicating viral genome is very dynamic and accessible, leading to easier and less specific insertion by the transposase. This is evidenced by the pattern of signal across the viral genome that is completely overlapping in the macroH2A1, H3K27me3 and IgG conditions. Snapshots of the full viral genome are now included in the new Figure EV2c-d.

Furthermore, using CUT&Tag with macroH2A1 antibody, we expect the transposition rate to be identical between WT and macroH2A1 KO conditions for the Ecoli and viral genomes. This is because we assume that the transposition in these two genomes is non-specific since there is no macroH2A1 present. Then, we expect the spike-in normalized CUT&Tag enrichment on the viral genome to be the same between WT and macroH2A1 KO conditions. Since IgG should not be affected by macroH2A1 KO, we expect the IgG enrichment to be same between WT and macroH2A1 KO conditions. Thus, non-specific background would result in higher enrichment in an apparent signal on viral genome in the macroH2A1 KO condition.

Combined with this expectation for background transposition and the following: 1) the distribution of the CUT&Tag signal across the viral genome is virtually identical between IgG, macroH2A1, and H3K27me3 CUT&Tag signal in WT and macroH2A1 KO cells (see new Figure EV2c-d), 2) that there is no colocalization between macroH2A1 or H3K27me3 with viral genomes by immunofluorescence (see new Figure EV2f-g), and 3) the whole genome correlation of the signals across CUT&Tag samples on the viral genome, but not the host, are virtually identical as presented in a heat map (see new Figure EV1g vs EV2e), we conclude that the viral CUT&Tag signal is noise. Therefore, any analysis of the signal on the viral genomes would not be biologically meaningful.

Also, Figure S1 shows that the viral genome is CUT&Tag'ed with H3K27me3 antibody as efficiently in macro H2A1 WT and KO cells, and in both cases above the background signal from IgG control antibody. The authors conclude that the signal with the specific antibody

"mirrors" that of the control antibody, but "mirroring" is not defined and the actual data show that there is a large increase in signal with the specific antibody. Not surprisingly, the background signal also increases, as the number of genomes increase while infection progresses. The authors conclude that "these results indicated that there was a significant background signal from the viral genome that could not be accounted for", but no evidence supporting this conclusion is presented. The data show clear signal above the background from the viral genome and that this signal is not affected by the presence or absence of macroH2A1. This section of the manuscript has to be thoroughly re-analyzed as there is clear H3K27 signal.

We agree with the reviewer that as presented in the current manuscript it seems as though there is a real H3K27me3 signal. However, as stated in the above comment, the pattern of this signal matches that of all other conditions, including IgG, suggesting it is not a real signal, cross-reacted or otherwise, but rather an artifact of the methodology. See new Figure EV2.

The concentration of tazemetostat used is high. Normally, concentrations of around 1µM are used in cells, and 10µM is often cytotoxic (for example <https://doi.org/10.1038/s41419-020-03266-3>; <https://doi.org/10.1158/1535-7163.MCT-16-0840>). The effects on H3K27me3 presented in figure S1b appear to be normalized to mock infected treated cells. If so, they do not allow to evaluate the effectivity of the treatment. Cell viability after the four days treatment must be evaluated, the claimed "depletion" of H3K27me3 must be clearly demonstrated (the blots in figure S5 are not sufficient as presented), and levels of different histone methylations must be tested to support the claimed specificity of tazemetostat for H3K27me3 at the high concentrations used.

While we agree with the reviewer that the cytotoxicity of any inhibitor is an important aspect to take into account, in this instance the reviewer is incorrect. The reviewer has cited papers that highlight the potential use of tazemetostat as a cancer-cell specific treatment for colorectal and B-cell cancers. In both of these cases, the primary conclusion is that tazemetostat's cytotoxic property is largely correlated to mutation in EZH2. In fact, WT EZH2 treated cells had a more "cytostatic" response, which shows that tazemetostat is not toxic with WT EZH2 (Brach et al. Mol Cancer Ther. 2017, PMID 28835384) as is the case in our system. Furthermore, the Tan et al. study shows a non-transformed human fibroblast (CCD-18co) and embryonic colon epithelial (FHC) as "healthy controls" for their work in colorectal cancer cell lines in Figure 1D. These 2 cell lines, which are comparable to the WT HFF cells we used, show no reduction in viability at a log fold greater concentration than the 10 µM used in our paper.

Nevertheless, we agree with the reviewer that cytotoxicity should be formally ruled out. In our original experiment, we recorded cell counts at the harvested mock, 4-, 8-, and 12 hpi and found no difference in the number of cells over the course of infection (see new Figure EV3e). We also used trypan blue staining as a measure of cell viability upon tazemetostat treatment and found no toxicity. These results are presented in new Figure EV3f.

Furthermore, we agree with the reviewer that total H3 levels by western blot should be included in any comparison of H3 modification. While these were included in some figures, they were unintentionally omitted in others. In our revised manuscript we have now included these blots together with quantification of triplicate biological samples of H3K27me3 levels normalized to total H3. See new Figures 3, 4, EV1, and EV5.

Minor comments.

Reference No.27 is misquoted in lines 250-251, which state that it shows that "HSV-1 titers, but not viral replication, were reduced upon EZH2 inhibition." The reference actually shows inhibition of HSV-1 infectivity, DNA levels and mRNA for ICP4, ICP22 and ICP27. This reference uses much shorter treatments (12 h and only after infection). It also shows that inhibition of EZH2/1 up regulates expression of antiviral genes.

We appreciate that the reviewer has pointed out a discrepancy between our results using an EZH2 inhibitor (tazemetostat) and those from reference 27 (Arbuckle et al., mBio, 2017 PMID 28811345) that requires clarification. The reviewer states that the treatments were 12 hours after infection, however, this is incorrect. In the Arbuckle et al. study, the authors used multiple different inhibitors at high doses for short treatments before infection and noted that this caused an upregulation in antiviral genes that blocked infection progression of multiple viruses including HCMV, Ad5 and ZIKA. Importantly, these genes include multiple immune signaling and interferon stimulated genes. In our study, we specifically use a much lower dose of EZH2 inhibitor, with respect to the IC50 value, and waited 3 days to ensure a steady state. In our system, any initial burst of immune response from the inhibitor would likely have subsided by the time we do our infection. Furthermore, supplemental figure EV1 from the Arbuckle et al. study states that EZH1/2 inhibitors do not affect nuclear accumulation of viral genomes and suppress HSV-1 IE expression in an MOI-independent manner (Arbuckle et al. Supplemental Figure 1). These results in fact support our conclusions that it is not any antiviral effect of inhibition of EZH2 that causes the decrease in titers that we observe.

To clarify, the IC50 value of the inhibitors used in the Arbuckle et al. study are 10 nmol/L (GSK126) and 4 nmol/L (GSK343). The IC50 is a measurement used to denote the amount of drug needed to inhibit a biological process by 50% and is commonly used in pharmacology to compare drug potency. In the Arbuckle et al. study, GSK126 was used at a concentration range of 15-30 μ M, that is 1500-3000x more than the IC50 level as converted from nmol/L to μ M, and GSK343 was used at a concentration range of 20-35 μ M, that is 5000-8750x more than the IC50 level, to see changes in viral mRNA levels. The IC50 value for tazemetostat is 11 nmol/L which means that one would need to use a much higher molarity of tazemetostat, at least 28 μ M which would be 2500x the IC50 value, to achieve the comparable biological changes as the inhibitors used in the Arbuckle et al. study. Thus, we are confident that the 10 μ M concentration used in our study is an appropriate and non-toxic amount that would not impact antiviral responses at the dose and times that we used. As shown above and reported in multiple studies (for example: Knutson et al. Molecular Cancer Therapy 2014 PMID 24563539, Tan et al. Cell Death and Disease 2020 PMID 33311453 cited above, and Zhang et al. Neoplasia 2021 PMID 34246076, among others) the concentration of tazemetostat that we used is not toxic to the

cells. Importantly, it was also reported that a global decrease in H3K27me3 by EZH2 inhibition using a 10 μ M concentration of tazemetostat (here referred to by the identifier EPZ6438) did not impact HSV-1 RNA transcript accumulation measured by bulk sequencing (Gao et al. *Antiviral Res* 2020 PMID 32014498), consistent with our findings.

In our revised manuscript, we have now included a discussion of these important points. See lines 409-428.

HFF are primary human cells but they are fibroblasts whereas the primary target of HSV-1 replication is epithelial cells. The wording used "they represent a common site of infection in humans" must be edited

We agree with the reviewer and have updated the text. See lines 109.

Disruption of macroH2A (1 and 2) results in general defects in nuclear architecture, not just peripheral chromatin (<https://doi.org/10.1242/jcs.199216>);, see also figure 1c and 5a, presenting invaginated and lobulated nuclei). The manuscript would benefit from including a broader discussion of the effects of macroH2A defects on the general nuclear architecture.

We agree with the reviewer and our revised manuscript now includes a more in-depth discussion of the impact of macroH2A and other heterochromatin marks on nuclear structure. See lines 373-374 and 394.

The title should be edited, as "egress" in virology is commonly used to refer to the egress of virions from the cell, not to the nuclear egress of capsids. Adding the words nuclear and capsid should be sufficient to address this issue.

We agree with the reviewer and will update the title to read "HSV-1 exploits host heterochromatin for nuclear egress". Given that we are measuring multiple aspects of infection, we feel that adding the word 'capsid' is not necessary.

It is unclear why preferential changes in expression of housekeeping genes would indicate "stress responses to infection". The rationale for this conclusion must be fully articulated and supported.

We agree with the reviewer that it may not be immediately clear as to why changes in housekeeping gene expression represent a stress response. In a recent study that we cite in our manuscript, Hennig et al. (PLOS Path 2018 PMID 29579120) demonstrate that changes in chromatin accessibility and gene transcription during HSV-1 infection resemble those that occur upon heat shock or salt stress. These results strongly support the model that global transcription changes caused upon stress (heat, salt, infection etc.) result in dramatic alterations to chromatin structure. In support of this notion, in our revised manuscript we now include analysis of these

datasets based on our macroH2A1-defined clusters. Importantly, we found that the regions defined by gain of macroH2A1 (i.e. clusters 5 and 6) also exhibit significant decreases in new transcription at just 1-2 hours of exposure to salt and heat stress. These data, which are presented in new Figure EV3b-c, strongly support our model in which macroH2A1 is deposited on active genes to generate heterochromatin as a response to the stress of infection. We also discuss these results further in the revised manuscript, see lines 210-220, 233-236, and 424-426.

Statistical methods must be fully described in materials and methods and the number of biologically independent experiments must be stated in each figure.

We agree with the reviewer and have included these details in each figure legend.

Reviewer #2 (Significance (Required)):

The major strengths of the manuscript lie on the comprehensive analyses of the effects of knocking histone macroH2A in the nuclear architecture and chromatin organization. These analyses indicate that peripheral heterochromatin is defective in the KO. Another strength lies on the analyses of the new heterochromatin domains in HSV-1 infected cells. The relationship between the lack of correlation between the changes in gene expression and global heterochromatin domains defined by macroH2A1 with the main conclusion is less clear.

The major weakness is that the data presented do not strongly support the conclusions. Additional experiments are required to support the main conclusion that the effects in peripheral heterochromatin result in a biologically significant effect on capsid egress. The authors should also consider that the additional experimentation may not support the conclusion that macroH2A or H3K27me3 play critical roles in the nuclear egress of capsids.

To support our conclusions, we have carried out an entirely different set of experiments to track capsid movement. Bosse et al. PNAS 2015 PMID 26438852 and Aho et al. PLOS Path 2021 PMID 34910768 use live-imaging and single-particle tracking to characterize capsid motion relative to host chromatin. These approaches allowed the authors to discover that infection-induced chromatin modifications promote capsid translocation to the INM. They showed that 1) HSV-1 infection alters host heterochromatin such that open space is induced at heterochromatin boundaries, termed "corrals", in which viral capsids diffuse and 2) the movement of viral capsids through the host heterochromatin is the rate limiting step in HSV-1 nuclear egress.

To test our hypothesis that macroH2A1-dependent heterochromatin specifically is required, we collaborated with Dr. Jens Bosse to carry out these same experiments in our macroH2A1 KO and paired control cells. We tracked RFP-VP26 using spinning-disk confocal live imaging to track individual capsid movement within the nucleus. We found that capsids in cells lacking macroH2A1 traveled much shorter distances on average. This is represented graphically by the mean-square displacement (MSD) of capsid movement in macroH2A1 KO cells plateauing at $\sim 0.4 \mu\text{m}^2$ vs $0.6 \mu\text{m}^2$ in WT cells, which represents the size of the "corral", or space through

which capsids diffuse. The average corral size in macroH2A1 KO cells is ~300 nm less than the average corral size in WT cells (two-thirds the size). These results are consistent with the finding that macroH2A1 limits chromatin plasticity both in vitro (Muthurajan et al. J Biol Chem 2011 PMID 21532035) and in cells (Kozlowski et al. EMBO Rep 2018 PMID 30177554). These data strongly support our hypothesis that macroH2A1-dependent heterochromatin is critical for the translocation of HSV-1 capsids through the host chromatin to reach the INM. Furthermore, these data support the model in which macroH2A1 allows for the increase of open space induced during infection. Loss of this open space restricts the movement of capsids in the nucleus, as quantified by our live-imaging experiments. These data are now included in the new Figure 5 and EV5 and described in lines 348-372 and 1011-1037.

NOTE: These experiments were done in a separate lab using the same cells and MOI we used for our TEM studies. It is important to note that because this was done by live imaging where the full nucleus and cell are visible, the appropriate number of capsids is apparent.

Another major weakness is that the results of CUT&Tag of the viral genome are dismissed without proper justification. The authors conclude that the results invalidate the assays, but the results are consistent with cross-reactivity of the macroH2A1 antibody with another protein that interacts with the viral genomes and with H3K27me3 being associated with the viral genomes irrespectively of macroH2A1.

We agree with the reviewer that as presented the viral genome reads were dismissed without thorough justification. As stated above, we are confident that the patterns we detected do not represent a biologically relevant signal but rather an artifact of the experimental set up. Furthermore, it is well known in the field that normalizing replicating viral genomes during lytic infection in any kind of chromatin profiling technique is fraught with inconsistencies as each cell may have a different copy number of viral genomes at any given time point. Therefore, we feel strongly that any analysis of the viral genome chromatin profile during a lytic replication at this point in time would require single cell sequencing which is beyond the scope of this study. We appreciate that this was not clearly presented in the original manuscript and in our revised submission we have included a full supplemental figure documenting the negative data that support our conclusions (see new Figure EV2).

If the authors had additional data supporting the claim that these results do not reflect cross-reactivity or association with the viral genomes, these data must be presented. Without that additional data, the conclusions are not supported and these discussions must be removed from the manuscript. The authors may still opt to not analyze any association with the viral genomes, but they should not dismiss them as artifactual without actual evidence to support this claim. Previously published literature is also misquoted.

This study makes an incremental contribution to the previously published evidence showing that HSV-1 capsids egress the nucleus through channels in between the peripheral chromatin. It shows that disruption of the heterochromatin at the nuclear periphery, and the nuclear

architecture in general, may have a modest effect on capsid egress. This information may be of interest mostly to a specialized audience focused on the egress of nuclear capsids.

While we agree with the reviewer on many points as stated above, we respectfully disagree that our study is merely an incremental contribution of interest only to a specialized audience focused on nuclear egress. As reviewer 2 states earlier, the strength of our study lies in the “comprehensive analyses of the effects of knocking histone macroH2A in the nuclear architecture and chromatin organization”, which would be of interest to a broad chromatin audience as well as virologists. Together with the new data presented here and a revised manuscript, we feel that our study would be of interest to a broad audience in the chromatin and virology fields as reviewers 1 and 3 also pointed out. Chromatin is generally analyzed in the context of how it might affect gene expression and the impact of chromatin on biological processes such as viral infections, and its structural role in the nucleus is not commonly considered. Here, we demonstrate an important example of the glaring effects of chromatin structure on the biological nuclear process of infection.

Reviewer #3 (Evidence, reproducibility and clarity (Required)):

Lewis et al. reveal an unexpected role for heterochromatin formation in remodeling the nucleus to facilitate egress of the nuclear-replicating virus HSV1. By performing TEM in HSV1-infected primary human fibroblasts, the authors show that capsids accumulate at the inner nuclear membrane in regions of less densely stained heterochromatin, in agreement with studies in established cell lines. The authors go on to reveal that heterochromatin in the nuclear periphery of HSV1-infected primary fibroblasts was dependent on the histone variant macroH2A1 and is enriched with H3K27me3. CUT & Tag was used to profile macroH2A1 over time during lytic HSV1 infection and showed that both macroH2A1 and H3K27me3 were enriched over newly formed heterochromatic regions 10s-100s of Kb in length in active compartments. Remarkably, loss of macroH2A1 or H3K27me3 reduced released, cell free infection virus progeny and increased intranuclear capsid accumulation without detectably impacting the proportion of mature genome containing capsids, virus genome or protein accumulation. Their finding that newly remodeled heterochromatin forms in HSV infected cells and is a critical determinant for the association of capsids with the inner nuclear membrane is consistent with a critical role in egress.

I have only relatively minor editorial suggestions listed below to improve the manuscript:

Line 92: This subtitle should be revised to more precisely state the findings shown in the Fig 1 data. While the first part of the statement "HSV1 capsids associate with regions of less dense chromatin" is consistent with what is shown, the final phrase "...to escape the nucleus" is an interpretation of the data inferred from the static image.

We agree with the reviewer and have amended our text to more accurately describe the figure. See lines 138-139.

Line 96: I am not sure the statement that fibroblasts represent a "common" site of infection is supported by ref 15. Fibroblasts do, as indicated in ref 15, express the appropriate receptor(s) for virus entry and in culture support robust virus productive growth. However, in human tissue, infection of dermal fibroblasts appears rare, suggesting it may not be a "common" site of infection (PMCID: PMC8865408). Maybe simply revise wording to indicate fibroblasts represent "a site of infection or can be infected in tissue?".

We agree with the reviewer, as was also pointed out by reviewer 2, and have amended the text. See lines 109.

Line 126-127: As written it states that "...regions of the host genome that increase during infection", implying these genome regions are amplified (increase). I think the authors mean that infection increases binding of mH2A1 and H3K27me3 to broad regions of the host genome. Please clarify.

We agree with the reviewer that this was written ambiguously. As was pointed out by reviewers 1 and 2, the increase in these marks depends on the type of measurement. Therefore, we have modified the text in a revised manuscript to focus instead on the redistribution of these marks during infection. See line 138-139.

FlgS1, a,b,c,d: please indicate that 4,8,12 indicate hpi, correct? And indicate that in the legend M indicates Mock.

This is correct and we have updated this in the figure legend. See lines 625-627.

Line 197: "active compartments". Do the authors mean transcriptionally active compartments? Please clarify

This is correct and have clarified this in the text. See line 248.

Line 232: please replace "productive" with "infectious"

We agree with the reviewer and have amended our text. See line 295.

Line 233 - The authors conclude mH2A1 is important for egress, ruling out assembly before even bringing it up. As I read on, it is clear the authors addressed this important issue later on in the manuscript. That said, it was a bit jarring to conclude egress is important without addressing the assembly possibility at this juncture in the manuscript. One way to remedy this would be to move the Fig S6 assembly/capsid type data (lines 286-297, Fig S6) and surrounding text earlier to support the conclusion that mH2A1 did not detectably influence assembly, but is important for egress.

We agree with the reviewer that the order of presentation makes it difficult to follow. Our revised manuscript now includes these important data within the same figure. See new Figure 5.

Line 244: "progeny production" - it would be helpful to specify "cell free or released infectious virus progeny"

Line 248: change "produced" to released"

Line 273 replace "productive" with "infectious virus progeny released from infected cells"

Fig S5c: Was the plaque assay performed on cell free supernatants? This should be indicated.

We agree with the reviewer and have made all these changes in the text. See lines 285-287.

Reviewer #3 (Significance (Required)):

The experiments are well executed, the data are solid with appropriate statistical analysis and their analysis sufficiently rigorous, and the manuscript is clearly written. Moreover, the finding that HSV manipulates host heterochromatin marks to facilitate nuclear egress is significant and exciting. The work reveals an unexpected role for newly assembled heterochromatin in egress of nuclear replicating viruses like HSV1.

Manuscript number: RC-2022-01723

Corresponding author(s): Daphne Avgousti, Srinivas Ramachandran

For ease of reading, we have listed a point-by-point response to all reviewer comments and color coded the revisions we have completed in green, those we plan to do in orange, and those we cannot complete in a reasonable time in red.

Referee #1:

Summary

This study by Lewis et al. examines the role of heterochromatin in the nuclear egress of herpesvirus capsids. They show that heterochromatin markers macroH2A1 and H3K27me3 are enriched at specific genome regions during the infection. They also show that when macroH2A1 is removed or H3K27me3 is depleted (both of which reduce the amount of heterochromatin at the nuclear periphery), the capsids are not able to egress as effectively. This is interesting since it could be argued that heterochromatin acts as a hindrance to the transport of viral capsids to the nuclear envelope and that the loss of it would allow capsids to reach the nuclear envelope more easily. However, this paper seems to show that heterochromatin formation, on the contrary, is necessary for efficient egress.

Overall, the study seems comprehensive. Besides our earlier comments which have already been replied to and corrected by the authors, we have the following comments which need to be addressed before publication.

Major comments:

In line 62 it is said that in Myllys et al. 2016 the viral capsids are associated with the channels, but in reality, capsids were not observed in the study. The study merely shows that the channels emerge in late infection. The following sentence:

"Viral capsids traversing this dense chromatin in the nuclear periphery are associated with areas of less dense chromatin, also termed channels (Myllys et al, 2016)."

could be modified for example as this:

"The progress of HSV-1 infection is associated with areas of less dense chromatin in the nuclear periphery, also termed channels (Myllys et al,2016)."

We agree with the reviewer and have amended our text to the suggestion.

Similarly, in lines 111-113, it is claimed that in the studies of Myllys et al. and Aho et al., "viral capsids were observed to interact with the inner nuclear membrane in regions of less dense staining". However, as mentioned above, in the paper by Myllys the capsid localization was not analyzed. In the study of Aho et al., it was shown by simulation that capsids can reach the inner nuclear membrane via channels at the marginalized chromatin. Therefore the following sentence:

"These results are consistent with previous reports in African green monkey kidney cells (Vero) and human B cells where viral capsids were observed to interact with the inner nuclear

membrane in regions of less dense staining (Aho et al, 2017; Myllys et al, 2016)." should be edited for example as follows:

"These results are consistent with a previous report in African green monkey kidney cells (Vero) showing that viral capsids can reach the inner nuclear membrane via channels in the marginalized chromatin(Aho et al, 2017)."

We agree with the reviewer and have amended our text to the suggestion.

In Figure 5h, it seems that the capsid motion was analyzed in the whole nucleus. In infection, the chromatin is usually marginalized towards the nuclear periphery, and most of the capsids are moving in the replication compartment where there is practically no chromatin. How does this affect your conclusions on the capsid motion data since the data is used to draw conclusions on chromatin structure?

We thank the reviewer for bringing this important point to our attention. While it is true that the entire nucleus has been imaged in our live-tracking experiments, and not just the periphery, the periphery is also part of the whole nucleus. It is possible that capsids move more freely in the replication compartments than at the periphery. We did not classify individual tracks into chromatin-rich or empty regions as Aho et al. (2021) recently did. Rather, we assessed the average mean square displacement (MSD) for all particles detected within the confocal volume. Despite this approach, we observed a significant impact of macroH2A1 knockout on the MSD, which is likely due to a portion of capsids located in the nuclear periphery, as also depicted in Figure 5l. Thus, our measurements average all these movements together. Thus, our interpretation is that when averaged all together, the movement is still quite slower in the absence of macroH2A1-dependent heterochromatin, which is consistent with our model and the notion presented by the reviewer that capsids move more freely in the replication compartment. We will include this information in the discussion section of a revised manuscript.

In the capsid type segmentation of Figure 5f, it seems that probably many b-type capsids have been segmented as c-type. In Fig. 5a for example, you can see that most of the shown capsids are b-type (the center of the capsid is less electron-dense than the scaffold). Some are just more strongly stained than others. Also, the b-type example capsid shown in Fig 5e looks more like a c-type. In 5c the capsids are oversaturated due to imaging settings or faulty sample preparation, making it difficult to analyze the different types. In previous studies, it has been shown that most of the capsids are of b type (Aho et al. 2021, mentioned also in Cardone et al. Adv Exp Med Biol. 2012; 726: 423-439.). Please check the data again in this respect.

We agree with the reviewer that there is some oversaturation in the images presented in our revised manuscript and that most capsids should be b-type and not c-type. In a revised manuscript, we will rerun this analysis. We are confident that this is a minor error and will be corrected.

Minor comments

1. Line 590: Man-Whitney -> Mann-Whitney
2. Infection time point is not given for Figure 5 EM data

We agree with the reviewer and have amended our text accordingly.

Referee #2:

Lewis et al describes the analyses of the proposed roles of macroH2A1 and H3K27me3 in the nuclear egress of HSV-1 capsids by modulating cellular heterochromatin. The revised manuscript addresses some of the previous critiques and attempts to address some others. Additional experiments and analyses are included and some results are re-analyzed. Nonetheless, some of the main conclusions are not properly supported, reproducibility of many results is not described, and some of the analyses appear not to be sufficiently systematic. The manuscript also appears to lack a central focus, digressing between the effects of HSV-1 infection in host heterochromatin as evaluated by macroH2A1 and H3K27 and the roles of heterochromatin as evaluated by these markers and TEM on HSV-1 nuclear egress. Only the major points are discussed.

Figure 3f shows that there is a decrease in extracellular HSV-1 genomes in infections of macroH2A1 KO or tazemetostat-treated cells while figure 3e shows that there is no increase in intracellular genomes. Figure 4d and e show the same combination of results for clinical isolates from low and high shed patients. Figure EV5 does not analyze intracellular or extracellular genomes. These results fail to support the proposed model that the treatments inhibit capsid egress from the nucleus. The model would predict an increase in intracellular genomes concomitantly with the decrease in extracellular ones. Intranuclear encapsidated genomes may eventually be degraded, but if so the capsids would have to be degraded as well and would not be visible in TEM.

While we respectfully disagree with the reviewer that our manuscript lacks focus, we do acknowledge that there is a point to be discussed here. That is in that we do not detect higher numbers of genomes in the HSV-1 infected macroH2A1 KO cells despite lower genomes and infectious progeny in the supernatants. The reason for this is that we have already reached the maximum detection limit of our system at 8hpi such that there is no significant difference in the genomes detected between 8 and 12hpi (See Fig 4d). This means that we would also not be able to detect a decrease in the macroH2A1 KO cells or tazemetostat treated because there is no effect on viral replication. In fact, this is consistent with what has been previously reported as the bottleneck for HSV-1 egress: chromatin. Because replication is already progressing normally and at the highest rate the cell can achieve, it is very unlikely that we would detect any higher levels of viral genomes in the nucleus. In summary, we do not see an increase in viral genomes in the macroH2A1 KO cells because (1) we have already reached the detection limit and (2) the bottleneck is in fact egress itself. We will include this information in the discussion section of a revised manuscript.

Reproducibility is not addressed. Figures 1 or 2 do not mention the number of independent experiments. The n stated in figure 1f and g does not specify whether it refers to sections, nuclei, or biologically independent experiments. Figure 3a lists the number of biological replicates, but panels b-h do not. Figure 4c, d, and f describe the error bars as SD of three biological replicates, and reaches the conclusion that there are no statistical differences, but the power to detect any differences is not discussed. Panel 4f does not list the number of biological replicates. The n mentioned in figure 5 b and d do not specify whether they refer to sections, nuclei, or biological replicates. Panel h describes the number of tracks and nuclei, but does not specify whether equal number of tracks were evaluated in each nuclei or whether there were any biological replicates. The other panels in figure 6 do not indicate the number of biological replicates. Similar critique applies to the extended view figures. Materials and methods describe three biological replicates for RNA seq and two for CUT&TAG but the presentation does not properly clarify whether one of the replicates is presented, or the results are averaged in some way.

The reviewer is incorrect in this case. As mentioned in Methods under “CUT&Tag”: “Two biological replicates per time point were obtained from independent infections.” Thus, the data from two replicates were averaged for CUT&Tag. For RNAseq, the replicates were analyzed separately to obtain p-values and FDR, as described in the figure legend.

The description of the average of the mock infected cells refers to setting cutoffs for domain definition only. There is no indication that there is enough statistical power to detect changes in gene expression at the protein or transcript level in most cases.

The reviewer is again incorrect as we clearly state that we have enough statistical power to detect changes in transcript levels. As described in the Methods: “Genes were deemed differentially expressed if fold changes were greater than 2 in either direction and Benjamini-Hochberg adjusted p-values were less than 0.01.” As seen in volcano plots in S3C, we are able to robustly detect differentially expressed genes due to viral infection.

Error bars are large indicating a significant variability between experiments. It is not clear if the levels of all transcripts increase to a statistically significant extent between 4 and 8 hours post-infection.

Once again the reviewer has overlooked the statistical description in the figure legend. As clearly stated in the figure legend for Figure S3C: “Kolmogrov-Smirnov test (in R) was performed comparing logCPM values of genes at each time point with logCPM values of genes from mock dataset for each cluster. The p-values were corrected for multiple testing, and the time points and clusters with corrected p-value less than 0.05 are shaded.”, the comparisons were made to mock dataset. We did not perform comparisons between 4 and 8 hours post infection. The Reviewer offers no context where this comparison would be meaningful in the current manuscript.

The results presented in figure 4d and e are described as modest differences, but there is no statistical analyses to support the conclusion. The conclusion that the presumed heterochromatin accumulating at the nuclear periphery during HSV-1 infections represents new regions of heterochromatin that are macroH2A1 and H3K27me3 dependent is unsupported. The authors show that there are changes in the bulk heterochromatin, but CUT&TAG provides no spatial information and no FISH or similar evidence is presented to support the conclusion. The authors have revised the criteria for the quantification of the intranuclear capsids. However, the quantification in the revised manuscript still appears insufficiently systematic, the results appear variable, and the batch effect appears to be important. As the previous review indicated, capsid density in the control sample of the tazemetostat-treated cells appears to be no different from that in the macroH2A1-KO cells, in experiments performed in different days. Figure 5b presents only a minimal difference in nuclear capsid density in WT or KO cells and the number of independent sections, cells, or biologically independent repeats analyzed is not described. The "n" listed for control and KO cells are not defined and therefore it is not possible to evaluate the biological reproducibility of the marginal differences with large distribution overlap between the WT and macroH2A1 KO cells. Panel h presents "representative" live-cell images, but there is no evaluation of the reproducibility or variability of these results. Intranuclear capsid density in macroH2A KO cells, or number of capsids "at the inner nuclear membrane" in tazemetostat-treated cells (referred to as "H3K27me3-depleted"), are not presented. One would expect from the proposed model that the tazemetostat-treated cells would have a higher total nuclear capsid density as well and therefore this information is critical to the manuscript. Also, it is not clear why figure 5b is presented as box and whiskers and figure 5d as bar graphs. The nuclear morphology of the macroH2A1 KO and tazemetostat-treated cell presented in figures 1 and 5 are quite dissimilar, with the latter being thin and elongated, differences which may affect the intranuclear capsid localization in so far as the volume to surface area would differ in the different nuclear morphologies presented.

The rebuttal letter states that PMID 2885384 shows that tazemetostat is not toxic with WT EZH2 in PMID 28835384, but Table 1 in that publication shows that tazemetostat was cytotoxic to WT EZH2 cells at concentrations between 0.5 to 5.5 μ M after 7 days of exposure and at lower concentrations after 14 days.

The reviewer is mistaken with respect to the relevance of PMID 28835384 to our study. We performed tazemetostat treatment for 4 days. With PRC2 inhibition, one expects a 2-fold reduction in H3K27me3 with each cell division. With 7 days treatment (assuming a 24 hour doubling time), one would expect a 128-fold reduction in H3K27me3 compared to 16-fold reduction due to our treatment time. Thus, the long exposure data is not relevant to our experiments. As shown in Figure 1C of PMID 28835384, at 4 days of treatment with WT EZH2, there is no difference in viability for multiple cell lines. We directly show no loss of viability of the cells we use at 4 days of treatment in Figure 4f. Thus, the Reviewer's point is irrelevant to our manuscript.

The other manuscript cited as showing that non-transformed fibroblasts and embryonic colon epithelial cells showed no reduction in viability at a log fold greater concentration than the 10 μ M

used in the experiments presented has been just retracted (March 29) because of lack of confidence in the results presented (<https://www.nature.com/articles/s41419-023-05764-6>). This citation must be removed from the manuscript without prejudice.

This manuscript was retracted after our submission. We have now removed this citation from our manuscript.

The authors cite Gao et al., 2020 as showing that there was no change in viral gene expression when cells were treated with tazemetostat prior to infection, but the reference cited shows a decrease in H3K27me3 after only 4h tazemetostat treatment without indicating whether treatment was before or after infection. Regardless of previous results obtained with different cell lines, the authors must properly present the results evaluating the cytotoxicity of tazemetostat to the cells used under the conditions used. Figure EV4e presents cell count, but not viability, during 12h, and EV4f cell viability, but not cell counts, for 4 days. 12 hours is insufficient to detect changes in cell number or viability in most cases, whereas the data presented in EV4f does not evaluate cell doubling. The data that has to be presented is the daily number of viable cells during the four day treatment. The results of these studies may show that 10 μ M is not toxic to these cells for four days, or that it has some degree of toxicity, which must be included in the analyses and interpretation of the results.

We respectfully disagree that tazemetostat is toxic in our system. As figure EV4f clearly shows, and is supported by other literature as described above, and in our manuscript, we do not observe any significant difference in cell counts or viability.

The manuscript has been edited to highlight the technical artifacts of the CUT&TAG of the viral genome, and the results supporting this conclusion are presented as expanded view figures. These changes address a previous apparently major concern. However, the new figure EV2 panels f and g appear to show lower levels of expression of ICP8 in the macroH2A or tazemetostat-treated cells than in their respective controls, which is inconsistent with the western blots in figures 3, 4, EV1 and EV4, as well as the conclusion that HSV-1 protein expression is not affected by these treatments. This new internal inconsistency, which may be a technical artifact, must be addressed. The rebuttal letter describes that transcription of each genes was evaluated independently, but the original critique appears to have been directed to the clustering of the genes in six different groups for analyses.

We respectfully point out that the reviewer is inconsistent with themselves making it impossible for us to respond. Nevertheless, we will attempt to address the discrepancy in two points:

- (1) First, to quote the original review regarding this point in entirety: “An enhanced analyses of the RNA-seq data, analyzing all individual genes rather than pooling them together, would provide better support to these conclusions.” This sentence does not mention “clustering of the genes in six different groups for analyses.” We can only reply to the critique text and not what was in the Reviewer’s mind. Thus, our rebuttal was appropriate.*

(2) Second, the Reviewer is still incorrect. We did not cluster genes. Clustering genes would be antithetical to our analysis philosophy for chromatin domains. We clustered macroH2A chromatin domains first. And then analyzed expression of genes that fall within these domains.

The revised manuscript still discusses the anticorrelation between transcription levels and macroH2A1 by analyzing the clusters and adds volcano plots. The latter identify genes that are up or downregulated in each cluster, but do not analyze the correlation between transcription and macroH2A1 levels on a gene. Scatter analyses of the transcription levels and macroH2A1 in individual genes appears more appropriate to support the conclusion reached regarding the correlation between macroH2A1 and transcription.

We disagree with the reviewer. Since macroH2A1 deposition is across large regions of chromosomes, quantification of macroH2A1 at gene level is not appropriate. Importantly, the expression changes at specific domains occurs even in macroH2A KO (Figure S3b), which clearly argues for transcriptional changes to be upstream of macroH2A deposition.

Note that the volcano plot in figure 3a do not show up in the manuscript for review. As an editorial point, but in need of correction, the redistribution of cellular chromatin to the nuclear periphery had been described long before the two papers cited in the discussion, such as Monier et al., Nature cell biology 2000 and dating as far back as the 1960's (Schwartz and Roizman, 1969 Virology; Sirtori et al., 1967 Cancer Research).

We respectfully disagree with Reviewer 2 that our statistical analyses lack sufficient power or that our results are not reproducible. We have presented three biological replicates and indicated as such in every appropriate instance. We agree that the retracted paper, which was retracted after our paper was submitted, was problematic and have removed this from our manuscript. We disagree that the articles mentioned from the 60s are referring to the same type of chromatin analysis we are showing in this manuscript with substantially more advanced methodology and analysis.

June 7, 2023

RE: JCB Manuscript #202304106T

Dr. Daphne Avgousti
Fred Hutchinson Cancer Center
1100 Fairview Avenue North C1-201
Seattle, WA 98107

Dear Dr. Avgousti:

Thank you for submitting your revised manuscript entitled "HSV-1 exploits host heterochromatin for nuclear egress". We would be happy to publish your paper in JCB pending final revisions necessary to meet our formatting guidelines (see details below) and text changes requested by the new reviewer. As this reviewer was satisfied overall, we will evaluate text changes requested without further input from either the new reviewer or the original reviewers.

A. MANUSCRIPT ORGANIZATION AND FORMATTING:

Full guidelines are available on our Instructions for Authors page, <http://jcb.rupress.org/submission-guidelines#revised>. Submission of a paper that does not conform to JCB guidelines will delay the acceptance of your manuscript.

1) Text limits: Character count for Articles is < 40,000, not including spaces. Count includes abstract, introduction, results, discussion, and acknowledgments. Count does not include title page, figure legends, materials and methods, references, tables, or supplemental legends.

2) Figures limits: Articles may have up to 10 main figures and 5 supplemental figures/tables.

3) Figure formatting: Scale bars must be present on all microscopy images, including inset magnifications. Molecular weight or nucleic acid size markers must be included on all gel electrophoresis. Please avoid pairing red and green for images and graphs to ensure legibility for color-blind readers. If red and green are paired for images, please ensure that the particular red and green hues used in micrographs are distinctive with any of the colorblind types. If not, please modify colors accordingly or provide separate images of the individual channels.

** Please include molecular weight markers with all Western blots.

4) Statistical analysis: Error bars on graphic representations of numerical data must be clearly described in the figure legend. The number of independent data points (n) represented in a graph must be indicated in the legend. Statistical methods should be explained in full in the materials and methods. For figures presenting pooled data the statistical measure should be defined in the figure legends. Please also be sure to indicate the statistical tests used in each of your experiments (either in the figure legend itself or in a separate methods section) as well as the parameters of the test (for example, if you ran a t-test, please indicate if it was one- or two-sided, etc.). Also, if you used parametric tests, please indicate if the data distribution was tested for normality (and if so, how). If not, you must state something to the effect that "Data distribution was assumed to be normal but this was not formally tested."

** Please describe error bars and statistical tests used in Fig 1F/G, Fig 5B/D and Supp Fig 5G.

** Please describe error bars in Fig 4F, Fig 5F/G, Supp Fig 1D/E, Supp Fig 2A/B, Supp Fig 3A/B, and Supp Fig 4.

5) Abstract and title: The abstract should be no longer than 160 words and should communicate the significance of the paper for a general audience. The title should be less than 100 characters including spaces. Make the title concise but accessible to a general readership.

6) Materials and methods: Should be comprehensive and not simply reference a previous publication for details on how an experiment was performed. Please provide full descriptions in the text for readers who may not have access to referenced manuscripts. We also provide a report from SciScore and an associate score, which we encourage you to use as a means of evaluating and improving the methods section.

** When citing prior publications in the methods section, please ensure the relevant methodology is still described in full.

7) Please be sure to provide the sequences for all of your primers/oligos and RNAi constructs in the materials and methods. You must also indicate in the methods the source, species, and catalog numbers (where appropriate) for all of your antibodies. Please also indicate the acquisition and quantification methods for immunoblotting/western blots.

8) Microscope image acquisition: The following information must be provided about the acquisition and processing of images:

- Make and model of microscope
- Type, magnification, and numerical aperture of the objective lenses
- Temperature
- Imaging medium
- Fluorochromes
- Camera make and model
- Acquisition software
- Any software used for image processing subsequent to data acquisition. Please include details and types of operations involved (e.g., type of deconvolution, 3D reconstitutions, surface or volume rendering, gamma adjustments, etc.).

10) Supplemental materials: There are strict limits on the allowable amount of supplemental data. Articles may have up to 5 supplemental figures. Please also note that tables, like figures, should be provided as individual, editable files. A summary of all supplemental material should appear at the end of the Materials and methods section.

13) ORCID IDs: ORCID IDs are unique identifiers allowing researchers to create a record of their various scholarly contributions in a single place. At resubmission of your final files, please consider providing an ORCID ID for as many contributing authors as possible.

Please note that JCB now requires authors to submit Source Data used to generate figures containing gels and Western blots with all revised manuscripts. This Source Data consists of fully uncropped and unprocessed images for each gel/blot displayed in the main and supplemental figures. Since your paper includes cropped gel and/or blot images, please be sure to provide one Source Data file for each figure that contains gels and/or blots along with your revised manuscript files. File names for Source Data figures should be alphanumeric without any spaces or special characters (i.e., SourceDataF#, where F# refers to the associated main figure number or SourceDataFS# for those associated with Supplementary figures). The lanes of the gels/blots should be labeled as they are in the associated figure, the place where cropping was applied should be marked (with a box), and molecular weight/size standards should be labeled wherever possible. Source Data files will be directly linked to specific figures in the published article.

Journal of Cell Biology now requires a data availability statement for all research article submissions. These statements will be published in the article directly above the Acknowledgments. The statement should address all data underlying the research presented in the manuscript. Please visit the JCB instructions for authors for guidelines and examples of statements at (<https://rupress.org/jcb/pages/editorial-policies#data-availability-statement>).

B. FINAL FILES:

Thank you for this interesting contribution, we look forward to publishing your paper in Journal of Cell Biology.

Sincerely,

Andres Leschziner
Monitoring Editor
Journal of Cell Biology

Tim Fessenden
Scientific Editor
Journal of Cell Biology

Reviewer #1 (Comments to the Authors (Required)):

This study by Lewis et al. focuses on the role of heterochromatin in the nuclear egress of herpesvirus capsids. They show that heterochromatin formation in the nuclear periphery, in particular with macroH2A1 and H3K27me3 markers, are important for capsids to access the inner nuclear membrane.

The depletion of H3K27 interestingly leads to impressive lining up of capsids at the inner nuclear membrane, while the absence of macroH2A1 leads to capsids scattered in the nucleus. These heterochromatin marks are required for the secretion of new virions and viral genomes, but not for intracellular replication or protein production. These results clearly highlight a role for heterochromatin in viral egress.

The revised study is comprehensive and the data added in the revised version of the manuscript are really helping to support the conclusions. In particular, live capsid tracking is a good complementary experimental approach to the transmission electron microscopy.

All responses to previous reviewers are satisfactory and a revised manuscript with an extended discussion will improve the clarity of results' interpretation.

Reviewer #1

All responses to reviewer #1 are satisfactory and a revised manuscript containing the comments mentioned in the discussion will improve the clarity of results' interpretation.

Reviewer #2

The fact that there are not more viral genomes detected inside the cells KO for MacroH2A1 or treated with tazemetostat is

puzzling, as there should be an accumulation of the viral genomes/products not secreted. The authors gave explanations for that, which they propose to include in the discussion of a revised manuscript. I would be in favor of this option in the absence of other obvious experimental way to address the question.

The discussion on the toxicity of the tazemetostat treatment is well addressed in the rebuttal letter. From the data provided, there is no major effect on cell viability with the doses and time used.

The authors have to make sure that all other points on statistics and reproducibility of the results have been addressed in the results, legends and methods sections of the article.

Manuscript number: JCB Manuscript #202304106T

Corresponding author(s): Daphne Avgousti, Srinivas Ramachandran

Reviewer #1:

Summary

This study by Lewis et al. examines the role of heterochromatin in the nuclear egress of herpesvirus capsids. They show that heterochromatin markers macroH2A1 and H3K27me3 are enriched at specific genome regions during the infection. They also show that when macroH2A1 is removed or H3K27me3 is depleted (both of which reduce the amount of heterochromatin at the nuclear periphery), the capsids are not able to egress as effectively. This is interesting since it could be argued that heterochromatin acts as a hindrance to the transport of viral capsids to the nuclear envelope and that the loss of it would allow capsids to reach the nuclear envelope more easily. However, this paper seems to show that heterochromatin formation, on the contrary, is necessary for efficient egress.

Overall, the study seems comprehensive. Besides our earlier comments which have already been replied to and corrected by the authors, we have the following comments which need to be addressed before publication.

Major comments:

In line 62 it is said that in Myllys et al. 2016 the viral capsids are associated with the channels, but in reality, capsids were not observed in the study. The study merely shows that the channels emerge in late infection. The following sentence:

"Viral capsids traversing this dense chromatin in the nuclear periphery are associated with areas of less dense chromatin, also termed channels (Myllys et al, 2016)."

could be modified for example as this:

"The progress of HSV-1 infection is associated with areas of less dense chromatin in the nuclear periphery, also termed channels (Myllys et al,2016)."

We agree with the reviewer and have amended our text to the suggestion.

Similarly, in lines 111-113, it is claimed that in the studies of Myllys et al. and Aho et al., "viral capsids were observed to interact with the inner nuclear membrane in regions of less dense staining". However, as mentioned above, in the paper by Myllys the capsid localization was not analyzed. In the study of Aho et al., it was shown by simulation that capsids can reach the inner nuclear membrane via channels at the marginalized chromatin. Therefore the following sentence:

"These results are consistent with previous reports in African green monkey kidney cells (Vero) and human B cells where viral capsids were observed to interact with the inner nuclear membrane in regions of less dense staining (Aho et al, 2017; Myllys et al, 2016)."

should be edited for example as follows:

"These results are consistent with a previous report in African green monkey kidney cells (Vero) showing that viral capsids can reach the inner nuclear membrane via channels in the

marginalized chromatin(Aho et al, 2017)."

We agree with the reviewer and have amended our text to the suggestion.

In Figure 5h, it seems that the capsid motion was analyzed in the whole nucleus. In infection, the chromatin is usually marginalized towards the nuclear periphery, and most of the capsids are moving in the replication compartment where there is practically no chromatin. How does this affect your conclusions on the capsid motion data since the data is used to draw conclusions on chromatin structure?

We thank the reviewer for bringing this important point to our attention. While it is true that the entire nucleus has been imaged in our live-tracking experiments, and not just the periphery, the periphery is also part of the whole nucleus. It is possible that capsids move more freely in the replication compartments than at the periphery. We did not classify individual tracks into chromatin-rich or empty regions as Aho et al. (2021) recently did. Rather, we assessed the average mean square displacement (MSD) for all particles detected within the confocal volume. Despite this approach, we observed a significant impact of macroH2A1 knockout on the MSD, which is likely due to a portion of capsids located in the nuclear periphery, as also depicted in Figure 5l. Our measurements average all these movements together. Thus, our interpretation is that when averaged all together, the movement is still quite slower in the absence of macroH2A1-dependent heterochromatin, which is consistent with our model and the notion presented by the reviewer that capsids move more freely in the replication compartment. We have included this information in the discussion section of our revised manuscript.

In the capsid type segmentation of Figure 5f, it seems that probably many b-type capsids have been segmented as c-type. In Fig. 5a for example, you can see that most of the shown capsids are b-type (the center of the capsid is less electron-dense than the scaffold). Some are just more strongly stained than others. Also, the b-type example capsid shown in Fig 5e looks more like a c-type. In 5c the capsids are oversaturated due to imaging settings or faulty sample preparation, making it difficult to analyze the different types. In previous studies, it has been shown that most of the capsids are of b type (Aho et al. 2021, mentioned also in Cardone et al. Adv Exp Med Biol. 2012; 726: 423-439.). Please check the data again in this respect.

We agree with the reviewer that there is some oversaturation in the images presented in our revised manuscript and that most capsids should be b-type and not c-type. In our revised manuscript, we have rerun this analysis and found that indeed the B type was the most abundant. Importantly, the proportions were still unchanged in the different conditions indicating that capsid formation is not affected by heterochromatin disruption.

Minor comments

1. Line 590: Man-Whitney -> Mann-Whitney
2. Infection time point is not given for Figure 5 EM data

We agree with the reviewer and have amended our text accordingly.

Reviewer #4:

This study by Lewis et al. focuses on the role of heterochromatin in the nuclear egress of herpesvirus capsids. They show that heterochromatin formation in the nuclear periphery, in particular with macroH2A1 and H3K27me3 markers, are important for capsids to access the inner nuclear membrane.

The depletion of H3K27 interestingly leads to impressive lining up of capsids at the inner nuclear membrane, while the absence of macroH2A1 leads to capsids scattered in the nucleus. These heterochromatin marks are required for the secretion of new virions and viral genomes, but not for intracellular replication or protein production. These results clearly highlight a role for heterochromatin in viral egress.

The revised study is comprehensive and the data added in the revised version of the manuscript are really helping to support the conclusions. In particular, live capsid tracking is a good complementary experimental approach to the transmission electron microscopy.

All responses to previous reviewers are satisfactory and a revised manuscript with an extended discussion will improve the clarity of results' interpretation.

Reviewer #1

All responses to reviewer #1 are satisfactory and a revised manuscript containing the comments mentioned in the discussion will improve the clarity of results' interpretation.

We have added the discussion points as requested by reviewer 1, see above.

Reviewer #2

The fact that there are not more viral genomes detected inside the cells KO for MacroH2A1 or treated with tazemetostat is puzzling, as there should be an accumulation of the viral genomes/products not secreted. The authors gave explanations for that, which they propose to include in the discussion of a revised manuscript. I would be in favor of this option in the absence of other obvious experimental way to address the question.

We agree with the reviewer and have added this to the revised manuscript.

The discussion on the toxicity of the tazemetostat treatment is well addressed in the rebuttal letter. From the data provided, there is no major effect on cell viability with the doses and time used.

The authors have to make sure that all other points on statistics and reproducibility of the results have been addressed in the results, legends and methods sections of the article.

We agree with the reviewer and have amended our text accordingly to reflect all reproducibility and statistical tests conducted.